# Fully implanted battery-free high power platform for chronic spinal and muscular functional electrical stimulation

Alex Burton [1,9], Zhong Wang [2,9], Dan Song [3], Sam Tran [2], Jessica Hanna[1], Dhrubo Ahmad [1], Jakob Bakall[1], David Clausen[1], Jerry Anderson[1], Roberto Peralta[1], Kirtana Sandepudi[3], Alex Benedetto[4], Ethan Yang[2], Diya Basrai[2], Lee E. Miller[2,3,4,5], Matthew C. Tresch [3,5,6] ✉ & Philipp Gutruf [1,7,8] ✉

Electrical stimulation of the neuromuscular system holds promise for both scientific and therapeutic biomedical applications. Supplying and maintaining the power necessary to drive stimulation chronically is a fundamental challenge in these applications, especially when high voltages or currents are required. Wireless systems, in which energy is supplied through near field power transfer, could eliminate complications caused by battery packs or external connections, but currently do not provide the harvested power and voltages required for applications such as muscle stimulation. Here, we introduce a passive resonator optimized power transfer design that overcomes these limitations, enabling voltage compliances of ± 20 V and power over 300 mW at device volumes of 0.2 cm², thereby improving power transfer 500% over previous systems. We show that this improved performance enables multichannel, biphasic, current-controlled operation at clinically relevant voltage and current ranges with digital control and telemetry in freely behaving animals. Preliminary chronic results indicate that implanted devices remain operational over 6 weeks in both intact and spinal cord injured rats and are capable of producing fine control of spinal and muscle stimulation.

Electrical stimulation of the central and peripheral nervous system is a highly successful approach for treating a wide range of clinical disorders in humans, such as restoration of hearing[1], electroanalgesia[2], or suppression of motor disorders such as Parkinson's[3]. These approaches often use electrical stimulation to either replace lost electrical neural signals, as in cochlear implants, or to interrupt dysfunctional neural circuits, as in deep brain stimulation. However, there is a growing appreciation for potential applications using electrical stimulation not only to restore lost function, but also to retrain or rehabilitate residual neural circuits by enhancing endogenous neuroplasticity or neuromodulation[4–6]. The promise of such approaches is that a transient intervention involving continuous targeted electrical modulation might induce changes in neural function that accelerate rehabilitation[6,7] or even deliver outcomes not possible with current approaches.

One fundamental challenge in these applications is supplying and maintaining the power necessary to source the currents and voltages

[1]Department of Biomedical Engineering, University of Arizona, Tucson, AZ 85721, USA. [2]Department of Neuroscience, Northwestern University, Chicago, IL 60611, USA. [3]Department of Biomedical Engineering, Northwestern University, Evanston, IL 60208, USA. [4]Interdepartmental Neuroscience, Northwestern University, Chicago, IL 60611, USA. [5]Department of Physical Medicine and Rehabilitation, Northwestern University, Chicago, IL 60611, USA. [6]Shirley Ryan AbilityLab, Chicago, IL 60611, USA. [7]Bio5 Institute and Department of Neurology, University of Arizona, Tucson, AZ 85721, USA. [8]Department of Electrical and Computer Engineering, University of Arizona, Tucson, AZ 85721, USA. [9]These authors contributed equally: Alex Burton, Zhong Wang. ✉e-mail: m-tresch@northwestern.edu; pgutruf@email.arizona.edu

required to achieve therapeutic outcomes, especially in applications involving large currents and voltages over extended time periods. Many human applications use large batteries that must be periodically recharged or replaced, which can create barriers to adoption by patients[8]. In preclinical experiments in small animals such as rats and mice, this challenge is even more critical, since battery size is limited and frequent battery replacement will interfere with animals' behavior. Alternatively, power can be supplied in small animals through external cabled connections, but this approach increases risk of infection, causes the formation of scar tissue[9], can displace implanted electrodes[10], limits physical activities[11] and is incompatible with magnetic resonance imaging[12,13]. The use of either batteries or wired connections, therefore, makes it difficult to evaluate neuromodulation or rehabilitation strategies involving continuous, uninterrupted stimulation protocols lasting multiple days or weeks.

Wireless systems, in which stimulation energy is supplied through near-field power transfer, have the potential to overcome these limitations, enabling chronic applications without the need for batteries or external connections. However, the amount of power that can currently be harvested in these systems is limited, since the maximum power point voltage (i.e. the voltage at which most power can be harvested) depends on the inductance ratio and coupling between transmitter and receiver antennas[14]. This dependence results in a tradeoff between the operational voltage of a device and its power harvesting ability, making it difficult to engineer systems that can supply voltages and power in the desired range. There is therefore a critical need for technologies that can increase the amount of transferred power at the desired operational voltage suitable for chronic functional electrical stimulation.

There are additional challenges in applications for functional electrical stimulation using chronically implanted devices. Standard fabrication approaches for neural interface devices typically result in stiff, inflexible devices which limit body motion or cause discomfort which can affect functional outcomes[15]. The recent development of thin flexible circuits and soft, bio-compatible materials allows the fabrication of conformable interfaces for placement within the central and peripheral nervous systems. Importantly, this class of devices can also be powered and controlled wirelessly for fully implanted systems[16–19]. These flexible devices can be nearly imperceptible to the host due to the reduced micromotions minimizing tissue irritation[15,20] and their increased bio-compatibility enables longer functional lifetimes, making them suitable for long-term applications.

Finally, many functional electrical stimulation applications involve complex stimulation protocols, delivered across multiple locations in the central and peripheral nervous system in precise temporal relationships and with highly regulated stimulation levels. Current battery-free systems typically use passive, unregulated voltage control[21] or relatively simple voltage programming[16,22,23]. Existing devices capable of delivering controlled currents with precise and versatile programming require application-specific integrated circuits (ASICs) that are expensive and do not provide a general platform that can be used across a wide range of applications[24].

Additional challenges arise in applications targeting multiple areas that require very different currents or voltages; depending on the density of cell bodies and axons surrounding a stimulation electrode, the distance between cells and electrode as well as the electrical interface, required currents can widely vary[25,26]. For example, in the case of intraspinal stimulation, where the electrode is situated near many neurons and axons, functional currents can range from 0.5 μA[27] to 15 μA[28]. Epidural spinal stimulation electrodes, which contact fewer cells than intraspinal stimulation but still contact a relatively high density of fibers, use current amplitudes typically ranging from 50 to 500 μA[29–31]. Muscle stimulation electrodes, which may contact and recruit even fewer highly branched axons, can require currents up to 4.5 mA in rats[32–34]. Furthermore, electrode impedances and their current amplitudes may require voltages from a few mV to tens of volts[16,28,31]. Devices are typically capable of delivering either low or high ranges of currents/voltages, but not both ranges simultaneously that allow for simultaneous electrical stimulation of multiple neuromuscular sites and for multichannel abilities.

Here, we introduce fully implanted devices that overcome these challenges, providing a voltage range of up to ±20 V, current-controlled stimulation with a range of 40 μA to 4.7 mA, and independent control of 8 channels in a fully implantable package that is powered at distance. This device represents the highest figure of merit for all categories (Supplementary Table 1)[23,24,35–41], making it suitable for long term, uninterrupted functional electrical applications for preclinical investigations in freely moving small to medium animal subjects. The devices exploit a passive resonator scheme that substantially increases power harvesting capabilities at a designed voltage, making it possible to stimulate multiple areas of the nervous system simultaneously, using the same implanted device. The circuit design incorporates off-the-shelf components providing scalable manufacturing at a cost that enables single-use deployment which is critical for chronic implantation that make reuse impossible or difficult. The devices are fully implanted, powered and controlled wirelessly, and are fabricated from flexible electronics. The materials and mechanical design of the flexible substrate provides a platform that integrates components to conform with the surrounding soft tissue using a network of ridged islands and flexible interconnects. As a consequence, the devices have minimal impact on overall animal behavior and can remain operational for extended periods of at least 6 weeks. Further, the devices are able to deliver highly versatile, precisely controlled, and charge-balanced stimulation and are designed for scalable manufacturing that enables easily deployable one-time-use systems to deliver long-term stimulation. Taken together, these advances in device performance enable the development and evaluation of a range of complex chronic stimulation protocols for functional restoration that have previously been unattainable. We demonstrate the capabilities of this device in one biomedical application: electrical stimulation of the spinal cord and muscles in freely behaving and spinal cord injured rats, showing that this device is capable of producing functional limb movements from spinal and muscle stimulation for over 6 weeks indicating good chronic stability and laying the foundation for further validation studies.

## Results
### Wireless FES device overview
In this work, wireless, battery-free designs and a monolithic device structure, are leveraged to create a soft, biocompatible, flexible device class that can be fully implanted in highly mobile areas of a rat, such as the back and hind leg. The systems offer a platform to deliver effective stimulation in freely moving subjects without affecting mobility or behavior, making them suitable for chronic experimental research. The system has immediate application to neuroprosthetics and neuromodulation therapies relevant to a wide range of clinical applications, including spinal cord injury (SCI). Figure 1a displays a rendering of a device implanted subcutaneously in the posterior lumbar region of a rat. The device has the capability for both spinal and muscular functional electrical stimulation (FES) using an eight-channel electrode array positioned over the spinal cord and two intramuscular electrodes in the biceps femoris posterior (Fig. 1a). This device can accommodate electrode designs for μA to mA of stimulation using both microelectrodes (<200 μm) for precise stimulation of a group of neurons and macro-electrodes (>200 μm) for intramuscular stimulation. This multimodal stimulation capability enables a wide range of strategies for using electrical stimulation for the restoration or rehabilitation of motor function. We use a monopolar configuration with a common return electrode placed in the thorax, as is common in rehabilitation applications[42]. In order to maximize the mechanical flexibility of the

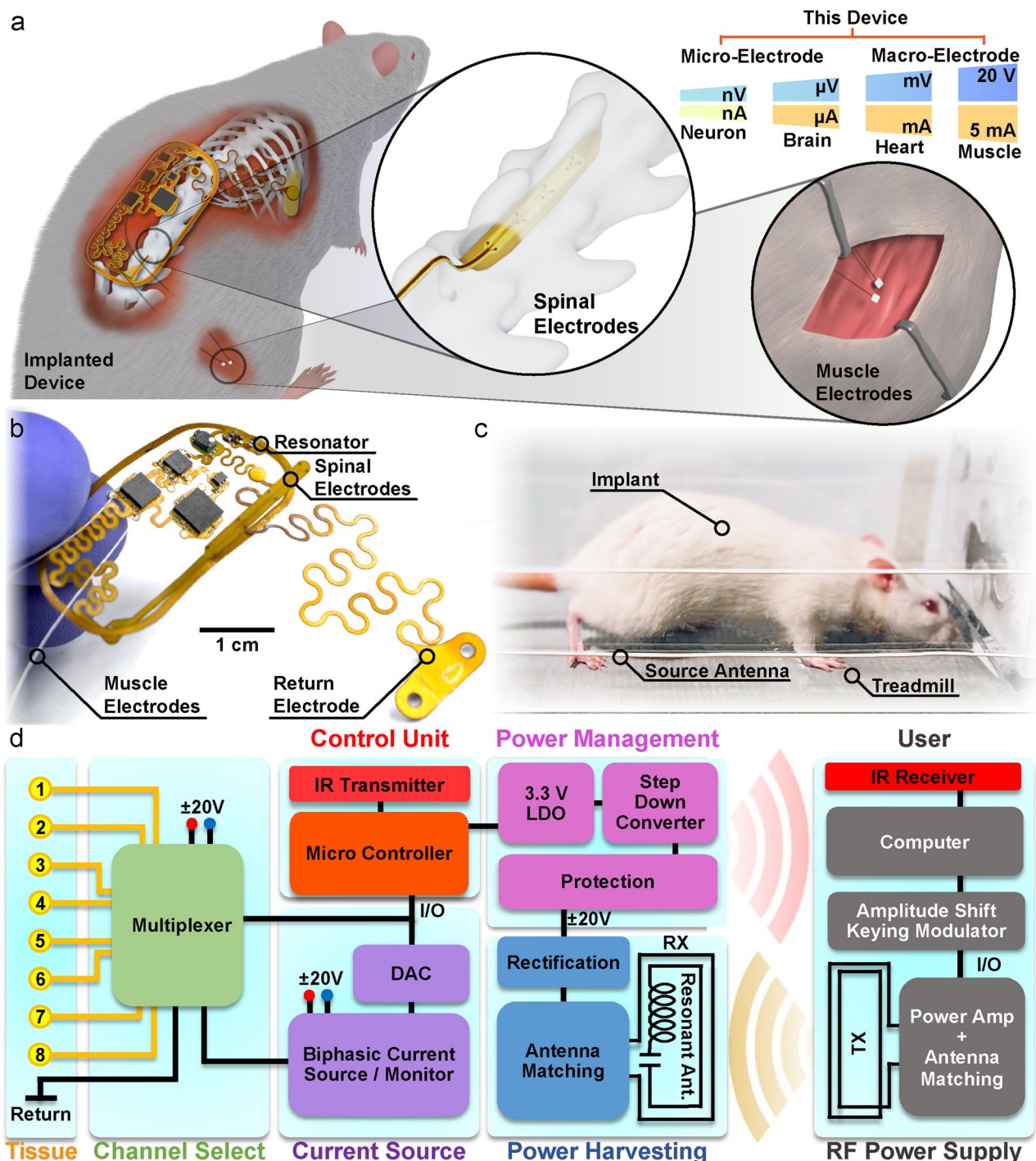

**Fig. 1 | Wireless battery-free functional electrical stimulation overview.**
**a** Rendering of implant in the small animal subject with rendered insets of electrode placement in the spinal cord and muscle. **b** Photograph image of the device. **c** Photograph of the animal running on a treadmill. **d** Block diagram of electrical functionalities of the device.

device and enable chronic stability, the integrated circuits (IC) in the device are placed in separate rigid islands and interconnected with strain-isolating serpentine traces[43,44] (Fig. 1b; Supplementary Fig. 1; see detailed component description in Supplementary Fig. 2a, b). Self-similar serpentines utilize a fractal design with repetitive patterning of interconnected curvatures that evenly distribute internal strain within the flexible substrate. We utilize this patterning structure to connect the device body to the spinal electrodes and the return electrode, to improve strain isolation, to facilitate easy implantation of the electrodes, and to improve device lifetimes[14,18]. The 20 mm × 40 mm device

platform consists of a monolithic, dual-sided, flexible circuit board (height: 100 μm) composed of two rolled annealed copper layers separated by a polyimide film, enabling scalable manufacturing with off-the-shelf components. This fully implanted, battery-free design, with its flexible mechanics and low displacement volume (0.2 cm³), minimizes the impact of the implanted device on animal appearance, spontaneous exploratory activity levels, and locomotor kinematics. These features enable free-running experiments with continuous, modulated stimulation for therapeutic treatments, such as FES rehabilitation during treadmill locomotion in animals with SCI (Fig. 1c).

The operating principle of the device, as shown in Fig. 1d, includes a system that delivers continuous wireless power to the implant using an instrumentalized cage and communicates using a custom amplitude shift keying (ASK) modulation of the radio frequency (RF) power. In order to improve power harvesting efficiency at the voltages necessary for peripheral stimulation, the device features a co-planar antenna that combines a passive resonator and receiver, using concepts from strongly coupled magnetic resonant systems to improve coupling and lower voltage of the maximum power point[45–48]. This design dramatically increases usable delivered power to the device, enabling dual voltage supplies up to ±20 V to drive stable biphasic currents over a wide range of electrode designs.

The dual voltage supply is achieved using a center-tapped antenna design with two single half-bridge rectifiers to create the positive and negative voltages with no external components and only minimally increased device footprint. The receiver antenna uses a center-tapped antenna to minimize circuit complexity while allowing for dual voltage supplies. A dual supply coupled with a precision current driver allows for a small circuit design (30 mm²) suitable for implantation while providing higher power efficiencies compared to designs using DC/DC converters to generate dual voltage supplies[49]. This is significantly smaller than traditional systems using a single supply to have the same biphasic stimulation response using a Howland current pump, H-bridge, and current monitor each requiring additional electrical traces and passive components ( > 50 mm²)[50–52]. A power management system incorporates voltage protection using Zener diodes and a step-down converter accompanied by a linear voltage regulator for stable operating voltages for the digital circuit. The low-power microcontroller allows the device to be programmed through ASK modulation without additional hardware and with minimal additional power consumption by schemes developed in our previous work[16] utilizing internal electrically erasable programmable read-only memory (EEPROM). The microcontroller uses a serial interface to communicate with an 8-bit digital-to-analog converter (DAC) and multiplexer in order to control stimulation parameters. This design enables the specification of arbitrary stimulation patterns, distributed across multiple channels, and at a wide range of currents and voltages. Information about real-time device performance is digitally transmitted transdermally through infrared (IR) communication[17]. Information on supply voltage and voltage applied to the electrode can be used to evaluate fault conditions (electrode shorts or disconnects, device failure), and to ensure accurate delivery of desired stimulation. Additionally, electrode impedances can be computed for the evaluation of stability in chronic applications.

## Resonant antenna design and electrical characterization

Wireless, battery-free platforms capable of delivering a wide range of currents and with high voltage compliances are critical for stimulation applications involving different excitable tissues. In the case of multimodal FES, for example, currents and voltages can vary by orders of magnitude when activating peripheral nerve, epidural spinal cord, or intramuscular electrodes. Although high voltage compliances are necessary for these applications, they are challenging for traditional 2-antenna near field systems, since their operational voltage depends on the inductance ratio and coupling between transmitter and co-planar antennas which is limited, especially for small animals. In order to overcome these challenges, we introduce a design that imbeds a passive resonator directly on the implanted device, improving power transfer efficiency and enabling high voltage compliances. The monolithic integration removes the need for auxiliary circuit components such as resonators that are located elsewhere in the tissue. This design can also be easily implanted, enabling it to be used as a scalable platform for chronic applications. This design also requires only a small number of passive components, resulting in small device footprints and high efficiencies compared to active approaches such as

maximum powerpoint tracking and management ICs. The device is optimized to operate in a treadmill enclosure (56 cm × 16 cm) with a 2-turn transmitter antenna with 4 cm spacing between the turns, and is optimized to match the range of vertical positions of the implanted device during normal behavior (Supplementary Fig. 3). Effective power transfer can be also be achieved with other transmitter antenna designs, for example using circular antennas in surgical settings during device implantation (Supplementary Fig. 4a–c)[14,18]. Figure 2a shows both the antenna setup used for wireless power transfer (WPT) and IR receivers located on the corners of the cage used to collect telemetry data from the implanted devices[17].

The benefits of using a resonant 3-antenna system, with passive resonator co-located on the implanted device, over the conventional 2-antenna system are coupling enhancement, better misalignment insensitivity and better bandwidth[53,54]. Figure 2b and Supplementary Fig. 5a, b illustrate the simulated B-field of the 3-antenna system, in which the receiver antenna and the resonator are co-located on the top and bottom sides of the implant. The geometric parameters of the implant antenna (i.e., the width, the gap between adjacent turns, the dimension and shape of the antennas) are chosen to optimize transmission efficiency at 13.56 MHz (Supplementary Fig. 6a, b) while minimizing the size of the implant. Adding a resonator improves the coupling between the transmitter and receiver as shown in simulations (Supplementary Fig. 7a)[54–57]. A comparison between the transmission efficiency of the 3-antenna WPT and the conventional 2-antenna WPT shows that adding the resonator antenna increases the transmission efficiency from 12% to 37% (Supplementary Fig. 7b). Although similar transmission efficiencies can be achieved in traditional 2-antenna systems by increasing the number of turns in the receiving antenna (e.g., using 8-turns increases transmission efficiency to 30%) the higher inductance caused by the increased turns substantially increases the voltage at the maximum power point, as shown in Fig. 2c. Such high voltages are not practical in most highly miniaturized designs. It is therefore difficult in traditional 2-antenna designs to increase power transfer efficiency while maintaining desired voltage levels.

Providing continuous power delivery is challenging for exploratory research in the large arenas necessary for complex behavioral testing or for evaluating behavior in larger animal models[58]. By increasing the efficiency of power casting, our design enables effective device performance over large areas, such as typical rat treadmill enclosures or home cages. The spatial distribution of harvested power within the treadmill enclosure exhibits a minimum of 120 mW in the center of the enclosure with maximum voltage compliance of ±20 V using a 2 kΩ resistor to match the operational load of the system during continuous operation while supplying the transmitter antenna 2 W of RF power (Supplementary Fig. 7c). The ability of this design to operate with low RF power makes setups substantially more energy efficient, a feature that is especially important for chronic applications. If needed, power at the implant can be increased by increasing RF power of the transmitter up to a maximum safe specific absorption rate of <20 mW kg⁻¹, which corresponds to an RF output power greater than the capabilities of the power amplifier (max 12 W) used in this work[58]. By increasing RF power to 6 W, up to 325 mW of power can be harvested in the center of the enclosure (Supplementary Fig. 7d), which is 5 times greater than the power consumption of comparable devices in the literature, providing a large operational margin to enable continuous and uninterrupted stimulation[59]. As a comparison, recently reported devices using a smaller transmitter antenna (45 cm × 12 cm; 540 cm²) and similarly sized receiver antenna (3.5 cm × 2.5 cm; 8.75 cm²) were able to harvest 50 mW using 4 W of RF power in the transmitter[59]. The device described here uses a larger transmitter antenna (56 cm × 16 cm; 896 cm²) and similar receiver (4 cm × 2 cm; 8 cm²) as compared to receivers used in a previous work[59] (3.5 cm × 2.5 cm; 8.75 cm²), however, this device is capable of harvesting 250 mW using 4 W of RF power in comparison to 50 mW at 4 W

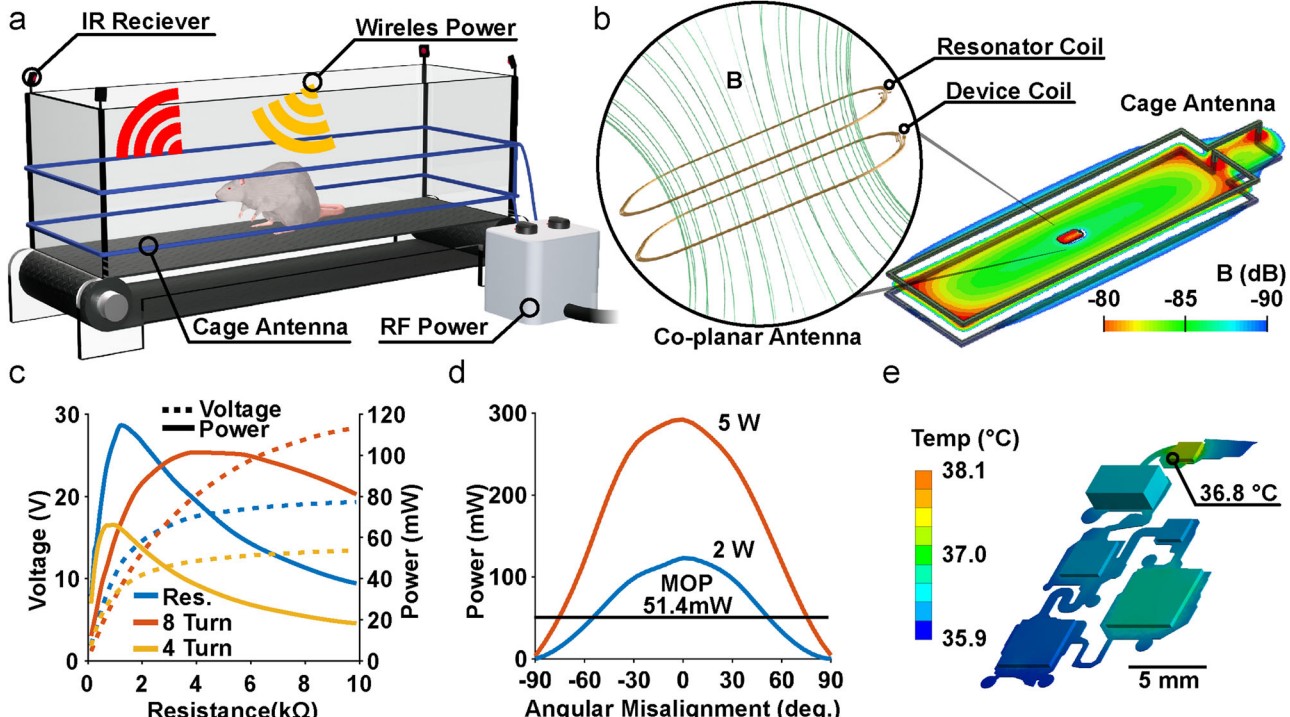

**Fig. 2 | Characterization of the antenna design and electrical device properties. a** Rendering of treadmill cage design with a dual loop transmitter antenna and infrared receivers. **b** Rendering of co-planar resonant antenna design showing the transmitter and receiver antenna coupling in Finite Element simulations of electromagnetic field flux. **c** Corresponding power vs. load curve in the center of 56 cm × 16 cm arena with RF input power of 2 W for the 3-coil WPT and 2-coil WPT. **d** Power harvesting capability vs. angular misalignments for both 2 W and 5 W of RF power. **e** Finite element simulations and measurements (displayed in numeric datapoint representing the hottest location measured) at steady state in saline solution when powered with 6 W of RF power.

of RF power in that previous study[59]. This is an increase of more than 500% harvested power over previous designs (5.71 mW cm$^{-2}$ to 31.25 mW cm$^{-2}$), enabling effective power harvesting in large arenas[59]. This performance is enabled by monolithically integrated passive resonator WPT introduced here that boosts power harvesting efficiencies with no increase to device footprint, requiring only an additional copper layer for the resonant antenna. This extraordinary harvesting capability allows for more sophisticated behavioral paradigms in complex 3-dimensional behavioral arenas, such as elevated, or even submerged, mazes[8,29]. The increased power harvesting capability also improves operational device stability during behaviors such as rearing, climbing, and preening, which often result in misalignment between antennae and a reduction of harvested power. The behavior that causes the greatest misalignment is rearing which represents 1% of total behavior time during light periods and 7% in dark periods[60,61]. To obtain an average of 51 mW using our design, sufficient for high-powered device operation, a 2 W RF power transmitter antenna can operate at an angle up to 54.4 deg; however, if higher misalignment tolerance is needed RF power can be increased to 5 W to enable stable operation up to ±75.3 deg as shown in Fig. 2d. Thus, with adjustments to the field power, our design can achieve substantial power transfer even for behaviors such as rearing which likely reflects the most severe case of misalignment. Additionally, devices implanted on the back of rats are subjected to bending. Physiological characterization describes a maximum possible bending radius of 3.4 cm (See below) which results in a reduction of power harvesting capabilities by ~20% as shown in Supplementary Fig. 8. These variations in device orientation and curvature can be compensated for by using additional RF power at the cost of increased energy consumption.

The average current consumption of the device on startup is 2.5 mA. During stimulation it averages 3.3 mA when supplied with ±20 V, as shown in Supplementary Fig. 7e and Supplementary Fig. 9a. The low power consumption of the digital components of the device is due to an efficient step-down regulation of the voltage from 20 V to 3.3 V as well as the low (4 MHz) operating frequency of the microcontroller clock and sleep events that suspend peripheral functionality of the microcontroller, using concepts from previous works[16,17]. Low power consumption and high voltage compliance (Supplementary Fig. 9b) also enable continuous data streaming of telemetric information such as voltage compliance and electrode impedance as shown in Supplementary Fig. 7f to monitor electrode status and current delivery. Thermal load (Supplementary Fig. 10a) on the surrounding tissue is tested from 2 W to 6 W RF input power with benchtop experiments and validated with steady-state thermal simulation (Supplementary Fig. 10a–c) indicating a minimal increase in surface temperatures (< 0.8 °C), well within safety requirements for implanted medical devices according to the American Association of Medical Instrumentation (Fig. 2e)[62,63].

## Stimulation control and delivery

Combined RF ASK and IR modulation[16–18], as developed in previous low power wireless systems, provides reliable two-way communication with the implants in a small footprint and with low power consumption as shown in Fig. 3a. The device is programmed via RF ASK by transmitting 3 bits for selection of a stimulation parameter (channel, amplitude, pulse width or period) followed by 8 bits to select the desired value. Information on stimulation parameters, voltage compliance, and fault conditions are transmitted back using IR communication.

This digital communication enables a multifunctional platform which can be reconfigured with different electrode characteristics according to application requirements. Figure 3b shows a range of electrode variations in monopolar configuration, all using a large surface area return electrode to reduce return path stimulation effects. The device is capable of controlling up to eight channels which can be distributed to a variety of peripheral and central nervous system

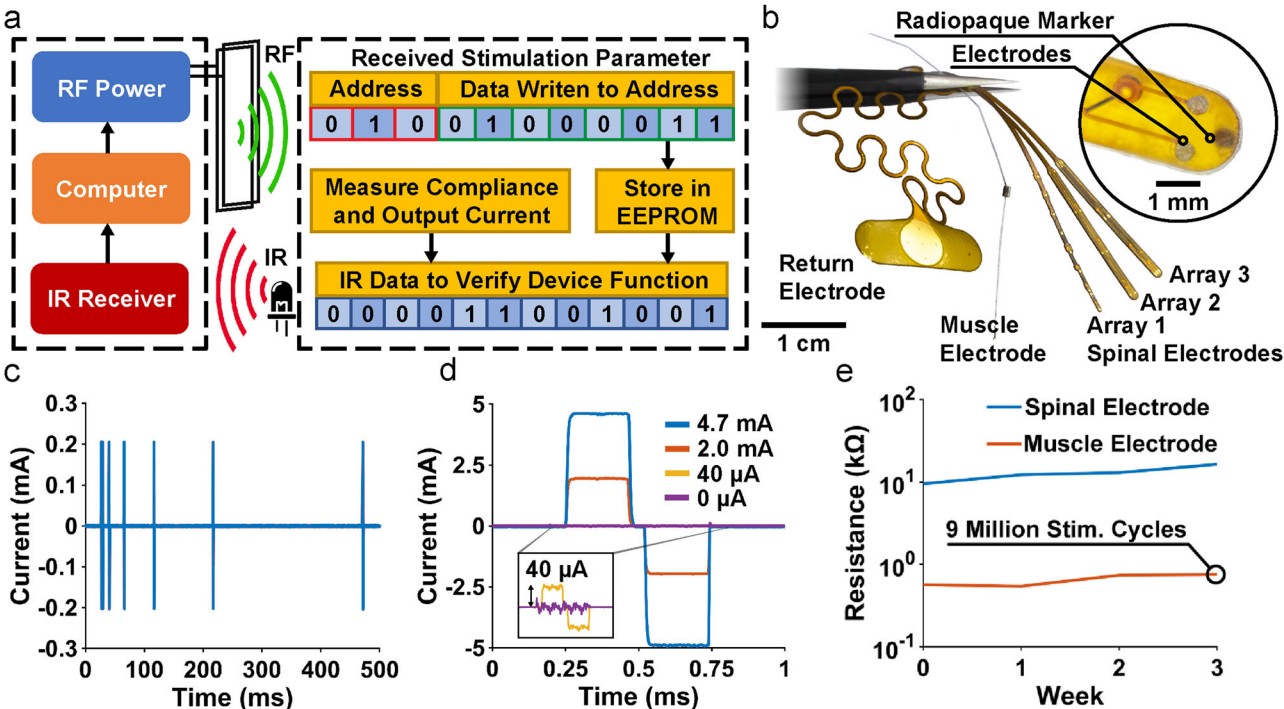

**Fig. 3 | Wireless communication, electrode design and characterization. a** Block diagram of communication scheme. **b** Photograph of return, spinal and muscle electrodes. **c** Temporal control of stimulation patterns. **d** Amplitude control of biphasic stimulation. **e** Change in the impedance of the electrode, measured at 1 kHz, after continuous stimulation of over 9 million cycles.

substrates. In this work, the device delivers stimulation to both central (spinal epidural) and peripheral (intramuscular) electrodes. For spinal stimulation, micro-electrodes with 200 μm diameter are placed epidurally on the spinal cord with a contact impedance of ~10 kΩ at 1 kHz as shown in Supplementary Fig. 11a, b. For muscle stimulation, braided stainless-steel wires coated with polytetrafluoroethylene with a crimped stainless-steel tube[64,65] to increase contact surface area and reduce impedance (~ 500 Ω at 1 kHz Supplementary Fig. 11c, d) are used. Note that the core technology and fabrication processes of this device are based on modular designs that enable a high degree of flexibility of neural interfaces including commercially available solutions or custom electrodes. For example, electrode leads could, in principle, be used to stimulate intracortical electrodes or cuff electrodes on peripheral nerves. The scalable design of the monolithic device body enables free positioning of stimulation sites within the device footprint as shown in Supplementary Fig. 12a–c to target sites distributed across the rostrocaudal and mediolateral extent of the spinal cord. The onboard microcontroller enables complex stimulation protocols that might be used to improve functional outcomes; e.g., variable stimulus intervals used to reduce muscle fatigue[66] as illustrated in Fig. 3c.

An 8-bit DAC sets the resolution of the current source which is capable of delivering up to ±24 mA. Setting the stimulation current range to ±4.7 mA allows for the balance of fine current control (40 μA) (Fig. 3d) with a dynamic range that is capable of both muscle and spinal stimulation. It should be noted that while high currents can be delivered to all channels, this may result in injury or animal discomfort in spinal cord applications; the primary intended use for high current stimulation is for muscle stimulation. For higher resolution, a lower dynamic range can be chosen by changing feedback resistors on the device; e.g., with a dynamic range of ±1.8 mA the resolution is 7 μA (Supplementary Fig. 14a). Currents are directed dynamically to each of the eight channels with the multiplexer (40 μs switching latency), and stimulation parameters are controlled independently for each channel. Supplementary Fig. 13

and Supplementary Fig. 14b show an example in which each stimulation channel for both the muscle and spinal electrodes is wired to resistive loads (1 kΩ and 16.5 kΩ) that are similar to typical electrode impedance magnitudes, then tested in vitro with stimulation pulse durations between 100-1000 μs which are similar to typical durations used in FES applications[67–69]. In this example, the maximum current error is only ±12 μA (7 μA is due to the voltage error of the DAC and 5 μA due to the current driver) for a set current of 1 mA. This matches experimental data showing a maximum current error of ± 12 μA (Supplementary Fig. 15). This demonstrates the robustness of the analog front end. To deliver the programmed stimulation currents, especially in the case of high current applied through high impedance electrodes (Supplementary Fig. 14e), the device must be capable of operating with high voltage compliance. The device features a maximum voltage compliance of ±20 V as shown in Supplementary Fig. 14c, d, and validated with a 10 kΩ load in the treadmill cage enclosure with an RF power of 5 W, observing variations of ±4 V across the enclosure volume (Supplementary Fig. 14d).

An important aspect of chronic stimulation in clinical applications is the need for biphasic stimulation capabilities with the option to deliver anodic or cathodic leading pulses[70]. Biphasic pulses allow for charge balancing, designed to reduce charge accumulation and irreversible electrochemical processes at the electrode surface which can result in tissue damage or electrode degradation[71,72]. Charge balancing can also be achieved using a unidirectional current and a series capacitor[73]. However, this method requires a low output resistance and a correctly matched capacitor that depends on the electrode design. This is why in the devices described here, biphasic stimulation was used for active charge balancing which enables versatile use with custom electrodes. This process is evaluated ex vivo by recording direct current through the stimulation electrode during chronic stimulation in saline solution and integrating current over time. The residual charge during 1.8 mA biphasic stimulation pulses at a frequency of 87 Hz is a maximum of 4.6 nC after 57 stimulation cycles

(Supplementary Fig. 16a, b) which is within the limit of allowable residual charge before irreversible damage to either the tissue or electrode[74].

The device can be used with a variety of electrode materials, impedances, and designs; in our experiments, we use both spinal epidural and intramuscular electrodes. Although electrode performance is distinct from the functionality of the device, we evaluate the chronic performance of the electrodes used in our current design by performing accelerated rate testing. We test spinal electrodes both within the water window of the electrode material (gold (<±1.2 V)) and above the water window at 1 kHz for 14 days at room temperature (Supplementary Fig. 17a, c, e). This reflects more than 1 billion stimulation pulses; given a typical FES stimulation rate of 50 Hz. Not accounting for in vivo physiological factors, this results in several months of real experimental time. Muscle electrodes, show no change in impedance and no obvious change in electrode appearance for stimulation applied within or above the water window (Supplementary Fig. 17b, d, f). This observation is expected given the large surface area and consequent low current density and material bulk. The smaller spinal electrodes, on the other hand, increase impedance by less than 15kΩ when operated within the water window, enabling chronic operation in small animal models. When spinal electrodes are driven outside of the water window, electrode impedances increase much more substantially (above ±1.4 V) and dissolution of the gold electrode coating results in electrode failure after 7 days corresponding to 5 months of real experimental time. These results are generally expected for the electrode sizes and materials used here and are independent of the overall performance of the device. Devices described implanted in this study generally remain functional for 4-6 weeks. Importantly, because of our system's modular design, the stability of the device is independent of the electrode materials or designs utilized (e.g., platinum vs. gold). Thus, electrodes that have better chronic performance can be easily implemented without requiring changes to the fabrication techniques or compromising electronic device performance[16].

Stimulation using constant current mode (1 h sessions repeated 5 days a week for 16 weeks) using an accelerated lifetime test (9 million pulses at 1 mA at 1000 Hz at room temperature matching parameters used for spinal cord stimulation[75]) results in similar degradation (Fig. 3e) and compares to results from literature (Supplementary Fig. 18a–c)[76].

## Device mechanics and materials

The materials used to fabricate the device and the design of its overall mechanics were chosen to maximize the stable long-term operation and reduce the impact on surrounding tissues. The device consists of a thin, dual-layer flexible circuit board with a polyimide (PI) backing, combined with gold (Au) plated copper (Cu) traces populated with off-the-shelf ICs as shown in Fig. 4a. Devices are conformally coated with Parylene-C allowing for both biocompatibility with surrounding tissues and protection from corrosive biofluids. The encapsulation is thermally stressed in an accelerated rate test at 37 °C, 60 °C, and 90 °C showing an acceleration factor of 0.32 using Arrhenius scaling with an estimated device lifetime of 3 months at physiological temperatures (37 °C) (Supplementary Fig. 19a–d). This thin layer stack (100 μm) results in flexible device mechanics that enable the device to conform to the curvature of the body at the implantation site, minimizing impact on animal mobility or adverse interactions with surrounding soft tissue. To further improve mechanical biocompatibility, rigid islands consisting of ICs and a high density of circuit traces primarily made of hard materials such as silicon and copper (70 GPa – 180 GPa) are fully encapsulated with silicone (360 kPa – 870 kPa) to match the elastic modulus between the surface of the device and soft body tissues (0.1 kPa – 10 kPa and 1 kPa – 5 MPa respectively)[15,77–82]. Highly stretchable interconnects link the device body to the electrodes,

decoupling the movement of the implanted electrodes from the main device body. By incorporating strain-isolating mechanical designs using self-similar serpentine interconnects[83] that minimize tensile strain within the copper traces, device deformations are kept well below the plastic regime, thereby enabling indefinite strain cycles without damage.

Performance of these interconnects is evaluated using finite element simulation to ensure elastic deformation of all conductive elements within the stretchable interconnects under physiologically relevant levels of strain estimated from a μCT image of the spinal and return electrode displacement (Supplementary Fig. 20). Supplementary Fig. 21a (left) shows a simulated strain profile of the copper traces (visibility of the plastic layers is turned off for clarity) from the side and top view indicating a maximum strain of 0.77 % when displaced (x = 3 mm, z = 5 mm) which approximates the deformation the interconnect experiences when the spinal electrode is inserted between the lumbar vertebrae (see Supplementary Fig. 20). The simulation results are further validated with a benchtop experiment closely matching deformation patterns of the computational results (Fig. 4c). We also examine the strain caused by the placement of the return electrode, which is positioned laterally by elongating the interconnect by as much as 50% (Supplementary Fig. 21b). This is accomplished with self-similar serpentines that improve strain-isolating mechanics to minimize copper stresses[43]. Supplementary Fig. 21b (left) shows the simulated strain profile of the copper traces in top and side views. Maximum strain of 2% is observed when the interconnect is displaced (y = 20 mm). Computational results match bench top tests closely as shown in Fig. 4d (right). Engineered elastic-regime deformation of the copper traces within the interconnects[14] enables repeated strain cycles (>1.5 million, Supplementary Fig. 22) without loss of electrical conductivity in the trace, as demonstrated in cyclic strain experiments with monitoring of trace impedance (Fig. 4b). Safety margin for these interconnects is determined by the maximum strain before failure. For the spinal electrode interconnect, the maximum strain is 311% which corresponds to a displacement of 56 mm (Fig. 4c). For the return electrode, the maximum strain is 330% which corresponds to a displacement of 140 mm (Fig. 4d). In both cases, displacements and strains are well beyond those that might occur during device implantation of which none of the 21 implantations show signs of electrode damage immediately after surgery (Supplementary Table 2). Without the use of strain-isolating serpentine interconnects the maximum strain significantly decreases to a maximum strain of 10.7% which corresponds to a displacement of 3.1 mm (Supplementary Fig. 23), demonstrating the robust mechanical properties of this device.

## Compatibility of implanted devices

The procedures for implanting the device are similar to those described in previous work on spinal epidural stimulation in the rat[29]. The electrode is inserted into the epidural space through a partial laminectomy between L3 and L4 and positioned at the L1 and L2 vertebral levels to span the rostro-caudal extent of the lumbar spinal cord. It is then cemented to a vertebral screw placed in the L4 vertebra (Supplementary Fig. 24). The antenna and electronics are placed subcutaneously between back muscles and skin, centered over the spinal column and between the pelvis and ribs (Fig. 5a). Muscle electrodes are tunneled subcutaneously to the hindlimb and inserted intramuscularly in individual muscles (biceps femoris posterior in these examples) (Supplementary Fig. 24). The overall procedure takes ~2-3 hours. Postoperative X-rays are used to monitor the placement and integrity of the implanted device (Fig. 5a). Tungsten markers within the spinal electrode are used to evaluate electrode placement and to estimate contact locations in the spinal column (Fig. 5a). Figure 5b illustrates the mechanical flexibility of the implanted device, with the device readily conforming to both extended and highly curved body positions. The overall configuration of the implanted device and individual

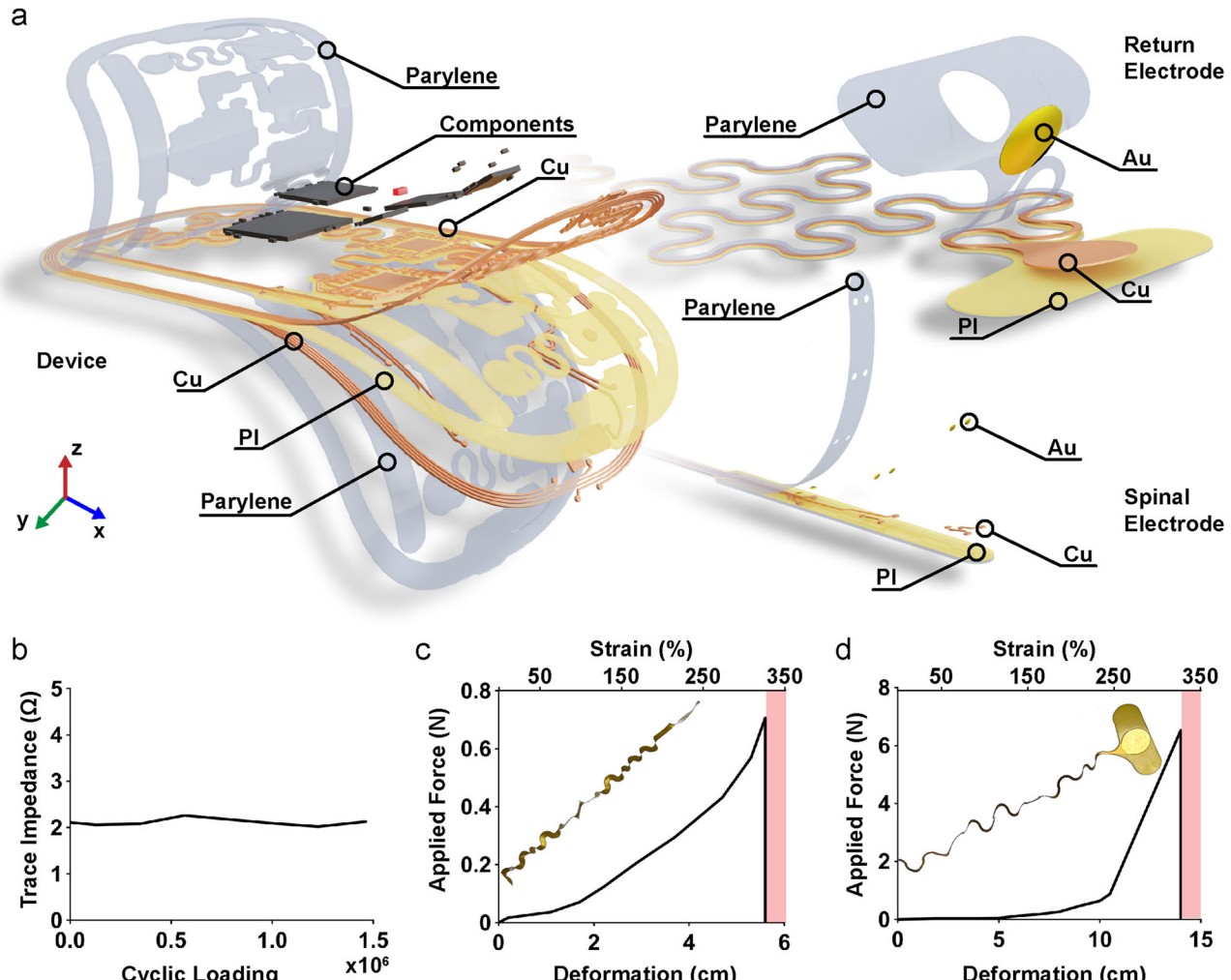

**Fig. 4 | Material overview and mechanical characterization of the device.**
**a** Exploded view rendering of each material layer used in the device, the return electrode, and the spinal electrode. **b** Change in trace impedance of the serpentine interconnect during cyclic loading measured at 1 kHz. **c** Strain stress curve of the serpentine structure for the spinal electrode, and **d** the return electrode.

components are visualized in vivo with μCT imaging, illustrating the location of the components when fully implanted (dorsal view in Fig. 5c and cross-sectional view in Supplementary Fig. 25).

The subdermal implantation of the device enables animals to recover quickly after surgery (Supplementary Fig. 26) so that chronically implanted animals are visually indistinguishable from naïve animals (Fig. 5d). Spontaneous exploratory activity levels decrease in the first week after implantation but return to normal levels by the second week (Fig. 5e). Hindlimb kinematics, evaluated as ranges of joint motion, during treadmill locomotion are similar in animals measured before and after implantation (Fig. 5f). In-cage trajectory tracking for 30 minutes indicates less exploration in the first week after surgery compared to before surgery (see Supplementary Fig. 27), but exploration returns to control levels in week 2. These observations highlight the advantages of this fully implanted system, showing that after an initial recovery period of ~1 week, the implanted device has minimal impact on animal behavior and locomotion.

**Stimulation control and animal kinematics**
Evaluation of the performance of chronically implanted devices in anesthetized animals is conducted using a circular transmitter antenna (Fig. 6a) positioned above the rat for power supply and communication (RF power 3-5 W). Movements evoked by spinal and muscle stimulation are monitored by kinematic tracking of the limb with

DeepLabCut (DLC), with the magnitude of the evoked movement quantified as the maximal displacement of a point on the toe (Fig. 6b, c). Figure 6d illustrates an example of the effects of gradually increasing current amplitude applied to contact on the spinal electrode (channel 1, 4 Hz, 500 ms duration), showing that the device can be used to produce graded activation of neural substrates with responses saturating at high levels of stimulation. A similar example is illustrated in Fig. 6e for graded responses evoked from intramuscular stimulation (c5, 4 Hz, 500 ms duration). As summarized in Fig. 6f, the amplitude of the evoked response from both spinal and muscle sites varies systematically with applied current, resulting in standard recruitment curves that can be used to control hindlimb movements. Importantly, in this experiment the device is able to control current ranges appropriate for both spinal (10 μA to 1 mA, typically ~200 μA) and muscle (1 mA to 5 mA, typically ~2 mA) sites, providing graded and well-controlled recruitment of each. Similar recruitment curves are obtained for each individual spinal and muscle contact site on the implanted device (Supplementary Fig. 28). The performance of the device is monitored through IR communication throughout all stimulation trials, and in all cases illustrated here the harvested power provides sufficient margins; i.e. the harvested voltage was always greater than the voltage required to source the required currents (Supplementary Fig. 29). Note that the current ranges used here do not fully saturate this muscle, likely reflecting the large size of biceps

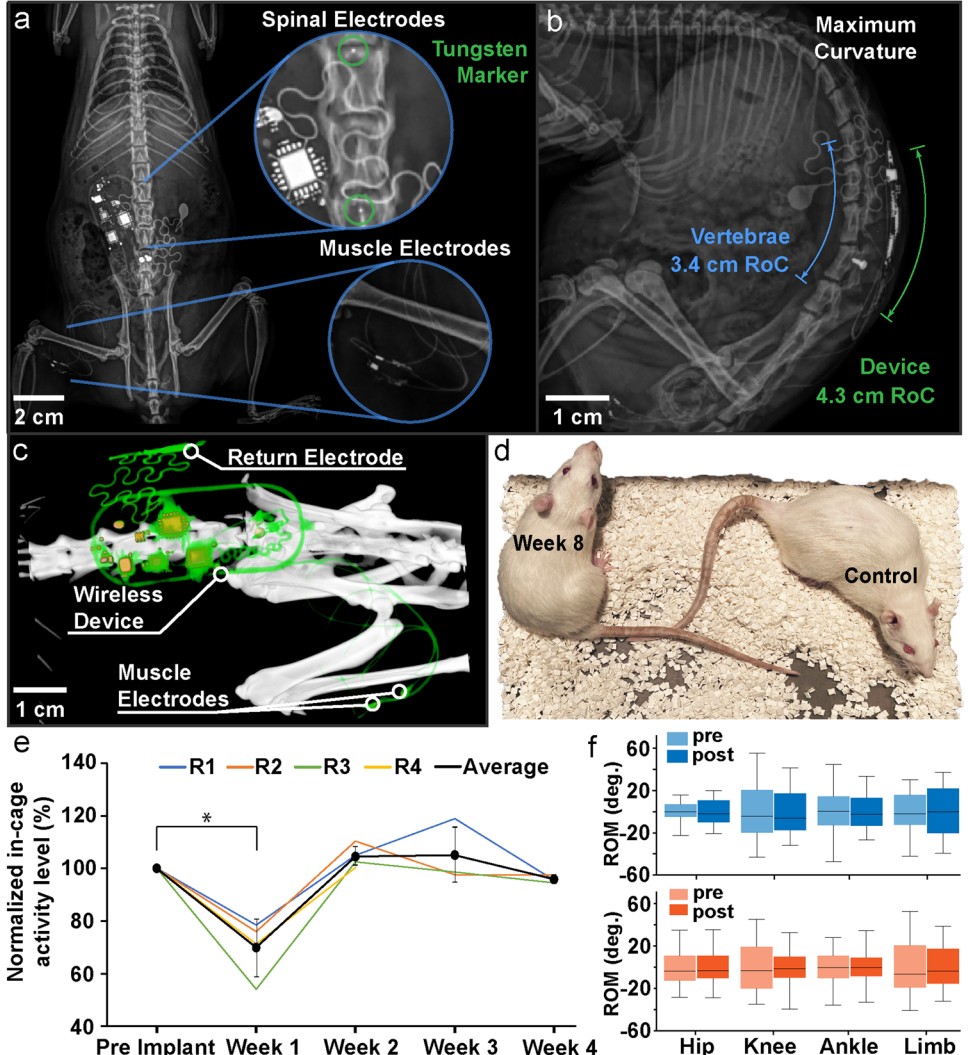

**Fig. 5 | Spinal and muscle implantation procedures, in-vivo radiological imaging, and behavioral effects of implantation. a** X-ray image showing both spinal electrodes with tungsten markers and muscle electrodes in the biceps femoris. **b** Curvature of the device conforming to the spine in hunched postures. **c** μCT reconstruction of the implanted device (dorsal view). **d** A comparison between implanted (eight weeks) and naive rats. **e** Four weeks of 30 min in-cage travelling distances after implantation compared to baseline levels of 4 subjects (R1-R4, $n = 4$ independent animals). Data are presented as mean values +/- SEM. Repeated measure ANOVA is used, and post hoc analyses are adjusted for multiple comparisons (Bonferroni correction ($\alpha_{corrected} = \alpha/n$), two-sided). * in the panel represents $p = 0.00402$. **f** Range of motion of the hind limb from rats ($n = 2$ independent animals) while running on the treadmill before and after implant, showing the minimal impact of the device on the animal's mobility. The box plots are plotted with the median as the center, first and third quartiles as the lower and upper bounds of the inter-quartile range (IQR), and the whiskers extend to the 1.5× IQR.

femoris. Figure 6g illustrates the temporal control afforded by the device, with stimulation frequency applied to a muscle electrode increased stepwise from 4 to 100 Hz (at a constant current of 4.5 mA and a train duration of 1000 ms; c2) to evoke responses that increase from unfused single muscle twitches to fully fused, tetanic contractions.

The location of the implanted electrode is determined using the tungsten fiducial marks visualized in x-rays and mapped to evoked responses of corresponding spinal sites (Fig. 6h), with limb flexion responses evoked from rostral sites and limb extension responses evoked from caudal sites. These results demonstrate the capabilities of our device, showing that it is able to precisely control the timing and amplitude of stimulation applied to both spinal and muscle sites in order to evoke a range of graded, functional movements.

### Chronic device performance
Full implantation and use of flexible materials avoids many of the complications typically observed in chronic applications due to poor

matching between device and tissue mechanics and the use of externalized connections that must be repeatedly connected and disconnected. Stiff electrical components and external attachments not only lead to device failure and impede animal behavior, they can also result in device motion and instability. Implanted devices typically remain functional 4-6 weeks following implantation. The subdermal device body itself drifts only minimally after one month (see Supplementary Fig. 30). Figure 6i shows recruitment curves, for spinal (upper) and muscle (lower) electrodes in a device implanted in a subject for 6 weeks. Recruitment curves are similar, though not identical, across the timespan (data for all stimulation sites of this device are illustrated in Supplementary Fig. 31). Supplementary Fig. 32 shows the changes in the current threshold needed to evoke a movement at each site on one implanted device; on average, threshold currents change by 0.067 mA over the 4 weeks.

There is no consistent trend in the changes of recruitment curves or threshold currents over weeks for these devices, suggesting that any changes are not due to device failure, tissue damage, or electrode

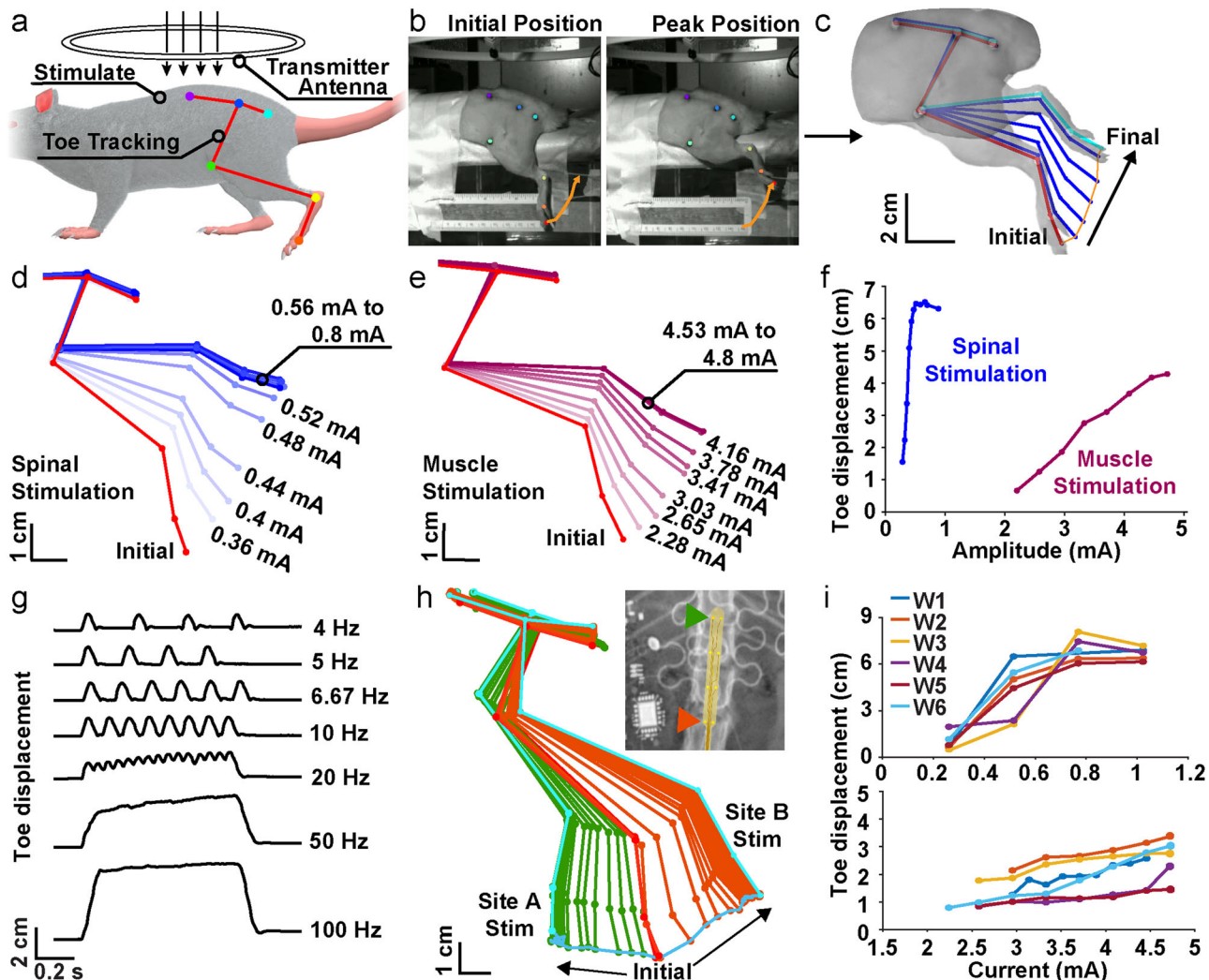

**Fig. 6 | Stimulation programmatic control and evoked kinematics. a** Schematic illustrating wireless stimulation using a circular antenna setup to power devices in anesthetized animals. **b** Skin marker positions during rest and at the peak displaced position caused by stimulation. The orange line indicates the trajectory of the toe marker. **c** Reconstructed hindlimb kinematics from DeepLabCut labelling. **d** Amplitude modulation from a stimulating spinal channel (channel 1, 0.36 mA to 0.8 mA). **e** Amplitude modulation from a stimulating muscle channel (channel 5, 2.28 mA to 4.8 mA). **f** Recruitment curves quantified by toe displacements from gradual increases in stimulation intensities of (**d**) and (**e**). **g** Toe movements evoked by modulating stimulation frequency applied to a muscle channel (4 Hz to 100 Hz). **h** Hindlimb schematics showing responses (flexion and extension) corresponding to stimulation of c1 and c4 electrode sites indicated in X-ray. In-vivo X-ray indicating the position of an implanted electrode panel. The overlying trace of the electrode panel was scaled and aligned to the fiducial marks observed in the X-ray. **i** Recruitment curves were generated from the same spinal channel (channel 0) and muscle channel (channel 2) for six weeks.

encapsulation but more likely reflect either variability in animals' state between weeks (e.g., level of anesthesia) or changes in electrode placement. The position of the implanted spinal electrode is tracked with embedded x-ray markers and does not drift substantially over one month after implantation (Supplementary Fig. 30), with an average absolute rostral/caudal displacement of 0.5 ± 0.6 mm and rotation of 0.8 ± 0.5 degrees. Because of the high sensitivity to the position of the spinal electrodes, even these small movements could account for some of the changes in recruitment curves and threshold currents. Together, these observations demonstrate that devices remain functional and capable of producing functional movements and modulating neural function over several weeks after implantation. A detailed record of all devices implanted in the context of this work with documented device failure modes is presented in Supplementary Table 2.

The flexibility of this device and its lack of external connections allows it to be used in unanesthetized, awake-behaving animals, enabling chronic experiments for interventions such as rehabilitation after spinal cord injury. For the example shown in Fig. 7, the stimulation device is implanted and animals are given a complete, midthoracic spinal cord transection (Fig. 7a). After recovery from these procedures, animals are then placed on a motorized treadmill and locomote using only their forelimbs, while stimulation is applied through the implanted device to activate their hindlimbs (Fig. 1c). Stimulation of spinal sites is able to produce functional motion, evoking limb protraction capable of lifting the animal's body (stimulation site c3, 1.2 mA at 50 Hz, duration 1000 ms); (Fig. 7c) and limb retraction (stimulation site c0, 1.2 mA at 50 Hz, duration 1000 ms); (Fig. 7d). Similar to the performance in anesthetized animals, varying stimulation strength of spinal (Fig. 7b left) or muscle sites (Fig. 7b right) in these awake behaving animals evoked movements of graded amplitude (measured while hindlimbs are lifted off the ground; Supplementary Fig. 33). The greater variability evoked from spinal sites (Fig. 7b vs. Fig. 6f, Supplementary Fig. 28) likely reflects variations in spinal excitability present in unanesthetized animals (e.g., due to variations in sensory inputs).

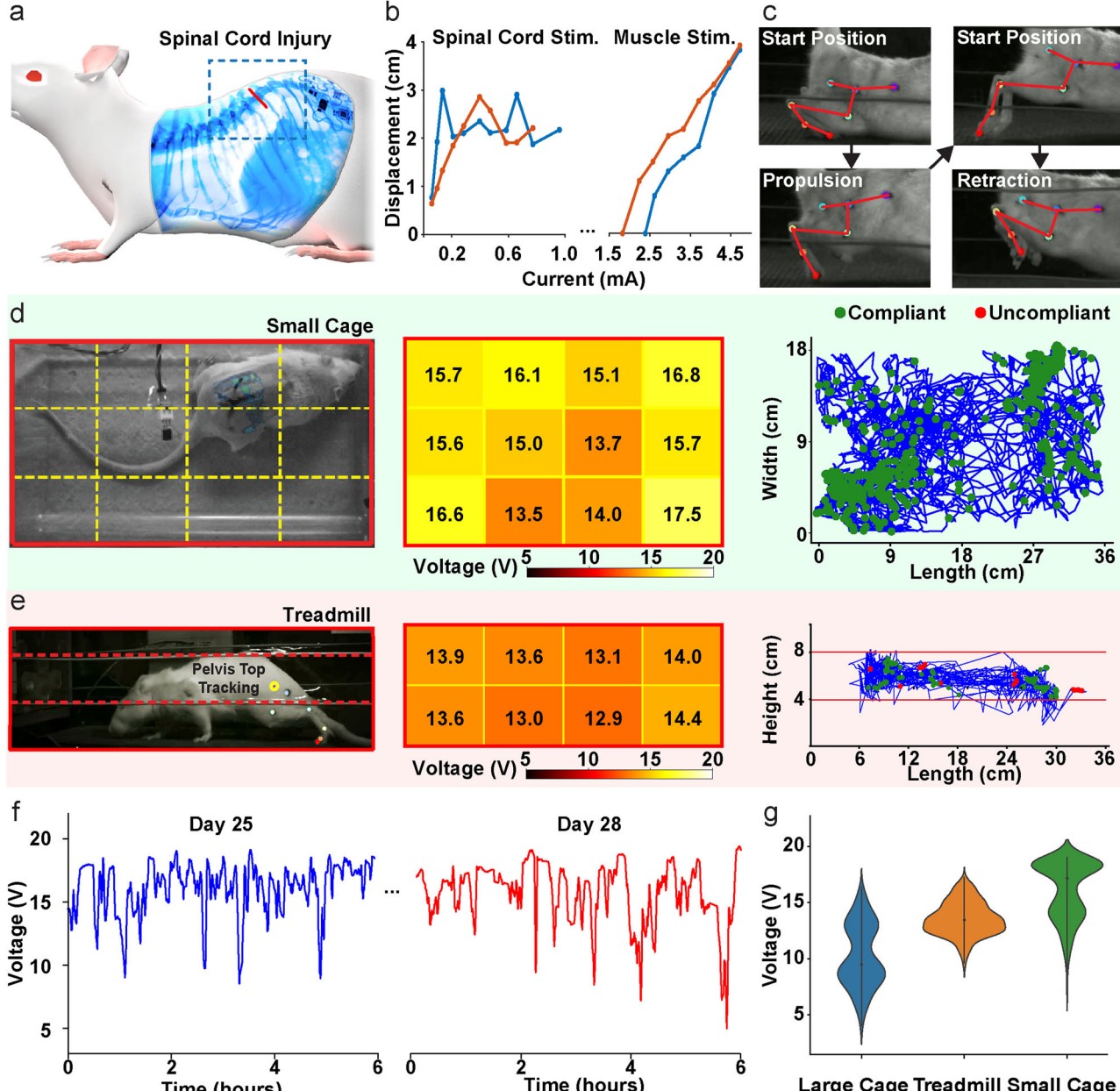

**Fig. 7 | Stimulation-evoked movements in spinal cord transected animals. a** An awake, spinal cord transected (T10) animal is stimulated. **b** A limb propulsion movement is evoked from stimulation of a spinal electrode site (c3) while the animal is locomoting on a treadmill with unparalyzed forelimbs. **c** Stimulation of spinal electrode site (c0) evoked limb retraction. **d** Stimulation in cage with freely moving spinal cord injured animals. Middle plot illustrates the average voltage compliance at each region of the cage. Right plot illustrates movement trajectories

(blue) measured over 2 hours. Green marks indicate points at which stimulation is compliant; red marks indicate uncompliant stimulation. **e** Freely behaving animal on a moving treadmill (conventions follow (d)). **f** Voltage compliances on individual trials over two 6-hour sessions and sessions recorded 25 and 28 days following device implantation. **g** Distributions of voltage compliances measured in three enclosures of increasing sizes: large cage (29 cm × 36 cm), treadmill (16 cm × 56 cm), small cage (18 cm × 36 cm).

To evaluate system reliability with freely behaving subjects we perform stimulation during free behavior in cages and on a treadmill (Fig. 7d, e). Subjects are tracked with markers positioned on the pelvis and back. Compliance voltage, current, and pulse timing are communicated by the implant and synchronized with the position of the subject.

Figure 7d illustrates results from a 2 h session in a cage using 282 μA stimulation current and a pulse width of 114 μs every 10 s. In this case 694/694 stimulation event trials are compliant (i.e., the desired current is delivered). The system voltage compliance is 15.3 ± 2.5 V averaged across all trials. Figure 7e illustrates the evaluation of system performance during locomotion on a moving treadmill with the same

stimulation parameters. The enclosure area is larger (16 cm × 56 cm vs 18 cm × 36 cm) and results in a lowered average system voltage of 12.70 ± 1.0 V and a larger number of trials are uncompliant (52/219 trials). It should be noted however that stimulation even in the case of a non-compliant event is still delivered, although with a slightly reduced current.

The devices enable long-lasting, uninterrupted stimulation in freely behaving animals. System voltages measured in two 6 h sessions with continuous stimulation (282 μA, 114 μs pulses, repeated every 10 s) are illustrated in Fig. 7f for a small cage enclosure. These sessions are recorded 25 and 28 days after the implantation of the device. The average system voltage for day 25 is 16.2 ± 2.4 V and 15.8 ± 2.8 V for day

28 with only intermittent drops in system voltage. System performance in a variety of cages during free behavior is displayed in Fig. 7g (detailed plots shown in Supplementary Fig. 34). As expected, voltage compliance is reduced in larger enclosures (29 cm × 36 cm); however, even in the largest enclosures sufficient power is harvested to enable stimulation at high voltages (9.9 ± 2.8 V).

## Discussion

In summary, we demonstrate a fully subdermally implantable device platform that has broad utility in therapy and assistive function for electrical stimulation of the nervous system. Devices are enabled by resonant power transfer with on-device, co-located passive resonators resulting in high-efficiency power transfer that surpasses other existing approaches by ~500% in power harvesting capabilities and enables a dual voltage supply with ±20 V in large experimental arenas. By taking advantage of the continuously available power, device designs with high channel counts, current-controlled biphasic stimulation, and complex waveform delivery are enabled. Devices are digitally programmed and deliver telemetry for detailed monitoring of electrode performance during chronic stimulation. Small footprint, stretchable, and battery-free system designs enable full implantation with stable electrode location over months of operation with no measurable impact on behavior and animal health, thereby providing a substantial advance over device architectures that break the skin for externalized hardware.

We illustrate the utility of this device by demonstrating its ability to produce movements through stimulation of both muscle and spinal sites. These sites require high voltage compliance and a range of currents, with spinal sites requiring hundreds of μA and muscle sites requiring mA. Device efficacy is demonstrated in both healthy animals and in animals with chronic spinal cord injuries with consistent efficacy and stability over 6 weeks of operation. These capabilities are illustrated by the production of consistent amplitude movements from both spinal and muscle sites using similar ranges of currents over several weeks, demonstrating the consistent ability of devices to generate currents and voltages necessary for neuromodulation or functional restoration. Chronic stability of devices is demonstrated by stable delivery of stimuli in freely behaving subjects in environments relevant to FES research and enabling operational durations that are not achievable with battery-powered systems. It should be emphasized that additional experiments across a broader spectrum of scenarios beyond the three we have presented here are required to comprehensively determine the full potential of the chronic performance of the device. Nonetheless, the characteristics of the devices demonstrated here make them suitable candidates for automated investigation of rehabilitation of neural circuits on clinically relevant timescales and with adequate statistical power to investigate underlying mechanisms that currently prevent widespread clinical adoption. This platform is designed using established techniques for flexible printed circuit board manufacturing, SMD population, and laser de-paneling as shown in Supplementary Fig. 35. The combination of the device capabilities and manufacturing process has been optimized to be used as an investigational tool by the neuroscience community to study neural circuit rehabilitation in freely moving subjects at a cost that enables single device use for chronic rodent studies. When translating this technology for use in large animals, magnetic resonant coupling starts to become inefficient due to the operational distance between the receiver and transmitting coil exceeding 1 m. To solve this issue, implanted devices can be combined with wearable systems to enable directed power delivery and communication, which is possible with power consumptions suitable for battery operation as demonstrated in Supplementary Fig. 36 where a small NFC reader chip with 150 mW RF power yields 76 mW of harvested power at the implant. This approach might enable the devices described here to be translated into human experiments, although further development will be required to enable such designs.

## Methods

### Ethical approval
This research compiles all relevant ethical regulations. The study protocol was approved by the Institutional Animal Care and Use Committee of Northwestern University (protocol number: IS00011700).

### Flexible Circuit Fabrication
The flexible circuit board was designed in AutoCAD 2021 and fabricated in panels by an external manufacturer (PCBWay). The devices in the panels were defined using a UV (355-nm) laser ablation system (LPKF; Protolaser U4). Devices were washed with isopropyl alcohol and deionized (DI) water and electroplated at 6 V with nickel for 1 minute (Gold Plating Services, Surface Activator Solution). Devices were rinsed with DI water and electrodes were brush plated with gold (Gold Plating Services, 24k Brush gold solution) at 2 V for 1 minute per electrode pad and rinsed with more DI water. A custom programming board utilizing Arduino as ISP was used to program the microcontroller (Atmel; Attiny 84 A) before it was mounted on the flexible circuit. Devices were populated with electrical components (Supplementary Figs. 1 and 2b) and reflowed with a hot air gun at 350 °C using low-temperature solder (Chip Quik; TS391LT). They were then tested contactless with a reflection bridge (Siglent; SSA 3032X; RB3X20) to adjust the tuning capacitors to tune implant device antenna resonance to 13.56 MHz. Stainless steel wires for muscle electrodes were soldered onto the device using the low-temperature solder (Chip Quik; TS391LT) and stainless-steel flux (Superior Flux and Manufacturing Company; Superior #71). A small stainless-steel tube (27 G, 1 mm long) was crimped to the exposed wire at its distal end. A tungsten foil (Alfa Aesar, CAS# 7440-33-7) was laser ablated (LPKF; Protolaser U4) to create 200 μm diameter radiopaque markers which were glued to the spinal electrode using cyanoacrylate glue (Super Glue Corporation, 15187). Test pads within the device structure were used to test electrical functionality before encapsulation.

### Device encapsulation fabrication
The devices were sonicated with isopropyl alcohol for 10 min and air-dried. Thermal Epoxy (Henkel Loctite, 3621) was used as a potting compound around all electrical components and edges of ICs. Devices were cured on a glass slide in the oven at 120 °C for 5 min and degassed for 10 min. Electrode pads were covered with Kapton tape. The devices were suspended on a wire and coated with Parylene-C in a Parylene P6 coating system (Diener electronic GmbH, Germany) using 5.0 g of Parylene-C dimer. Devices were flipped and coated a second time for a total Parylene thickness of 18 μm. Kapton tape was removed from the electrodes and devices were tested to ensure reliable 2-way device communication (Vishay Semiconductors; TSDP34156) and stimulation response (Siglent; SDS 1202X-E). Devices were submerged in a saline bath over 24 hours and tested again to ensure successful Parylene encapsulation. The devices were finally dip-coated in platinum cure silicone (Smooth-on, Ecoflex 00-30) cured at 50 °C for at least 2 hours, and tested once more before implantation to ensure fully functional operation across all eight channels.

### Antenna characterization
Devices with differing receiver antenna designs (4-turn, 8-turn, and co-planar 4-turn with a resonator) were fabricated and connected with SMA connectors. The transmitter antenna for all devices was created with a two-turn winding at 4 cm and 8 cm heights around a treadmill cage (56 cm by 16 cm) and connected to an antenna tuner board terminated with an SMA connector. Both the transmitter and receiver

antennas were tuned to maximize their q factor at 13.56 MHz. Each device was characterized with S11 and S21 magnitude measurements between 10 MHz and 20 MHz using a vector network analyzer (OMI-CRON Lab, Bode 100).

A high-frequency structure simulator, Ansys Electronics Desktop (AED) 2021 R1, was used to simulate the electromagnetic behavior of the receiver antenna designs with 4-turns, 8-turns, and a co-planar resonator. 3D models of the transmitter, receiver, and resonator antennas were designed and simulated using AED 2021 R1. To adjust the resonance to the desired 13.56 MHz frequency, a matching network was optimized using the software ANSYS-Simplorer 11.0.

### Power harvesting and consumption characterization

Power harvesting characteristics for the 4-turn, 8-turn, and co-planar 4-turn with resonator receiver antennae were determined by measuring voltage after the rectification circuit (Aneng; AN8008) across shunt resistors with the devices placed at both 4 and 8 cm height in the center of the treadmill cage (56 cm×16 cm) using a dual loop transmitter antenna with between 2 W and 6 W RF power. For the voltage supply stability tests of the digital circuit, a shunt resistor of 2 kΩ was used to match the average system load during operation (Aneng; AN8008). Device power was measured in the 3D cage volume at heights of 4 cm and 8 cm. Power harvested during angular misalignments between −90° and 90° relative to the cage floor between the transmitter antenna and receiver antennae were measured. Power harvesting was tested in the center of a 56 cm×16 cm x 16 cm cage with 2 W and 4 W of RF power while bending the device over a fixed radius of curvatures controlled by 3D printed bending fixtures between 2 cm and the device lying flat in 1 cm increments. Transient current consumption was measured using a modified current meter (Low-PowerLab; CurrentRanger) and acquired on an oscilloscope (Siglent; SDS 1202X-E) while powered with 10 V, 15 V, and 20 V supplies. Data transmission measurements occurred concurrently with power consumption characterization and were measured within the cage at 2 W of RF power at heights of 4 cm and 8 cm for 5 minutes using an IR receiver to count skipped transmission events while a direct measurement of voltage was taken on the digital supply (Aneng; AN8008).

### Thermal simulation and testing

Devices were powered for 5 minutes and the temperature on the top of each IC (held with Kapton tape) was measured directly with a thermocouple (Type K Thermocouple; Omega Engineering) at room temperature (22 °C). Temperatures of active components operating in RF cage powers between 2 W and 6 W were subsequently captured. The power consumption of each IC (MUX: 0.8 mW, MCU: 2 mW, Zener: 8.0 mW, DAC: 0.3 mW, Step-Down: 3.6 mW, Current Mirror: 19.4 mW) was estimated from their data sheets and used in a simulation model using ANSYS 2021 R1 to test device heating in air to validate the benchtop test. The thermal conductivity, heat capacity, and density of different materials were as follows: component potting (0.5 W m$^{-1}$ K$^{-1}$, 1000 J kg$^{-1}$ K$^{-1}$, and 1350 kg m$^{-3}$), polyimide (0.2 W m$^{-1}$ K$^{-1}$, 1100 J kg$^{-1}$ K$^{-1}$, and 1470 kg m$^{-1}$), copper (400 W m$^{-1}$ K$^{-1}$, 385 J kg$^{-1}$ K$^{-1}$, and 8900 kg m$^{-3}$), inner dies (130 W m$^{-1}$ K$^{-1}$, 678 J kg$^{-1}$ K$^{-1}$, and 2320 kg m$^{-3}$), and saline (0.6 W m$^{-1}$ K$^{-1}$, 4180 J kg$^{-1}$ K$^{-1}$, and 1000 kg m$^{-3}$).

### Encapsulation testing

Encapsulated devices were submerged in 0.01 M phosphate buffer, 0.0027 M potassium chloride and 0.137 M sodium chloride (Sigma, P4417) in sealed glass vials at 37 °C, 60 °C, and 90 °C and monitored for 2-way device communication (Vishay Semiconductors; TSDP34156) and stimulation response (Siglent; SDS 1202X-E) (LowPowerLab; CurrentRanger) weekly. The Arrhenius equation was used to determine the acceleration rate using the time of failure with devices subjected to thermal stresses of 60 °C and 90 °C. Devices were mounted on a

custom testing rig (Supplementary Fig. 22) that allowed for the translation of the spinal electrode at offset planes (vertical offset of 5 mm) while cycling the deformation of the electrode serpentine connection by 3 mm to match deformations in the μCT image (Supplementary Fig. 20).

### Electrode Testing

A range of 1 kΩ to 10 kΩ resistive loads were attached to electrodes and the voltage across them was measured (Siglent; SDS 1202X-E) to determine stimulation current for the 4.7 mA and 1.8 mA dynamic range devices respectively (dynamic range adjusted in circuit hardware). Increasing resistive loads were applied to each stimulation channel and the voltage was measured (Siglent; SDS 1202X-E) across the resistor for the muscle electrode with a 4.7 mA stimulation level and for the spinal electrode with 1 mA. Electrodes were placed in a 1x phosphate-buffered saline (PBS) solution (003002, Gibco, Life Technologies) at room temperature and the stimulation current was measured with a current to voltage converter (LowPowerLab; CurrentRanger) and acquired on an oscilloscope (Siglent; SDS 1202X-E) and integrated over time to calculate net charge. Devices stimulated continuously at 1 mA with a pulse width of 200 μs and period of 1 ms were submerged in 0.01 M phosphate buffer, 0.0027 M potassium chloride, and 0.137 M sodium chloride (Sigma, P4417) and monitored every week to check the electrode surface and impedance. Impedance measurements were collected using a function generator with a sine wave output of 100 mV pk-pk sweeping from 1 Hz to 11 kHz over 130 ms. A current meter measuring high-side current of the electrodes (LowPowerLab; CurrentRanger) was acquired on an oscilloscope (Siglent; SDS 1202X-E) incorporating a passive 10 MHz low pass filter. The envelope of the current and voltage reading are then used to calculate the impedance of the electrode between 10 Hz to 10k Hz. To evaluate the chronic stability of spinal and muscle electrodes, a series of tests was conducted with experimental parameters spanning the water window of the gold electrode (± 0.2 V, ± 0.6 V, ± 0.8 V, ± 1.0 V, and ±1.2 V) which were generated by a function generator (Siglent, SDG 1032X), while monitoring changes in impedance with a digital storage oscilloscope (Siglent, SDS 1202X-E) over a 100Ω shunt resistor at a 1 kHz biphasic stimulation with 100% duty cycle (50% positive and 50% negative pulse), performed at room temperature in a 0.1 M phosphate buffer saline (003002, Gibco, Life Technologies) solution. Additionally, the spinal electrode was subjected to testing at ± 1.4 V, a voltage exceeding the water window of gold to probe electrode lifetime outside of safe stimulation limits.

### Mechanical testing and simulation

To measure failure load of the stretchable interconnects, the device body was mounted on a scale (Mettler Toledo; AB104-S) and the electrode was fixed to a custom 3D printed guide rail for linear displacement. Displacement was measured with a digital caliper while the resulting force was measured from the scale. Electrodes were stretched until there was a loss of electrical conductivity within any copper traces. The linear interconnect was mounted on a custom stretching stage with a guide rail connected to a stepper motor (Creality, 17HS16-2004S1) to measure linear displacement. Displacement was measured with a digital caliper while the linear interconnect was stretched until mechanical failure. For fatigue testing a custom 3D printed jig was fitted with a servo motor to linearly displace the spinal electrode serpentine by 4 cm corresponding to elastic deformation acquired from finite element simulations. The serpentine was stretched over 1.5 million cycles and trace impedance was monitored daily. Ansys 2019 R2 Static Structural was used to simulate the direct elastic strain in the copper traces of the serpentine electrode interconnects caused by imposed displacements. The model was simulated using nonlinear mechanical elements with an element size manually input as 1$^{-2}$ mm. The Young's modulus (E) and Poisson's ratio (ν) are E$_{Polyimide}$ = 4 GPa, ν = 0.34; E$_{Copper}$ = 121 GPa, ν$_{Copper}$ = 0.34. Motion was simulated by

fixing the base of the serpentine interconnects with a fixed support and the applied deformation was added to the tip of the electrodes with a pivot joint. Device deformation while implanted in the animal was analyzed using ImageJ 1.53t to calculate deformation in the device in X-ray images of the rat post-implantation.

## Surgeries and implantation

All surgical instruments were autoclaved prior to surgery. Because high temperatures could degrade the Parylene coating of the implanted device, it was sterilized using ethylene oxide (ETO) and vented for 48 h before surgery. Female Sprague-Dawley rats (250-350 g) were anesthetized with 3% isoflurane with oxygen and placed on a water heating pad. The toe pinch reflex was used to check the depth of the anesthesia. Meloxicam (1-2 mg/kg of body weight, Covetrus® 5 mg/ml) was subcutaneously administered pre-surgery. Ophthalmic ointment (Puralube® Vet ointment) was applied to the eyes to prevent corneal drying. After establishing the sterile field, a 2–3 cm midline incision was made in the skin over lumbar L3-L4 vertebrae. Muscles overlying the vertebrae were separated from the lateral sides of spinous processes and muscle and connective tissue were removed from the transverse processes. Screws were placed in the L4 spinal vertebra. We then performed a partial laminectomy to open the intervertebral space between L3 and L4 vertebrae. The device was carefully handled with rubber-coated instruments to prevent accidental damage to the encapsulation coating. We inserted the spinal electrode completely into the subvertebral space so that it was located epidurally. After the insertion, several droplets of mixed dental cement (Ortho-Jet™ Package) were applied to bond the spinal electrode to the anchoring screws. A 1 cm incision was made on the hind limb over the biceps femoris. The stainless-steel wires used as muscle electrodes were tunneled from the device through from the dorsum to the hindlimb and implanted in the posterior head of the biceps femoris. Incisions were sutured (Sharpoint A663N Nylon Suture with FS-1 RC Needle) closed. Buprenorphine SR (1.2 mg/kg) was given subcutaneously 45 min before the end of surgery. Meloxicam (1-2 mg/kg of body weight, Covetrus 5 mg/ml) injection was administered daily for three days to relieve post-surgical pain, and the condition of the rat was closely monitored.

## Spinal cord transection

Implanted animals received a spinal cord transection after the device had been tested to make sure it functioned normally and was capable of effective stimulation. The animal was anesthetized with 3% isoflurane and the skin overlying the thoracic vertebrae was shaved and disinfected. An incision was made to expose thoracic vertebrae, and a laminectomy was performed at T10 to expose the T12-13 spinal segments. An iris scissor was used to cut the spinal cord, and the completeness of the transection was verified by observing the gap between the rostral and caudal cut ends of the spinal cord. The incision was closed with sutures, and analgesics (Meloxicam, Buprenorphine-SR) were given for three days after surgery. The bladder was manually expressed twice daily.

## X-ray and μCT procedures

The implanted animal was anesthetized with 3% isoflurane-oxygen before being transferred to the radiation platform. It was placed under the X-ray tube, and the aperture was adjusted to optimize the spatial resolution of the images. The X-ray beam was set up as 60 kVp, 40 mA, and 125 ms exposure time to balance the radiation penetration and animal safety. In addition to imagining the position of each implant across animals, in one animal three radiographs were taken while the animal was in naturally prone, laterally hunched, and laterally extended positions. To investigate the physical stability of the spinal electrode, we measured the angle between the lines formed by the electrode panel and by the vertebral column and measured the distance between the caudal marker on the electrode panel and the implanted screw immediately after surgery and one month later (Fig. 7d).

## μCT imaging

For μCT imaging, subjects were anesthetized with isoflurane and placed on the heated μCT bed. Images were acquired with a preclinical μPET/μCT imaging system, NanoScan scanner (Mediso-USA, Arlington, VA). Data was acquired with "medium" magnification, <60 μm focal spot, 1 × 4 binning, with 720 projection views over a full circle, and a 300 ms exposure time and 70 kVp. The projection data was reconstructed with a voxel size of 68 μm using filtered (Butterworth filter) back projection with Nucline v2.01 (Mediso USA, LLC., Arlington, VA). The reconstructed data was visualized and segmented in Amira 2020.2 (FEI, Houston, TX).

## Behavioral analyses and video recording

We recorded animals' spontaneous exploratory behavior by tracking their movements over 30 minutes after being placed in a novel housing cage. A marker placed on the back of the animal was tracked by an overhead camera (2 Hz) using DeepLabCut 2.2.0[84]. K-means clustering was used to extract 20 frames from each video and those frames were hand-labelled for training DeepLabCut models. The training stopped when the model's loss was less than 0.01. The trained model was used to label the rest of the videos. The automatically labelled frames were examined manually. If the confidence level was low, more frames were hand-labelled, and the model was re-trained. Any frames that had a tracking confidence level below 70%, were excluded (< 5% of frames across all animals). Animals' travel distances were measured the week before and for three weeks after the implantation. A repeated-measure ANOVA with significance level α = 0.05 was used to compare the changes of the activity levels over the recovery period.

To investigate animal locomotion changes after implantation, we trained the rats three times a week for two weeks on treadmill locomotion before the implantation surgery. Before each treadmill recording, the hindlimbs were shaved and pelvis top, hip joint, pelvis bottom, knee joint, ankle joint, metatarsal joint and toe tip were marked. The speed was 10 cm/s and limb motion was captured at 200fps with two horizontally separated cameras. Marker positions were then tracked with DeepLabCut 2.2.0, as described above. We computed hip, knee, ankle angle, limb angle (defined as the angle between the pelvis top, hip, and MTP markers), and vertical toe displacement.

## Device test protocol

We tested the animal responses to stimulation under anesthesia (3% isoflurane- oxygen mix) while placed on a water heating pad. The body was elevated and secured on a platform so that the hindlimb was naturally hanging in the air. The hindlimbs were shaved and the same landmarks described in the previous section were labelled. We used the NeuroLux system for the programmatic control of the stimulation with custom control over the RF field via the TTL port of the device. The transmitter antenna and IR detector were placed over the implanted device (Fig. 6a). Real-time feedback about device performance was provided by the IR communication. If the delivered currents were not within voltage compliance, the RF power was increased as necessary. We then collected recruitment curves for each implanted electrode by varying current amplitudes. We first determined the threshold current, taken as the first current level capable of eliciting a visible hindlimb movement. The current intensity was then gradually increased to measure the recruitment curves for each channel. Evoked hindlimb movements were monitored using video recordings and processed with DeepLabCut 2.2.0, as described above. Devices were tested weekly to document the chronic performance of the device.

## Device characterization in freely behaving subjects

We tested the wireless control of the device in freely behaving animals within a range of enclosures. Spinal transection and device implantation

procedures were implemented as described above. Studies were performed in a large cage ($29 \times 36$ cm$^2$), a moving treadmill ($16 \times 56$ cm$^2$), and a small cage ($18 \times 36$ cm$^2$). The treadmill and small cages were instrumented with two-turn primary antenna at 4 cm and 8 cm height, while the large cage was instrumentalized with primary antenna windings at 4 cm, 8 cm and 12 cm height. The coils were connected to a tuner box and a NeuroLux system. The infrared receiver was positioned over the cage center to record stimulation parameters including intensities, channels, voltages, compliances, etc. from the implanted device while the animal was freely moving in the cages. A camera was placed above the small and large cages and lateral to the treadmill to track animal trajectories. Identified points on the animals were tracked using DeepLabCut 2.2.0.

We evaluated these procedures in one animal after a spinal cord injury. After the first implant and spinal surgery, the animal was allowed to recover and observed for five days. Then the animal went through initial tests in all three cages to familiarize it with the stimulation and cage environments and determine optimal RF power and stimulating parameters. The devices were programmed to automatically activate the transmitter antenna every 10 seconds and deliver a single pulse with the intensity of 282 μA with a pulse width of 114 μs. A 1-hour continuous stimulation session was then recorded in the large cage and in the small cage, and 30 min for treadmill locomotion on separate days. Two prolonged continuous stimulating sessions for six hours separated by three days, were then recorded in the small cage configuration.

## Mobile operation proof of concept
Proof of concept experiments were conducted using a small coil integrated with an NFC reader IC (STMicroelectronics, ST25R3911B-DISCO) and a 2 kΩ shunt resistor to match the operational load of the system during continuous operation while voltage measurements were taken on a digital multimeter (AstroAI, DM130B) to calculate harvested power.

## Statistics & reproducibility
The main focus of this study is to develop and test the functionality and safety of the wireless fully implanted devices for neural and muscle stimulation rather than to investigate their clinical intervention efficacy, accordingly, the sample size was not predetermined, and the experiments were not randomized. No data was excluded from the analysis.

## Reporting summary
Further information on research design is available in the Nature Portfolio Reporting Summary linked to this article.

## Data availability
The source data generated in the study has been deposited in and available at the University of Arizona Data Repository to be accessed from https://doi.org/10.25422/azu.data.24329134 and are also provided alongside this manuscript. All other data supporting the findings of this study are available within the article and its supplementary files. Any additional requests for information can be directed to, and will be fulfilled by, the corresponding authors. Source data are provided with this paper.

## Code availability
The code used to operate the device in this study have been deposited in the Center for Open Science database that can be accessed at https://doi.org/10.17605/OSF.IO/AN4BS.

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

## Acknowledgements

We acknowledge support from the imaging work performed at the Northwestern University Center for Advanced Molecular Imaging generously supported by NCI CCSG P30 CA060553 awarded to the Robert H Lurie Comprehensive Cancer Center. Imaging work was performed at the Northwestern University Center for Advanced Molecular Imaging (RRID:SCR_021192) generously supported by NCI CCSG P30 CA060553 awarded to the Robert H Lurie Comprehensive Cancer Center. We acknowledge support from National Institute of Biomedical Imaging and Bioengineering of the National Institutes of Health T32EB000809 (AB), ARCS Foundation (AB), The University of Arizona Department of Biomedical Engineering startup funds (PG), Core Facilities Pilot Program (CA-CFPP NANO-3310342) (PG), NIH-NINDS NS112535 (MCT).

## Author contributions

Alex Burton (A.Bu.), Alex Benedetto (A.Be.) Conceptualization: A.Bu., Z.W., D.S., M.C.T., P.G. Methodology: A.Bu., Z.W., D.S., S.T., J.H., L.E.M., M.C.T., P.G. Investigation: A.Bu., Z.W., D.S., S.T., J.H., D.A., J.B., D.C., J.A., R.P., K.S., A.Be., E.Y., D.B., L.E.M., M.C.T., P.G. Visualization: A.Bu., Z.W., S.T., J.H., J.B., D.C., L.E.M., M.C.T., P.G. Funding acquisition: M.C.T., P.G. Project administration: L.E.M., M.C.T., P.G. Supervision: A.Bu., Z.W., L.E.M., M.C.T., P.G. Writing – original draft: A.Bu., ZW, DS, JH, LEM, MCT, PG. Writing – review & editing: A.Bu., Z.W., D.S., J.H., L.E.M., M.C.T., P.G.

## Competing interests

The authors declare no competing interests.
