## [Peer Review File · Nature Communications]

REVIEWER COMMENTS

Reviewer #1 (Remarks to the Author):

- What are the noteworthy results?

The authors present a wirelessly powered, battery-free platform to enable chronic electrical stimulation of nerves and muscles in rodents. The platform is versatile, allows for high voltage compliances and a broad range of stimulus current intensities, has hence the potential to be used for a range of tissue targets and with a broad range of electrode-tissue impedances (which in turns means a broad range of electrode materials and sizes).

- Will the work be of significance to the field and related fields? How does it compare to the established literature? If the work is not original, please provide relevant references.

The basis of the platform using polyimide-copper-parylene with integrated discrete electronic components, as presented here, has been presented with minimal variations in several other publications from the authors [14, 15, 16]. The novelty of this paper is the addition of the resonator coil to increase the power harvested by the implant. This concept, however, itself is not new, and even its use in neural implants is also not novel, see:

S. A. Mirbozorgi, P. Yeon and M. Ghovanloo, "Robust Wireless Power Transmission to mm-Sized Free-Floating Distributed Implants," in IEEE Transactions on Biomedical Circuits and Systems, vol. 11, no. 3, pp. 692-702, June 2017, doi: 10.1109/TBCAS.2017.2663358

Y. Jia et al., "A mm-sized free-floating wirelessly powered implantable optical stimulating system-on-a-chip," 2018 IEEE International Solid - State Circuits Conference - (ISSCC), 2018, pp. 468-470, doi: 10.1109/ISSCC.2018.8310387.

- Does the work support the conclusions and claims, or is additional evidence needed?
- Are there any flaws in the data analysis, interpretation and conclusions? Do these prohibit publication or require revision?
- Is the methodology sound? Does the work meet the expected standards in your field?

Overall, the authors present some validation of their platform that demonstrates operation and promising results. However, in several occasions, tests have been performed without a proper justification of the test conditions, or at conditions that seem to yield more favourable results but are not the most stressful scenarios the implant would operate at (see my list below for specifics). Furthermore, I feel that several claims made in the article are out of proportion and not properly evidenced. Based on these, I feel the paper does not meet the rigorous standards Nature Communications sets for itself and I do not recommend this for publication in this journal.

Here is a list of such claims and points I feel need additional characterization/testing:

- o The authors claim that one of the main motivations for their proposed platform is to replace tethered setups to be able to evaluate neuromodulation or rehabilitation strategies involving continuous, uninterrupted stimulation protocols lasting multiple days or weeks. Specifically, the main claim of this paper, presented in lines 92-93: "Here, we introduce fully implanted devices that overcome these challenges, making them suitable for long term, uninterrupted functional electrical applications." Long-term, uninterrupted electrical stimulation is not demonstrated here.

- o Fig. 2c: demonstrates the power harvesting capabilities and angular misalignment at the centre of the arena, where power will be maximum. Simulations of the power harvesting capability are presented in Supplementary Fig. 6 for 2 heights in the treadmill cage. Especially with regards to the claims of the authors that this work is suitable for uninterrupted stimulation in freely behaving animals, the rigour of the characterisation presented here is not sufficient to support these claims. What are the limitations of this power harvesting in terms of the complete arena design? What is the minimum power and at which locations is this expected?

- o The antenna presented here is placed on a flexible circuit and spans an area large enough to expect that certain deformation will occur as animals move during behavioural experiments. Since the paper claims the proposed platform is suitable for such investigations, I would expect to see a measurement-based characterization of the power transfer in different scenarios of the transmitting

coil and during various relevant animal (hence antenna) movements, and documented cases when stimulation is not possible because of power unavailability (since this is a battery-less device).

o Claims regarding electrode stability:

Stability of electrodes and stimulation is only shown for what the authors describe as a typical FES protocol over 1 hour per day (line 306), and not the uninterrupted scenarios previously claimed in the document.

Accelerated lifetime test (line 307): should be performed at the maximum current intensity and continuous stimulation over a 6 week study to support the claims of the authors that the platform can be used for uninterrupted chronic stimulation.

Supplementary Fig. 14 a: why 1mA was chosen for this test (and not higher current)? zoomed in versions of the electrode sites should be presented, the images as shown now are not informing. "Minimal changes in electrode impedance and minimal degradation of electrode surfaces." (line 308 and 309): Please avoid the use of vague terms such as "minimal". There is an obvious change in electrode impedance, even though stimulation is performed at intensities below maximum. Please quantify the results and present them as percentages.

o Claims regarding stimulation safety:

Residual charge is presented for a 1.8 mA stimulation current (line 303), however, the platform is capable of a maximum of 4.7 mA. The residual charge should be calculated for this scenario as well. Reference 49, which is presented as argumentation regarding the limit of allowable residual charge before irreversible damage to either the tissue or electrode, does not present any proof of this (other than very vaguely mentioning a safety limit of 15nC/phase without any further comments or references. If the authors would like to claim stimulation safety they need to do a more rigorous investigation on this topic and back their claims up properly.

Residual charge is not the only, or even not the main, indication of stimulation safety. In fact, looking at the voltage between the anode and cathode of the system during stimulation is a much more informative technique. In particular, the water window should be respected for the electrode polarization potentials, and the interpulse voltage should remain within certain limits. In addition, the charge density and charge per phase are also of importance for safe stimulation and these can be captured to an extent by the Shannon equation, in particular for macroelectrodes. See:

1. S. F. Cogan, K. A. Ludwig, C. G. Welle and P. Takmakov, "Tissue damage thresholds during therapeutic electrical stimulation," *Journal of Neural Engineering*, vol. 13, no. 2, April 2016.
2. Cogan, S. F. Neural stimulation and recording electrodes. *Annu. Rev. Biomed. Eng.* 10,275–309 (2008).
3. K. Kolovou-Kouri, S. Soloukey, F. J. P. M. Huygen, B. S. Harhangi, W. A. Serdijn and V. Giagka, "Dorsal Root Ganglion (DRG) Versatile Stimulator Prototype Developed for Use in Locomotion Recovery Early Clinical Trials," 2021 10th International IEEE/EMBS Conference on Neural Engineering (NER), 2021, pp. 1125-1129, doi: 10.1109/NER49283.2021.9441101.

• Line 125: the authors claim that the use of self-similar serpentine are used to improve device lifetimes: but this is not evidenced in the article.

Other feedback/comments/typos

References 38 and 39 have the same title and details, please check these and correct.

Line 88: "targeting multiple substrates" and line 89 "neural substrate": this terminology, in the context of research involving flexible devices, is confusing. I would strongly advise to replace the term substrates with a more accurate description (body areas? Activatable tissues? – since you are proposing muscle activation and not only neural tissue). Also, please explain why these need such different ranges of voltages and currents (the type of tissue, but also the electrode characteristics that affect this) and add appropriate references (now a paper of the authors' is cited as evidence for this claim, but the cited paper does not present detailed investigation into this issue, which I feel is not self-evidenced and merits a more thorough documentation and proper reference list).

Line 99: replace "flexible" with "versatile" (flexible can be confusing in the context of flexible electronics)

Line 123: I feel the term "self-similar" needs to be properly explained

Figure 1, a: macroelectrodes vs microelectrodes: please add sizes (electrode area), and references for these ranges claimed here. References should be articles that show neurostimulation studies in the

areas mentioned, and with documented electrode sizes. Also, I feel the labels are not correct, microelectrodes are probably used for neurons and brain applications, vs macroelectrodes for the heart and muscle (which is the case from what I see further down in the article, lines 268-272).
Fig. 3: Impedance spectroscopy data should be presented as impedance magnitude, and phase (Bode plots). Please add these and correct the "resistance" labels to "impedance magnitude" in Fig. 3e, Supplementary Fig. 9, and Supplementary Fig. 14 b and c.
Fig. 4: b: This is not a Bode plot, as claimed in the caption, please correct this. Also, please add information in the figure about the frequency at which this impedance is measured.
Line 361: "displacements and strains are well beyond those that might occur during implantation": please support this claim with values and references.

Line 86: "Existing devices also typically do not involve the dual power supplies required for biphasic, charge balanced stimulation protocols that are necessary to minimize tissue and electrode damage from chronic stimulation." This statement is untrue. Dual power supplies are NOT required for charge balanced stimulation protocols. There are other ways to implement charge balancing (which fundamentally depends on the remaining charge across the electrode-tissue interface and cannot be predicted with charged balanced waveforms.)

Also, line 299: "Biphasic pulses ensure charge balancing, to avoid charge accumulation and irreversible electrochemical processes at the electrode surface which can result in tissue damage or electrode degradation": it is common from literature that biphasic pulses do not, in fact, ensure charge balancing, see, among others, e.g.: 10.1146/annurev.bioeng.10.061807.160518 and N.d.N. Donaldson, P.E.K. Donaldson, "When are actively balanced biphasic ('Lilly') stimulating pulses necessary in a neurological prosthesis? I Historical background; Pt resting potential; Q studies," Med. Biol. Eng. Comput., vol. 24, pp. 41-49, January 1986. Please correct this statement.

Supplementary Fig. 11: please elaborate on what this figure demonstrates

Supplementary Fig. 12b: how should we interpret the impedance values? Were the output stages of the electronics connected to resistors of this nominal value? If so, why is the voltage waveform for e.g. C6 not a rectangular one? Normally electrodes are approximated by capacitors and resistors (in series or parallel) but this does not seem to have been the case here? If so, please mention the values of the capacitors used. In any case, please justify the selected load with respect to what is expected from an application perspective (e.g. why a load of 16.5 kom?) Also, in line 290, a maximum current error or +/- 8uA is mentioned, how is this calculated/measured?

Fig. 12e: what is M1 and M2? Please elaborate on what each part of this figure demonstrates.

- Is there enough detail provided in the methods for the work to be reproduced?

The methods described here go to a depth and detail level which is common in published articles. I have my reservations regarding the extent these allow work to be reproduced, but this is not the authors' fault, they have followed common practice.

Reviewer #2 (Remarks to the Author):

In this manuscript, Gutruf and colleagues demonstrated an implantable transient power delivery system based on a passive resonator approach coupled with active power electronics to convert alternating current to DC voltage. This approach was aimed to be suitable for powering embedded systems such as microcontrollers to create a customizable in vivo electrical stimulation. They demonstrated the capacity of this approach by developing in vivo rodent experiments that involved delivery of electrical stimulation to peripheral nervous system structures. While the manuscript is interesting, there are several issues that require the authors' attention.

- Authors provide a nice introduction about the current challenges in powering implantable devices. Specifically, for transiently powered, coiled-based systems "the amount of power ... depends on the inductance ratio and coupling between transmitter and receiver." However, the authors do not provide a clear statement as to what are the key advances in their approach. How does their design overcome these limitations or improve the current the efficiency of power delivery?

- line 219: Authors demonstrate 500% improvement in power delivery compared to the previously

reported experiments. What design element was responsible for this improvement?

- Authors claim the center-tapped antenna enabled a dual voltage supply system. It would be important for the authors to provide justification for their approach given its added complexity compared to power rail splitting where the ground is assigned to a middle potential.

- Authors provide a simple validation of the device's ability to deliver electrical stimulation. It would be important to demonstrate the specific use and advantages of having a multi-channel electrical stimulation.

- Why do the authors solely reply on experiments where the rodent is fixated? One could argue that the choice of such an experimental set-up does not demonstrate the capacity of such a system; wired implants could easily function in this set-up.

- What new experimental capabilities could this technology enable? How do the authors aim to translate this technology? These should be added in the discussion section.

Reviewer #3 (Remarks to the Author):

This manuscript proposes a novel passive resonator optimized power transfer design, as an alternative to conventional battery packs for the application of electrical stimulation for assistance or rehabilitation. The motivation for higher power wireless implanted systems is well stated and logically arranged. The submission includes detailed design information, methodology and overall specifications. Evaluation is conducted including extensive testing on rats with SCI. Results also include strain tests and long term testing. The vivo tests are comprehensive, showing stimulation parameters and resulting kinematics. Overall, the system is a significant advancement in current technology. The technical material is rigorous and sufficiently detailed, however more detail could be added to place the device in the broader context of its eventual application area:

1) Current battery-free systems [refs 16,22,23] are mentioned (page 4) however it would be useful to provide more information on these devices --- e.g. a brief summary of their intended application area, whether tested with human subjects in human trials, and perhaps a comparison table of main specifications. It may be useful to also mention implanted FES stimulators that contain a battery, but are recharged wirelessly, e.g. BION stimulators [1]. This would assist the reader in understanding the wider context within human assistance/rehab.

[1] Schulman JH, Mobley JP, Wolfe J, Regev E, Perron CY, Ananth R, Matei E, Glukhovskiy A, Davis R. Battery powered BION FES network. Conf Proc IEEE Eng Med Biol Soc. 2004;2004:4283-6. doi: 10.1109/IEMBS.2004.1404193. PMID: 17271251.

2) The clinical application to rats is detailed and comprehensive. However more information on how the technology could transfer to human users would be useful (perhaps a short paragraph in the discussion). In this context, is the voltage range, stimulation frequency and pulsewidth sufficient to elicit human muscle contracts? Would the cage antenna system be suitable to extend to human participants, perhaps in their own homes.

Minor typos: Fig 2 contains spelling errors: receiver, wireless

Reviewer response key:

Reviewer comments in black

Response to reviewer comments in green

Changes to the manuscript in red

Reviewer #1 (Remarks to the Author):

We thank the reviewer for their very detailed comments that provide a foundation for us to revise our manuscript and clarify concerns.

Comment 1:

- What are the noteworthy results?

The authors present a wirelessly powered, battery-free platform to enable chronic electrical stimulation of nerves and muscles in rodents. The platform is versatile, allows for high voltage compliances and a broad range of stimulus current intensities, has hence the potential to be used for a range of tissue targets and with a broad range of electrode-tissue impedances (which in turns means a broad range of electrode materials and sizes).

Our Response: We thank the reviewer for their positive feedback on the device capabilities.

Modifications to manuscript:

None

Comment 2:

- Will the work be of significance to the field and related fields? How does it compare to the established literature? If the work is not original, please provide relevant references.

The basis of the platform using polyimide-copper-parylene with integrated discrete electronic components, as presented here, has been presented with minimal variations in several other publications from the authors [14, 15, 16]. The novelty of this paper is the addition of the resonator coil to increase the power harvested by the implant. This concept, however, itself is not new, and even its use in neural implants is also not novel, see:

S. A. Mirbozorgi, P. Yeon and M. Ghovanloo, "Robust Wireless Power Transmission to mm-Sized Free-Floating Distributed Implants," in IEEE Transactions on Biomedical Circuits and Systems, vol. 11, no. 3, pp. 692-702, June 2017, doi: 10.1109/TBCAS.2017.2663358

Y. Jia et al., "A mm-sized free-floating wirelessly powered implantable optical stimulating

system-on-a-chip," 2018 IEEE International Solid - State Circuits Conference - (ISSCC), 2018, pp. 468-470, doi: 10.1109/ISSCC.2018.8310387.

Our Response: We thank the reviewer for the chance to modify our novelty statement to be more specific and highlight the advance of the presented work over the state of the art, including that from our own groups. Specifically, we would like to point out:

1. The work from our group highlighted by the reviewer is only similar with regard to the presented work in the use of powering frequency and overall flexible material platform used. Other aspects of the current work are substantially distinct from those previous studies. The work referenced by the reviewer targets 2 different organ systems, the heart (14), the brain (16) (reference 15 is a review article) and the configurations and challenges for those systems are vastly different from the systems considered here. For example, the stimulation mode for the heart was optogenetic and so was substantially different. The system described in our work here is also fundamentally distinct because the hardware scheme for this system is capable of constant current controlled stimulation with a wide voltage range, substantially surpassing the capabilities of our previous work and of other published approaches. For these reasons, we believe that the system described in our work is clearly novel and will have significant impact in the field.

In addition, the use of the flexible material platform used in previous work is intentional as our overall goal is to make the devices we develop in all of our work, including that described here, available and affordable to the academic community. This requires a material system that can be manufactured at scale and at a cost that enables single use implants, since chronically implantable soft devices are generally impossible to recovery and reuse. This is why we do not use ASICs, use techniques that enable automated assembly and why we put a lot of effort into monolithic integration. A great deal of effort has gone into the design of the flexible substrate to enable soft mechanics, circuit simplification, and firmware design enabling repeatable and scalable devices. We see this as a core novelty and a very important aspect of the demonstrated devices and we have highlighted this in our modified introduction. We also note that well over 300 labs worldwide have 13.56 MHz powering systems (Neurolux Inc.) that can be used to operate the devices presented in this work, further suggesting the impact of this technology. We have addressed this by main manuscript text changes accompanied by a table in the supplementary information that compares our against other work.

Modifications to the manuscript (page 5, line 101):

Devices are typically capable of delivering either low or high ranges of currents/voltages, but not both ranges simultaneously **that allow for simultaneous electrical stimulation of multiple neuromuscular sites and for multichannel abilities.**

Here, we introduce fully implanted devices that overcome these challenges, **providing a voltage range of up to $\pm 20V$, current controlled stimulation with a range of 40 μA to 4.7 mA and independent control of 8 channels in a fully implantable package that is powered at distance. This device represents the highest figure of**

merit for all categories (Supplementary Table 1)^{23,24,35-41}, making them suitable for long term, uninterrupted functional electrical applications. for preclinical investigations in freely moving small to medium animal subjects.

Modifications to the manuscript (page 5, line 111):

The devices exploit a passive resonator scheme that substantially increases power harvesting capabilities at a designed voltage, making it possible to stimulate ~~a wide range of biological substrates~~ multiple areas of the nervous system simultaneously, using the same implanted device. The circuit design incorporates off-the-self components providing scalable manufacturing at a cost that enables single use deployment which is critical for chronic implantation that make reuse impossible or difficult. The devices are fully implanted, are powered and controlled wirelessly, and are fabricated from flexible electronics. The materials and mechanicals design of the flexible substrate provides a platform that integrates components to conform with the surrounding soft tissue using a network of ridged islands and flexible interconnects. As a consequence, ~~they~~ the devices have minimal impact on overall animal behavior and can remain operational for extended periods of at least 6 weeks.

Modifications made to the manuscript (page 7, line 173):

The dual voltage supply is achieved using a center-tapped antenna design with two single half-bridge rectifiers to create the positive and negative voltages with no external components and only minimally increased device footprint. ~~The receiver antenna uses a center tapped antenna to minimize circuit complexity while allowing for dual voltage supplies. A dual supply coupled with a precision current driver allows for a smaller circuit design (30 mm²) suitable for implantation while providing higher power efficiencies compared to designs using DC/DC converters to generate dual voltage supplies⁴². This is significantly smaller than traditional systems using a single supply to have the same biphasic stimulation response using a Howland current pump, H-bridge, and current monitor each requiring additional electrical traces and passive components (>50 mm²)⁴³⁻⁴⁵. A power management system incorporates voltage protection using Zener diodes and a step-down converter accompanied by a linear voltage regulator for stable operating voltages for the digital circuit.~~

Addition to the supplementary figure (Supplementary Table 1)

Publication	Implanted Duration	Electrical Construction	Mechanics	Dimensions	Mode of WPT	Channels	Stimulation Capability	Stimulation Range	Compliance Voltage	Freely moving	Available Power
This Work	6 weeks	Off-the-shelf Components	Flexible	40 mm x 20mm x 1 mm	MRC	8	Biphasic Current Programmable	4.7 mA	± 20V	Yes	120 mW
³⁵	Acute	Off-the-shelf Components	Rigid	10.0mm x 8.5 mm x 2.4mm	High Frequency volume conductive	1	Biphasic Constant Current	2mA-4mA	2500 mV	No	5 mW
³⁶	Not Available	Off-the-shelf Components	Rigid	11 mm diameter x 5 mm thickness	Ultrasound	1	Voltage Programmable	6V at 10k ohm load	22.16V	No	5.98 mW
²³	Not Available	ASIC	Rigid	3 mm x 2.15 mm x 14.8 mm	MagnetoElectric	1	Biphasic Voltage Programmable	3.3 V at open load	3300 mV	No	4 mW
²⁴	Not Available	ASIC	Rigid	3 mm x 2 mm x 1 mm	Ultrasound	1	monophasic	0.4 mA	3300mV	No	65 µA
³⁷	Not Available	Off-the-shelf Components	Rigid	0.5 mm x 0.5 mm x 2.3 mm	Inductive Coupling	1	monophasic	25 µA at 10k ohm load	250 mV	No	5 mW
³⁸	Not Available	ASIC	Rigid	2.5 mm x 2.3 mm x 23 mm	MRC	1	Biphasic Current Controlled	0.7 mA	3300 mV	No	3.1 mW
³⁹	40 days	ASIC	Rigid	25.3 mm x 9.3 mm x 1.9 mm	Inductive	8	Biphasic Voltage Programmable	100 µA at 2k ohm load	10.5 V	No	Not Available
⁴⁰	4 days	ASIC	Rigid	30 mm x 15 mm x 5 mm	Inductive	36	Biphasic Current	525 µA at 1k ohm	2 V	Yes	43 mW
⁴¹	2-3 weeks	ASIC	Flexible	12 mm diameter x 0.2 mm	MRC	1	Voltage	N/A	500 mV	No	N/A

Supplementary Table 1. List of wireless battery-free systems for functional electrical stimulation.

- Although the references cited by the reviewer use resonators to improve power harvesting, the system configurations are fundamentally different from those presented here. In our devices we co-locate the resonator with the harvesting antennas thereby substantially improving harvesting efficiency over previous work. Further, our design enables us to tune the maximum power point to the desired operation voltage while still locating all components on one dual sided flexible substrate and using minimal passive components. This is a new approach and enables high power availability in large experimental enclosures without the need to locate powering coils directly on the subject. The resonators used in the work cited by the reviewer are implanted as free floating and separate from the harvesting antennas and other circuit components. The lack of integration with the harvesting antennas limits the performance of those systems as optimal performance might be compromised due to migration of either the resonator or the implanted antennas. Further, the free floating design, can be difficult to surgically implant correctly to target specific neural substrates. It is also worthwhile to note that coils providing power are directly located above the body for both of these publications, whereas our design uses power supplies located outside of animals' cages thereby enabling studies with freely behaving animals.

The cited articles also do not produce nearly enough current or compliance to drive muscle stimulation, further demonstrating the novelty of our overall device design.

In our literature research we could not find any wireless and battery free device that provides required compliance and power that is fully implantable. To make these points clearer we have modified the manuscript to outline these points of novelty.

Modifications to the manuscript (page 9, line 211):

In order to overcome these challenges, we introduce a design that imbeds a passive resonator directly on the implanted device, improving power transfer efficiency and enabling high voltage compliances. **The monolithic integration removes the need for auxiliary circuit components such as resonators that are located elsewhere in the tissue. This design can also be easily implanted, enabling it to be used as a scalable platform for chronic applications.** This design also requires only a small number of passive components, resulting in small device footprints and high efficiencies compared to active approaches such as maximum powerpoint tracking and management IC`s.

3. We would like to also highlight that the technological demonstrations outlined in points 1. and 2. above are not the only accomplishments that contribute to the novelty of this work. The demonstrations presented here, namely fine control over hindlimb motor function and gait pattern stimulation with a fully implanted and battery-free device have not been accomplished elsewhere. There is an existing body of work in rodents from the Courtine and Lacour lab^{R1}, however the approach is based on optogenetics, the electronics are externalized and battery powered and accomplishments do not show chronic stimulation. Referring back to point 2, for the neuroscience community a one-time use chronic device that is easily deployable, such as that described here, is critical to drive mechanistic and pre-large animal studies.

[R1] Kathe, Claudia, et al. "Wireless closed-loop optogenetics across the entire dorsoventral spinal cord in mice." *Nature biotechnology* 40.2 (2022): 198-208.

Modifications to the manuscript (page 5, line 121):

Further, the devices are able to deliver highly flexible, precisely controlled, and charge-balanced stimulation and **are designed for scalable manufacturing that enables easily deployable one-time use systems to deliver long-term stimulation.** Taken together, these advances in device performance enable the development and evaluation of a range of complex chronic stimulation protocols for functional restoration that have previously been unattainable.

Comment 3:

- Does the work support the conclusions and claims, or is additional evidence needed?
- Are there any flaws in the data analysis, interpretation and conclusions? Do these prohibit publication or require revision?

- Is the methodology sound? Does the work meet the expected standards in your field?

Overall, the authors present some validation of their platform that demonstrates operation and promising results. However, in several occasions, tests have been performed without a proper justification of the test conditions, or at conditions that seem to yield more favourable results but are not the most stressful scenarios the implant would operate at (see my list below for specifics). Furthermore, I feel that several claims made in the article are out of proportion and not properly evidenced. Based on these, I feel the paper does not meet the rigorous standards Nature Communications sets for itself and I do not recommend this for publication in this journal.

Our Response: We thank the Reviewer for their opinion and detailed points for which we provide new experiments, references, and clarifications to improve the rigor and statements of significance of the work.

Modifications to manuscript:

None

Here is a list of such claims and points I feel need additional characterization/testing:

Comment 4:

o The authors claim that one of the main motivations for their proposed platform is to replace tethered setups to be able to evaluate neuromodulation or rehabilitation strategies involving continuous, uninterrupted stimulation protocols lasting multiple days or weeks. Specifically, the main claim of this paper, presented in lines 92-93: “Here, we introduce fully implanted devices that overcome these challenges, making them suitable for long term, uninterrupted functional electrical applications.” Long-term, uninterrupted electrical stimulation is not demonstrated here.

Our Response: We thank the Reviewer for their considerations and agree that we could have demonstrated a wider range of performance of our device in the previous version of the manuscript. We first note, however, that our observation that devices operated successfully over triweekly multi hour stimulation sessions for 6 weeks of time in multiple subjects, clearly demonstrated a critical aspect of the long-term functional capabilities of our devices. Moreover, our bench tests on the long term functionality of these devices demonstrated their robust design and suitability for long term use. Both observations suggest that our system can be used for long-term, chronic studies.

However, we agree that our previous manuscript did not directly demonstrate the ability of our system for continuous operation in freely behaving animals, which is an important aspect of our proposed applications (e.g. in cage rehabilitation protocols in freely behaving animals). To more directly address the reviewer’s point (and that of the other

reviewers) we have performed additional experiments in the revised manuscript. Specifically, we performed experiments with continuous repeated stimulation in freely behaving animals for extended periods of time (up to 6 hours) on repeated days. We characterize stimulation compliance in freely behaving subjects in a cage and on a treadmill and we provide a characterization of system voltages during free behaviors for a variety of cages. Comparable performance using wired devices or battery powered devices would require frequent re-connection or adjustment of cables or recharging of batteries. Although longer term stimulation (e.g. 24 hour, uninterrupted protocols) were not possible with our currently approved protocols and animal facilities, we also performed accelerated rate tests that show that our device and electrodes remain functional over the periods required for chronic, uninterrupted stimulation protocols. We compared the performance our system to that described by others in the literature and found that our devices provide a 5 fold improvement on operational lifetime, 17% improvement on stimulation range and more than 500% power harvesting for wirelessly powered devices (see supplemental table 1 and response to comment 2). We thank the reviewer for this suggestion as we believe that it makes the capabilities of our device clearer and strengthens the manuscript.

Modifications to the main figure (Figure 7):

Figure 7. Stimulation-evoked movements in spinal cord transected animals. (a) An awake, spinal cord transected (T10) animal is stimulated. **(b)** A limb propulsion movement is evoked from stimulation of a spinal electrode site (c3) while the animal is locomoting on a treadmill with unparalyzed forelimbs. **(c)** Stimulation of spinal electrode site (c0) evoked limb propulsion and retraction. **(d)** Stimulation in cage with freely moving spinal cord injured animals. Middle plot illustrates the average voltage compliance at each region of the cage. Right plot illustrates movement trajectories (blue) measured over 2 hours. Green marks indicate points at which stimulation is compliant; red marks indicate uncompliant stimulation. **(e)** Freely behaving animal on a moving treadmill (conventions follow (d)). **(f)** Voltage compliances on individual trials over two 6-hour sessions and sessions recorded 25 and 28 days following device implantation. **(g)** Distributions of

voltage compliances measured in three different enclosures of increasing size: large cage (29X36cm), treadmill (16X56cm), small cage (18X36cm).

Modifications to the manuscript: page 24, 587)

The greater variability evoked from spinal sites (Fig. 7b vs. Figure 6f, Supplementary Fig. 239) likely reflects variations in spinal excitability present in unanesthetized animals (e.g. due to variations in sensory inputs).

To evaluate system reliability with freely behaving subjects we perform stimulation during free behavior in cages and on a treadmill (Fig. 7d,e). Subjects are tracked with markers positioned on the pelvis and back. Compliance voltage, current and pulse timing are communicated by the implant and synchronized with the position of the subject.

Fig. 7d illustrates results from a 2h session in a cage using 282 μ A stimulation current and a pulse width of 114 μ s every 10 s. In this case, 694/694 stimulation events trials are compliant (i.e., the desired current is delivered). The system voltage compliance is 15.3 \pm 2.5V averaged across all trials. Fig. 7e illustrates evaluation of system performance in during locomotion on a moving treadmill with the same stimulation parameters. The enclosure area is larger (16X56cm vs 18X36cm) and results in lowered average system voltage of 12.70 \pm 1.0V and a larger number of trials are uncompliant (52/219 trials). It should be noted however that stimulation even in the case of a noncompliant event is still delivered, although with slightly reduced current.

The devices enable long lasting, uninterrupted stimulation in freely behaving animals. System voltages measured in two 6 hour sessions with continuous stimulation (282 μ A, 114 μ s pulses, repeated every 10s) are illustrated in Fig. 7f for a small cage enclosure. These sessions are recorded 25 and 28 days after the implantation of the device. The average system voltage for day 25 is 16.2 \pm 2.4 V and 15.8 \pm 2.8 V for day 28 with only intermittent drops in system voltage. System performance in a variety of cages during free behavior is displayed in Fig. 7g. As expected, voltage compliance is reduced in larger enclosures (29X36cm); however, even in the largest enclosures sufficient power is harvested to enable stimulation at high voltages (9.9 \pm 2.8V).

Modifications to the manuscript: page 27, 643)

Device efficacy is demonstrated in both healthy animals and in animals with chronic spinal cord injuries with consistent efficacy and stability over 6 weeks of operation. These capabilities are illustrated by the production of consistent amplitude movements from both spinal and muscle sites using similar ranges of currents over several weeks, demonstrating the consistent ability of devices to generate currents and voltages necessary for neuromodulation or functional restoration. **Chronic stability of devices is demonstrated by stable delivery of stimuli in freely behaving subjects in environments relevant to FES research and enabling operational durations that are not achievable with battery powered systems.** Characteristics of the devices demonstrated here make them suitable candidates for automated investigation of rehabilitation of neural circuits on clinically relevant timescales and with adequate statistical power to investigate underlying

mechanisms that currently prevent widespread clinical adoption.

Modifications to the manuscript: page 39, line 896):

Device Characterization in Freely Behaving Subjects

We tested the wireless control of the device in freely-behaving animals within a range of enclosures. Spinal transection and device implantation procedures were implemented as described above. Studies were performed in a large cage (29X36cm²), a moving treadmill (16X56cm²) and a small cage (18X36cm²). The treadmill and small cages were instrumented with two-turn primary antenna at 4cm and 8cm height, while the large cage was instrumented with primary antenna windings at 4cm, 8cm and 12cm height. The coils were connected to a tuner box and a NeuroLux system. The infrared receiver was positioned over the cage center to record stimulation parameters including intensity, channel, voltages and compliances etc. from the implanted device while the animal was freely moving in the cages. A camera was placed above the small and large cages and lateral to the treadmill to track animal trajectories. Identified points on the animals were tracked using DeepLabCut.

We evaluated these procedures in one animal after spinal cord injury. After the first implant and spinal surgery, the animal was allowed to recover and observed for five days. Then the animal went through initial tests in all the three cages to familiarize it with the stimulations and cage environments and determine optimal RF power and stimulating parameters. The devices were programmed to automatically activate the transmitter antenna every 10 seconds and deliver a single pulse with intensity of 282 μ A with a pulse width of 114 μ s. A 1-hour continuous stimulation session was then recorded in the large cage and in the small cage, and 30 min for treadmill locomotion on separate days. Two prolonged continuous stimulating sessions for six hours separated by three days, were then recorded in the small cage configuration.

Comment 5:

o Fig. 2c: demonstrates the power harvesting capabilities and angular misalignment at the centre of the arena, where power will be maximum. Simulations of the power harvesting capability are presented in Supplementary Fig. 6 for 2 heights in the treadmill cage. Especially with regards to the claims of the authors that this work is suitable for uninterrupted stimulation in freely behaving animals, the rigour of the characterisation presented here is not sufficient to support these claims. What are the limitations of this power harvesting in terms of the complete arena design? What is the minimum power and at which locations is this expected?

Our Response: We thank the reviewer for this comment and the chance to add experimental insights to further document performance. Firstly, we have chosen the center of the cage because power transfer is lowest at this point, as shown from finite element simulations and measurements. Power transfer is maximum at the center of the arena only when antennas are small and circular – see handheld antenna characterization in figure S3. This can be seen from measurements presented in figure S6 (they are not simulations; simulations are presented in figure S4). From these measurements we have a minimum of 325 mW available at RF power levels of 6 W, this is x5 more power than the maximum power consumption of the device (all channels full current and voltage). Therefore, we are confident that devices can provide uninterrupted stimulation. To further evaluate this issue we have characterized spatially resolved angular misalignments within the cage at multiple operational levels (corresponding to rats and mice). This represents a scenario when animals are rearing (represents 1% during light periods and 7% in dark periods of typical behavior).^{R2} We have added additional description and relevant literature to the manuscript to better convey this point. It is important to note that these experiments are performed in free space, this is the standard in the field because tissue absorption is low at 13.56 MHz and near field coupling. In addition to these characterizations we have now characterized the harvested voltage and current compliance at the implant during chronic experiments as outlined in response to comment 5.

[R2] Makowska, I. Joanna, and Daniel M. Weary. "The importance of burrowing, climbing and standing upright for laboratory rats." *Royal Society open science* 3.6 (2016): 160136.

Modifications to the manuscript (Page 10, Line 255)

If needed, power at the implant can be increased by increasing RF power of the transmitter up to a maximum safe specific absorption rate of $< 20 \text{ mW kg}^{-1}$, which corresponds to a RF output power greater than the capabilities of the power amplifier (max 12 W) used in this work. By increasing RF power to 6 W, up to 325 mW of power can be harvested in the center of the enclosure (Supplementary Fig. 67d), **which is 5 times greater than the power consumption of comparable devices in the literature providing a large operational margin to enable continuous and uninterrupted stimulation⁵⁹.** As a comparison, recently reported devices using a smaller transmitter antenna (45 cm × 12 cm; 540 cm²) and similarly sized receiver antenna (3.5 cm × 2.5 cm; 8.75 cm²) were able to harvest 50 mW using 4 W of RF power in the transmitter.

Modifications to the manuscript (Page 11, Line 272)

The increased power harvesting capability also improves operational device stability during behaviors such as rearing, climbing, and preening, which often result in misalignment between antennae and a reduction of harvested power. **The behavior that causes greatest misalignment is rearing which represents 1% of total behavior time during light periods and 7% in dark periods⁶¹.** To obtain an average of 51 mW using our design, sufficient for high powered device operation, a 2 W RF power transmitter antenna can

operate at an angle up to ± 54.4 deg; however, if higher misalignment tolerance is needed increasing RF power can be increased to 5 W allows to enable for stable operation up to ± 75.3 deg as shown in Fig. 2d. Thus, with adjustments to the field power, our design can achieve substantial power transfer even for behaviors such as rearing which likely reflects the most severe case of misalignment.

Comment 6:

o The antenna presented here is placed on a flexible circuit and spans an area large enough to expect that certain deformation will occur as animals move during behavioural experiments. Since the paper claims the proposed platform is suitable for such investigations, I would expect to see a measurement-based characterization of the power transfer in different scenarios of the transmitting coil and during various relevant animal (hence antenna) movements, and documented cases when stimulation is not possible because of power unavailability (since this is a battery-less device).

Our Response: This is an excellent point brought up by the reviewer. We have characterized the max. deformation of the device and therefore the antenna can experience in figure 5b which is a bending radius of 3.4 cm, this radius is obtained by our physiological characterizations. We have characterized the loss of power harvesting ability which is 20% therefore the dominating loss in harvesting is angular misalignment. We have also deformed devices beyond physiologically possible state to provide insight for use in other animal models. Please also note the additions that characterize performance in vivo in response to reviewer comment 4.

Modification made to the manuscript (page 11, line 276):

To obtain an average of 51 mW using our design, sufficient for high powered device operation, a 2 W RF power transmitter antenna can operate at an angle up to ± 54.4 deg; however, if higher misalignment is needed increasing RF power can be increased to 5 W allowsto enablefor stable operation up to ± 75.3 deg as shown in Fig. 2d. Thus, with adjustments to the field power, our design can achieve substantial power transfer even for behaviors such as rearing which likely reflects the most severe case of misalignment. Additionally devices implanted on the back of rats are subjected to bending. Physiological characterization describes a maximum possible bending radius of 3.4 cm (Fig. 5b) which results in a reduction of power harvesting capabilities by ~20% as shown in Supplementary Fig. 8. These variations in device orientation and curvature can be compensated for by using additional RF power at the cost of increased energy consumption. The average current consumption of the device on startup is 2.5 mA. During stimulation it averages 3.3 mA when supplied with ± 20 V, as shown in Supplementary Fig. 67e and Supplementary Fig. 79a.

Addition made to the supplemental figures (Supplementary Figure 8):

Supplementary Fig. 8. Bending characterization of the antenna measuring harvested power with a 2K Ω load for increasing radius of curvatures for 2W and 4W of RF power. Physiologically relevant range marked in green.

Modification made to the manuscript (page 30, line 722):

Power harvested during angular misalignments between -90° and 90° relative to the cage floor between the transmitter antenna and receiver antennae was measured. Power harvesting was tested in the center of a 56 cm x 16 cm x 16 cm cage with 2W and 4W of RF power while bending the device over a fixed radius of curvatures controlled by 3D printed bending fixtures between 2 cm and the device lying flat in 1cm increments. Transient current consumption was measured using a modified current meter (LowPowerLab; CurrentRanger) and acquired on an oscilloscope (Siglent; SDS 1202X-E) while powered with 10 V, 15 V, and 20 V supplies.

Comment 7:

o Claims regarding electrode stability: Stability of electrodes and stimulation is only shown for what the authors describe as a typical FES protocol over 1 hour per day (line 306), and not the uninterrupted scenarios previously claimed in the document.

Our Response: As outlined in response to comment 2 we have performed several new experiments to demonstrate chronic stability of the platform. Firstly, we have performed new chronic *in vivo* experiments that demonstrate successful device performance for

periods well beyond what a typical battery powered device is capable of. We have also performed several bench accelerated lifetime characterizations, demonstrating that the device is capable of performing for extended periods necessary for long term, chronic experiments. Please see additions to the manuscript in response to comment 4.

Comment 8:

Accelerated lifetime test (line 307): should be performed at the maximum current intensity and continuous stimulation over a 6 week study to support the claims of the authors that the platform can be used for uninterrupted chronic stimulation. Supplementary Fig. 14 a: why 1mA was chosen for this test (and not higher current)? zoomed in versions of the electrode sites should be presented, the images as shown now are not informing.

Our Response: We thank the reviewer for the opportunity to clarify and have added experiments on accelerated lifetime tests as requested by the reviewer. Firstly, it is important to note that we show 2 classes of electrodes in our manuscript. 1mA of current is very high for spinal stimulation and after a literature review, we have found 1 mA^{R3,R4,R5,R6} in the spinal cord of rats as the max delivery. This is why 1mA was chosen for the accelerated lifetime tests for those electrodes.

Secondly, we have now provided an extensive characterization of the electrodes at accelerated rates as requested by the reviewer, and we have performed additional in vivo experiments demonstrating that these devices are suitable for chronic stimulation. It is also important to note that we use materials for the electrodes(gold) that are not optimized for longevity; this is a conscious decision because gold is an electrode material that is available from flex PCB manufacturers enabling the devices to be created at scale at low cost. If particularly demanding experiments such as high current spinal stimulation or peripheral nerve stimulation over especially long timescales is required, an additional electrode coating step can be performed to increase performance (such as platinum black coating outlined in our recent brain stimulation work).^{R7} However, it should be noted that this increases cost, and as stated in response to comment 2 we are aiming to create device platforms that can be used as disposable units.

The new experiments cover comments 8-10 and are presented in more detail in our response to comment 10 below.

[R3] Hogan, Matthew K., et al. "A wireless spinal stimulation system for ventral activation of the rat cervical spinal cord." *Scientific reports* 11.1 (2021): 14900.

[R4] Malone, Ian G., et al. "Closed-loop, cervical, epidural stimulation elicits respiratory neuroplasticity after spinal cord injury in freely behaving rats." *Eneuro* 9.1 (2022).

[R5] Cedeño, David L., et al. "Spinal evoked compound action potentials in rats with clinically relevant stimulation modalities." *Neuromodulation: Technology at the Neural Interface* 26.1 (2023): 68-77.

[R6] Gonzalez-Rothi, Elisa J., et al. "High-frequency epidural stimulation across the respiratory cycle evokes phrenic short-term potentiation after incomplete cervical spinal cord injury." *Journal of Neurophysiology* 118.4 (2017): 2344-2357.

[R7] Burton, Alex, et al. "Wireless, battery-free, and fully implantable electrical neurostimulation in freely moving rodents." *Microsystems & Nanoengineering* 7.1 (2021): 62.

Comment 9:

"Minimal changes in electrode impedance and minimal degradation of electrode surfaces." (line 308 and 309): Please avoid the use of vague terms such as "minimal". There is an obvious change in electrode impedance, even though stimulation is performed at intensities below maximum. Please quantify the results and present them as percentages.

Our Response: We thank the reviewer for pointing this out and new experiments have been performed and language has been modified in the text to describe outcomes without ambiguity. Specifically, we have performed an extensive accelerated rate analysis of the electrodes used in this study and results are presented in more detail in our response to the next comment.

Modifications to the manuscript (Page 15, Line 391)

Stimulation ~~of a typical clinical FES protocol using constant current mode~~ (1 hour sessions repeated 5 days a week for 16 weeks) using an accelerated lifetime test (9 million pulses at 1 mA at 1000 kHz at room temperature) results in ~~similar degradation~~ (Fig. 3e ~~and Supplementary Fig. 14a,b~~) and ~~minimal~~ compares to results from previous literature ~~degradation of electrode surfaces~~ (Supplementary Fig. 18a,b,c)⁷⁵.

Comment 10:

o Claims regarding stimulation safety: Residual charge is presented for a 1.8 mA stimulation current (line 303), however, the platform is capable of a maximum of 4.7 mA. The residual charge should be calculated for this scenario as well. Reference 49, which is presented as argumentation regarding the limit of allowable residual charge before irreversible damage to either the tissue or electrode, does not present any proof of this (other than very vaguely mentioning a safety limit of 15nC/phase without any further comments or references. If the authors would like to claim stimulation safety they need to do a more rigorous investigation on this topic and back their claims up properly. Residual

charge is not the only, or even not the main, indication of stimulation safety. In fact, looking at the voltage between the anode and cathode of the system during stimulation is a much more informative technique. In particular, the water window should be respected for the electrode polarization potentials, and the interpulse voltage should remain within certain limits. In addition, the charge density and charge per phase are also of importance for safe stimulation and these can be captured to an extent by the Shannon equation, in particular for macroelectrodes. See:

1. S. F. Cogan, K. A. Ludwig, C. G. Welle and P. Takmakov, "Tissue damage thresholds during therapeutic electrical stimulation," *Journal of Neural Engineering*, vol. 13, no. 2, April 2016. 2. Cogan, S. F. Neural stimulation and recording electrodes. *Annu. Rev. Biomed. Eng.* 10,275–309 (2008). 3. K. Kolovou-Kouri, S. Soloukey, F. J. P. M. Huygen, B. S. Harhangi, W. A. Serdijn and V. Giagka, "Dorsal Root Ganglion (DRG) Versatile Stimulator Prototype Developed for Use in Locomotion Recovery Early Clinical Trials," 2021 10th International IEEE/EMBS Conference on Neural Engineering (NER), 2021, pp. 1125-1129, doi: 10.1109/NER49283.2021.9441101.

Our Response: We thank the reviewer for the extensive comment and the additional references. The main goal for this manuscript is to showcase a platform that can operate over many weeks at a time in freely behaving animals and stimulate multiple substrates across a range of currents and voltages. Our device can be used with a range of electrodes and in our experiments we consider two main types for spinal and muscle stimulation. Stimulation safety and electrode longevity are closely tied to electrode geometry and materials and a thorough investigation into stimulation safety across a range of different electrode configurations could be an entire paper in itself. Future experiments using this platform can be performed to evaluate a range of additional electrode configurations in chronic rodent experiments. In fact the platform, because of its monolithic configuration can be rapidly altered to new paradigms or can accept electrode technologies that feature pad style interfaces, which are very common for soft electrodes to match the intended paradigm. So, if higher current injection capabilities, smaller size, higher flexibility etc. is desired this platform will be able to accommodate this.

We would like to point out that while the platform is capable of delivering 4.7mA stimulation this range is mainly useful for the muscle electrodes. We have done literature review to capture typical stimulation currents and the highest noted current in rodents is 1 mA with epidural electrode setups in the cord (see response to comment 8). We therefore chose this current when evaluating the spinal electrodes used in these experiments.

We appreciate the comment that we should provide more information on water window and voltage levels. We have also added experiments for evaluating muscle electrode lifetime performance and added residual charge experiments and compare outcomes with the suggested literature.

We also agree with the reviewer's point that voltage at the electrode interface and resulting electrochemical reactions are most important for determining electrode lifetime and tissue damage. Because in this application we have a very large return electrode that is positioned in the abdominal region and the impedance of the electrodes can vary substantially depending on implantation procedure and animal to animal variations it is hard to estimate chronic testing parameters. The devices we present do not have pulse to pulse voltage measuring capabilities, but only have the ability to check for compliance voltage violations which are provided by the constant current source. To characterize voltage and current applied to the electrodes used for epidural spinal stimulation we therefore set the current to 0.47 mA (determined to be suitable for freely moving subjects in vivo, see chronic experiment description) and gradually reduce the RF power to reduce the available compliance voltage of the device and record the system voltage in vivo at which the device first signals noncompliance. This voltage is a good estimate of the voltage required to produce the stimulation 0.47mA current at the electrode. The resulting voltage of 14.9V at 470 μ A yields a voltage of \sim 1.5 V at the electrode interface. This is obviously just a rough estimation because impedance is difficult to predict in vivo, but provides a good estimate of typical interface voltages.

To evaluate electrode stability more broadly we have performed chronic stability tests at a range of voltages at 1 kHz and peak to peak voltages of 2.4V to 0.4V over 14 days to characterize electrode limits. As suggested by the reviewer we have fixed the voltage for these experiments as this is the driving factor for electrode degradation, using estimated voltages as described above. We would like to point out that voltage at the electrode interface is not easily controllable in vivo in constant current mode which is standard for the field of neural rehabilitation.

For our experimental investigation we are stimulating 24/7 with a biphasic stimulus at 100% duty cycle (50% positive, 50% negative) at 1kHz, this represents 1,209,600,000 stimulation cycles for 14 days of continuous operation. This corresponds to 336 hours of stimulation time, which when paired with approvable protocols (6 h of continuous in vivo stimulation), results in 56 stimulation days. In actual applications with lower stimulation rates and intermittent stimulation periods (e.g. due to wake/sleep or other variations in activity) would easily enable long term studies with this device. This performance clearly demonstrates that our device is very well suited for chronic experiments in small animal models.

Addition made to the supplementary figure (Supplementary Figure 17):

Supplementary Fig. 17. Electrode degradation and changes for chronic accelerated electrode lifetime tests. (a) Spinal electrode degradation after 3 days, 7 days, 11 days and 15 days of biphasic stimulation at ± 0.2 V at 1 kHz and 100% duty cycle **(b)** Muscle electrode degradation after 3 days, 7 days, 11 days and 15 days of biphasic stimulation at ± 0.2 V at 1 kHz and 100% duty cycle **(c)** Spinal electrode degradation after 3 days, 7 days, 11 days and 15 days of biphasic stimulation at ± 1.2 V at 1kHz and 100% duty cycle. **(d)** Muscle electrode degradation after 3 days, 7 days, 11 days and 15 days of stimulation at ± 1.2 V at 1 kHz and 100% duty cycle **(e)** Impedance changes over time for spinal electrodes at voltage within the water window and above. **(f)** Impedance changes over time for muscle electrodes within water window and above.

Modifications to the manuscript (Page 32, Line 774)

A current meter measuring high-side current of the electrodes (LowPowerLab; CurrentRanger) was acquired on an oscilloscope (Siglent; SDS 1202X-E) incorporating a passive 10 MHz low pass filter. The envelope of the current and voltage readings are then used to calculate the impedance of the electrode between 10 Hz to 10k Hz. To

evaluate the chronic stability of spinal and muscle electrodes, a series of tests was conducted with experimental parameters spanning the water window of the gold electrode (± 0.2 V, ± 0.6 V, ± 0.8 V, ± 1.0 V, and ± 1.2 V) which were generated by a function generator (Siglent, SDG 1032X), while monitoring changes in impedance with a digital storage oscilloscope (Siglent, SDS 1202X-E) over a 100Ω shunt resistor at a 1kHz biphasic stimulation with 100% duty cycle (50% positive and 50% negative pulse), performed in a 0.1M phosphate buffer saline solution. Additionally, the spinal electrode was subjected to testing at ± 1.4 V, a voltage exceeding the water window of gold to probe electrode lifetime outside of safe stimulation limits.

Modifications to the manuscript (Page 15, Line 371):

The residual charge during 1.8 mA biphasic stimulation pulses at a frequency of 87 Hz is a maximum of 4.6 nC after 57 stimulation cycles (Supplementary Fig. 136a,b) which is within the limit of allowable residual charge before irreversible damage to either the tissue or electrode⁷⁴.

The device can be used with a variety of electrode materials, impedances, and designs; in our experiments we use both spinal epidural and intramuscular electrodes. Although electrode performance is distinct from the functionality of the device, we evaluate the chronic performance of the electrodes used in our current design by performing accelerated rate testing. We test spinal electrodes both within the water window of the electrode material (gold ($<\pm 1.2$ V)) and above the water window at 1 kHz for 14 days (Supplementary Fig. 17a,c,e). This reflects more than 1 billion stimulation pulses; given a typical FES stimulation rate of 50Hz, this corresponds to more than 10 months of real experimental time. Muscle electrodes, show no change in impedance and no obvious change in electrode appearance for stimulation applied within or above the water window (Supplementary Fig. 17b,d,f). This observation is expected given the large surface area and consequent low current density and material bulk. The smaller spinal electrodes, on the other hand, increase impedance by less than $15k\Omega$ when operated within the water window, enabling chronic operation in small animal models. When spinal electrodes are driven outside of the water window, electrode impedances increased much more substantially (above ± 1.4 V) and dissolution of the gold electrode coating results in electrode failure after 7 days corresponding to 5 months of real experimental time. These results are generally expected for the electrode sizes and materials used here and are independent from the overall performance of the device. Devices described implanted in this study generally remain functional for 4-6 weeks. Importantly, other electrode designs (e.g., platinum vs. gold) that have better chronic performance can be easily implemented with minimal changes in overall design or performance.

Stimulation ~~of a typical clinical FES protocol~~ using constant current mode (1 hour sessions repeated 5 days a week for 16 weeks) using an accelerated lifetime test (9 million pulses at 1 mA at 1000 kHz at room temperature) results in similar degradation (Fig. 3e ~~and~~

~~Supplementary Fig. 14a,b) and minimal compares to results from previous literature degradation of electrode surfaces~~ (Supplementary Fig. 18a,b,c)⁷⁵.

Comment 11:

- Line 125: the authors claim that the use of self-similar serpentine are used to improve device lifetimes: but this is not evidenced in the article.

Our Response: We thank the reviewer for pointing out the unsubstantiated claim. Electrode interconnect (and or connector) failure is a well know failure mode.^{R8,R9} Upon review we see that we did not clearly state this in the manuscript and have corrected this. We have already investigated in figure s17 and s18 the cyclic stability of our mechanical approach. To show the efficacy of this approach we have performed an experiment that shows the impact of cyclic deformation of a simple linear connection as compared to the serpentine used in the device to highlight improvement in terms of mechanical ability.

[R8] Zhang, Yihui, et al. "Mechanics of ultra-stretchable self-similar serpentine interconnects." *Acta Materialia* 61.20 (2013): 7816-7827.

[R9] Oldroyd, Poppy, and George G. Malliaras. "Achieving long-term stability of thin-film electrodes for neurostimulation." *Acta Biomaterialia* 139 (2022): 65-81.

Modifications to the manuscript (Page 18, Line 448)

For the return electrode, the maximum strain is 330% which corresponds to a displacement of 140 mm (Fig. 4d). In both cases, displacements and strains are well beyond those that might occur during implantation ~~of which none of the 21 implantations show signs of electrode damage immediately after implantation (Supplementary Fig. 23). Without the use of strain isolating serpentine interconnects the maximum strain significantly decreases to a maximum strain of 10.7% which corresponds to a displacement of 3.1 mm (Supplementary Fig. 24), demonstrating the robust mechanical properties of using serpentine interconnected within this device.~~

The procedures for implanting the device are similar to those described in previous work on spinal epidural stimulation in the rat.

Addition made to the supplemental figure (Supplementary Figure 24):

Supplementary Fig. 24. Strain stress curve demonstrating the mechanical properties of a linear interconnect.

Modifications to the manuscript (Page 32, Line 789)

To measure failure load of the stretchable interconnects, the device body was mounted on a scale (Mettler Toledo; AB104-S) and the electrode was fixed to a custom 3D printed guide rail for linear displacement. Displacement was measured with a digital caliper while the resulting force was measured from the scale. Electrodes were stretched until there was a loss of electrical conductivity within any copper traces. **The linear interconnect was mounted on a custom stretching stage with a guide rail connected to a stepper motor (Creality, 17HS16-2004S1) to measure linear displacement. Displacement was measured with a digital caliper while the linear interconnect was stretched until mechanical failure.**

Comment 12:

Other feedback/comments/typosReferences 38 and 39 have the same title and details, please check these and correct.

Our Response: We thank the reviewer for pointing this out. We have corrected the details of the citations.

Modifications to the manuscript (Page 11, Line 260)

The device described here ~~uses~~ a larger transmitter antenna (56 cm × 16 cm; 896 cm²) and similar receiver (4 cm × 2 cm; 8 cm²) as compared to receivers used in a previous work⁵⁹ (3.5 cm × 2.5 cm; 8.75 cm²), however this device is capable of harvesting 250 mW using 4 W of RF power in comparison to 50 mW at 4W of RF power in that previous study⁵⁹. This is an increase of more than 500% harvested power over previous designs (5.71 mW cm⁻² to 31.25 mW cm⁻²), enabling effective power harvesting in large arenas³⁹⁵⁹. This performance is enabled by monolithically integrated passive resonator WPT introduced here that ~~Our 3-antenna WPT design~~ boosts power harvesting efficiencies with no increase to device footprint, requiring only an additional copper layer for the resonant antenna.

Modifications to the manuscript (Page 43, Line 1090)

58. Gutruf, P. & Rogers, J. A. Implantable, wireless device platforms for neuroscience research. *Curr. Opin. Neurobiol.* 50, 42–49 (2018).

~~38 Burton, A. et al. Osseosurface electronics – Thin, wireless, battery-free and multimodal musculoskeletal biointerfaces. *Nat. Com.* (2021).~~

~~3959. Cai, L. et al. Osseosurface electronics – Thin, wireless, battery-free and multimodal musculoskeletal biointerfaces. *Nat. Com.* (2021).~~

Comment 13:

Line 88: “targeting multiple substrates” and line 89 “neural substrate”: this terminology, in the context of research involving flexible devices, is confusing. I would strongly advise to replace the term substrates with a more accurate description (body areas? Activatable tissues? – since you are proposing muscle activation and not only neural tissue). Also, please explain why these need such different ranges of voltages and currents (the type of tissue, but also the electrode characteristics that affect this) and add appropriate references (now a paper of the authors’ is cited as evidence for this claim, but the cited paper does not present detailed investigation into this issue, which I feel is not self-evidenced and merits a more thorough documentation and proper reference list).

Our Response:

We thank the reviewer for pointing out this potential confusion. We have replaced substrate with more specific terminology to clarify that this device can stimulate multiple sites across the nervous system.

We appreciate that the different ranges of stimulation currents and voltages has not been sufficiently discussed in the manuscript which we have addressed with modifications to the manuscript. There is a wide range of necessary voltages and currents because activation of neurons by extracellular stimulation depends on the density of cell bodies and axons surrounding the stimulating electrode as well as their distance from the electrode^{R17,18}. For intraspinal stimulation, where the electrode is surrounded by neural tissue in all directions, fibers may be activated by currents as little as 0.5 μA ^{R19} and functional movements produced by currents ranging from 1-15 μA ^{R20}. Unlike intraspinal stimulation, in the case of epidural spinal stimulation the stimulation electrode only delivers a charge to the surface of the spinal cord but is still in contact with an appreciable density of cell fibers. Typical current amplitudes range from 50 – 500 μA using a pulse width of 0.2 milliseconds^{R10}. Lower pulse widths would necessitate even higher amplitudes as was the case in our data (Fig 6d). From our own experiments (see above), we estimated that voltages of ~15V were necessary to produce currents of ~500 μA , similar to what can be derived from previous studies as well^{R15}.

Muscle stimulation produces muscle contraction primarily by activating motor neuron axons since they have a much lower threshold than muscle fibers^{R21,22} and so this form of stimulation still constitutes activation of the nervous system. Because of the diffuse branching of axons within the muscle, there is an even lower density of fibers surrounding the electrode compared to spinal stimulation. This lower density means that even higher amplitudes are needed to produce effective movements. Either pulse width or amplitude can be modulated to achieve functional movements^{R12-14,16}. For very short pulse widths (< 0.1 milliseconds), threshold amplitudes might be as high as 4.5 mA^{R13}. For a 0.1 millisecond pulse width, typical current amplitudes can range from 0.1 to 2.5 mA^{R11}. Due to their increased contact size, impedances of muscle electrodes are smaller than those of spinal electrodes by an order of magnitude (Supp Fig 9). Based on these characteristics, a maximum necessary voltage might be up to 4.5 V (4.5 mA * 1000 ohms).

Although our study is focused on neural stimulation for the restoration of movement, we also note that interventions involving electrical stimulation are not restricted to the nervous system; e.g. electrical stimulation can be used to improve wound healing or to treat cardiac arrhythmias. The device described in our work could be used in these applications as well, suggesting the broad impact of our work.

[R10] Capogrosso, M. et al. Configuration of electrical spinal cord stimulation through real-time processing of gait kinematics. *Nat Protoc* 13, 2031–2061 (2018).

[R11] Capogrosso, M. et al. A Computational Model for Epidural Electrical Stimulation of Spinal Sensorimotor Circuits. *J. Neurosci.* 33, 19326–19340 (2013).

[R12] Jarc, A. M., Berniker, M. & Tresch, M. C. FES Control of Isometric Forces in the Rat Hindlimb Using Many Muscles. *IEEE Transactions on Biomedical Engineering* 60, 1422–1430 (2013).

[R13] Ichihara, K., Venkatasubramanian, G., Abbas, J. J. & Jung, R. Neuromuscular electrical stimulation of the hindlimb muscles for movement therapy in a rodent model. *Journal of Neuroscience Methods* 176, 213–224 (2009).

[R14] Jung, R., Ichihara, K., Venkatasubramanian, G. & Abbas, J. J. Chronic neuromuscular electrical stimulation of paralyzed hindlimbs in a rodent model. *Journal of Neuroscience Methods* 183, 241–254 (2009).

[R15] Wenger, N. et al. Spatiotemporal neuromodulation therapies engaging muscle synergies improve motor control after spinal cord injury. *Nature Medicine* 22, 138–145 (2016).

[R16] Crago, P. E., Peckham, P. H. & Thrope, G. B. Modulation of Muscle Force by Recruitment During Intramuscular Stimulation. *IEEE Transactions on Biomedical Engineering* BME-27, 679–684 (1980). [R17] Ranck, J. B. Which elements are excited in electrical stimulation of mammalian central nervous system: A review. *Brain Research* 98, 417–440 (1975).

[R18] Gustafsson, B. & Jankowska, E. Direct and indirect activation of nerve cells by electrical pulses applied extracellularly. *The Journal of Physiology* 258, 33–61 (1976).

[R19] Jankowska, E. & Roberts, W. J. An electrophysiological demonstration of the axonal projections of single spinal interneurons in the cat. *The Journal of Physiology* 222, 597–622 (1972).

[R20] Tresch, M. C. & Bizzi, E. Responses to spinal microstimulation in the chronically spinalized rat and their relationship to spinal systems activated by low threshold cutaneous stimulation. *Exp Brain Res* 129, 401–416 (1999).

[R21] Peckham, P. H. & Knutson, J. S. Functional Electrical Stimulation for Neuromuscular Applications. *Annual Review of Biomedical Engineering* 7, 327–360 (2005).

[R22] Mortimer, J. Thomas. Motor prostheses. in *Handbook of Physiology* 155–187.

Modifications to the manuscript (Page 15, Line 391)

Stimulation ~~of a typical clinical FES protocol using constant current mode~~ (1 hour sessions repeated 5 days a week for 16 weeks) using an accelerated lifetime test (9 million pulses at 1 mA at 1000 kHz at room temperature) results in ~~similar degradation~~ (Fig. 3e ~~and Supplementary Fig. 14a,b~~) and ~~minimal~~ compares to results from previous literature ~~degradation of electrode surfaces~~ (Supplementary Fig. 18a,b,c)⁷⁵.

Modifications to the manuscript (Page 4, Line 80)

Finally, many functional electrical stimulation applications involve complex stimulation protocols, delivered across multiple ~~substrates~~ **locations in the central and peripheral nervous system** in precise temporal relationships and with highly regulated stimulation levels. Current battery-free systems typically use passive, unregulated voltage control²¹ or relatively simple voltage programming^{16,22,23}. Existing devices capable of delivering controlled currents with precise and flexible programming require application-specific integrated circuits (ASICs) that are expensive and do not provide a general platform that can be used across a wide range of applications²⁴. ~~Existing devices also typically do not involve the dual power supplies required for biphasic, charge-balanced stimulation protocols that are necessary to minimize tissue and electrode damage from chronic stimulation.~~

Additional challenges arise in applications targeting ~~multiple substrates~~ **areas** that require very different currents or voltages; depending on the ~~density of cell bodies and axons surrounding a stimulation electrode, the distance between cells and electrode as well as neural substrate and~~ **electrical interface**, required currents can ~~widely vary~~^{25,26} ~~range from nA to mA~~. For example, in the case of intraspinal stimulation, where the electrode is situated near many neurons and axons, functional currents can range from 0.5 μA ²⁷ to 15 μA ²⁸. Epidural spinal stimulation electrodes, which contact fewer cells than intraspinal stimulation but still contact a relatively high density of fibers, use current amplitudes typically ranging from 50 to 500 μA ²⁹⁻³¹. Muscle stimulation electrodes, which may contact and recruit even fewer highly branched axons, require currents up to 4.5 mA in rats³²⁻³⁴. Furthermore, ~~electrode impedances and their current amplitudes may require~~ **and** voltages from a few mV to tens of volts^{16,28,31}. Devices are typically capable of delivering either low or high ranges of currents/voltages, but not both ranges simultaneously ~~that allow for simultaneous electrical stimulation of multiple neuromuscular sites and for multichannel abilities.~~

Here, we introduce fully implanted devices that overcome these challenges, providing a voltage range of up to $\pm 20\text{V}$, current controlled stimulation with a range of 40 μA to 4.7 mA and independent control of 8 channels in a fully implantable package that is powered at distance. This device represents the highest figure of merit for all categories (Supplementary Table 1)^{23,24,35-41}, making ~~them~~ **it** suitable for long term, uninterrupted functional electrical applications ~~for preclinical investigations in freely moving small to medium animal subjects~~. The devices exploit a passive resonator scheme that substantially increases power harvesting capabilities at a designed voltage, making it possible to stimulate ~~a wide range of biological substrates~~ **multiple areas of the nervous system** simultaneously, using the same implanted device.

Comment 14:

Line 99: replace “flexible” with “versatile” (flexible can be confusing in the context of flexible electronics)

Our Response: We thank the reviewer for this comment, and we agree and have changed this in the manuscript accordingly.

Modification made to the manuscript (page 4, line 85):

Current battery-free systems typically use passive, unregulated voltage control or relatively simple voltage programming^{16,22,23}. Existing devices capable of delivering controlled currents with precise and ~~versatileflexible~~ programming require application-specific integrated circuits (ASICs) that are expensive and do not provide a general platform that can be used across a wide range of applications. ~~Existing devices also typically do not involve the dual power supplies required for biphasic, charge-balanced stimulation protocols that are necessary to minimize tissue and electrode damage from chronic stimulation.~~

Modification made to the manuscript (page 5, line 118):

As a consequence, ~~they~~the devices have minimal impact on overall animal behavior and can remain operational for extended periods of at least 6 weeks. Further, the devices are able to deliver highly ~~versatileflexible~~, precisely controlled, and charge-balanced stimulation ~~and are designed for scalable manufacturing that enables easily deployable one-time use systems to deliver long-term stimulation.~~ Taken together, these advances in device performance enable the development and evaluation of a range of complex chronic stimulation protocols for functional restoration that have previously been unattainable.

Comment 15:

Line 123: I feel the term “self-similar” needs to be properly explained

Our Response: We thank the reviewer for pointing this out. We agree that self-similar needs to be clearly described and have added more details of its structure and citations of technologies using self-similar structures.

Modification made to the manuscript (page 6, line 149):

In order to maximize the mechanical flexibility of the device and enable chronic stability, the integrated circuits (IC) in the device are placed in separate rigid islands and interconnected with strain-isolating serpentine traces^{47,48} (Fig. 1b; Supplementary Fig. 1; see detailed component description in Supplementary Fig. 42a,b). **Self-similar serpentines utilize a fractal design with repetitive patterning of interconnected curvatures that evenly distribute internal strain within the flexible substrate. We utilize this patterning structure**~~Self-similar serpentines are used~~ to connect the device body to the spinal electrodes and the return electrode, to improve strain isolation, to facilitate easy implantation of the electrodes and to improve device lifetimes^{14,18}.

Comment 16:

Figure 1, a: macroelectrodes vs microelectrodes: please add sizes (electrode area), and references for these ranges claimed here. References should be articles that show neurostimulation studies in the areas mentioned, and with documented electrode sizes. Also, I feel the labels are not correct, microelectrodes are probably used for neurons and brain applications, vs macroelectrodes for the heart and muscle (which is the case from what I see further down in the article, lines 268-272).

Our Response: We thank the reviewer for pointing this out. We agree with this definition and have updated the figure and the corresponding text.

Modification made to the manuscript (page 6, line 146):

Figure 1a displays a rendering of a device implanted subcutaneously in the posterior lumbar region of a rat. The device has the capability for both spinal and muscular functional electrical stimulation (FES) using an eight-channel electrode array positioned over the spinal cord and two intramuscular electrodes in the biceps femoris posterior (Fig. 1a). **This device can accommodate electrode designs for μA to mA of stimulation using both micro-electrodes ($<200\ \mu\text{m}$) for precise stimulation of a group of neurons and macro-electrodes ($>200\ \mu\text{m}$) for intramuscular stimulation. This multimodal stimulation capability enables a wide range of strategies for using electrical stimulation for the restoration or rehabilitation of motor function.** We use a monopolar configuration with a common return electrode placed in the thorax, as is common in rehabilitation applications.

Modification made to main figure (Figure 1):

Fig. 1. Wireless battery-free functional electrical stimulation overview. (a) Rendering of implant in the small animal subject with rendered insets of electrode placement in the spinal cord and muscle. **(b)** Photograph image of the device. **(c)** Photograph of the animal running on a treadmill. **(d)** Block diagram of electrical functionalities of the device.

Comment 17:

Fig. 3: Impedance spectroscopy data should be presented as impedance magnitude, and phase (Bode plots). Please add these and correct the “resistance” labels to “impedance magnitude” in Fig. 3e, Supplementary Fig. 9, and Supplementary Fig. 14 b and c.

Fig. 4: b: This is not a Bode plot, as claimed in the caption, please correct this. Also, please add information in the figure about the frequency at which this impedance is measured.

Our Response: We thank the reviewer for these corrections. We have made changes to both the figures and figure texts.

Modification made to main figure (Figure 3)

Fig. 3. Wireless communication, electrode design and characterization (a) Block diagram of communication scheme. **(b)** Photograph of return, spinal and muscle electrodes **(c)** Temporal control of stimulation patterns. **(d)** Amplitude control of biphasic stimulation. **(e)** Change in the impedance of the electrode, measured at 1 kHz, after continuous stimulation of over 9 million cycles.

Modification made to supplementary figure (Supplementary Figure 11)

Supplementary Fig. 911. Electrode impedance magnitude in reference to the return electrode. (a-b) Impedance magnitude of the spinal electrode in a x1 saline solution in both log and linear graph. **(c-d)** Impedance magnitude of the stainless-steel muscle electrode x1 saline solution in both log and linear graph.

Modification made to supplementary figure (Supplementary Figure 18)

Supplementary Fig. 18. Continuous 1 mA stimulation in saline. (a) Photograph image of the spinal electrode and muscle electrode stimulating in saline over four weeks. (b) Impedance sweep of the spinal electrode over multiple weeks. (c) Impedance sweep of the muscle electrode over multiple weeks.

Modification made to main figure (Figure 4)

Fig. 4. Material overview and mechanical characterization of the device. (a) Exploded view rendering of each material layer used in the device, the return electrode, and the spinal electrode. (b) **Bode plot of electrode impedance** Change in impedance of the electrode measured at 1kHz. (c) Strain stress curve of the serpentine structure for the spinal electrode, and (d) the return electrode.

Comment 18:

Line 361: "displacements and strains are well beyond those that might occur during implantation": please support this claim with values and references.

Our Response: We thank the reviewer for pointing this out. We have now added a reference to our documentation of surgeries.

Modifications to the manuscript (Page 18, Line 448):

For the return electrode, the maximum strain is 330% which corresponds to a displacement of 140 mm (Fig. 4d). In both cases, displacements and strains are well beyond those that might occur during implantation of which none of the 21 implantations show signs of electrode damage immediately after implantation (Supplementary Fig. 23). Without the use of strain isolating serpentine interconnects the maximum strain significantly decreases to a maximum strain of 10.7% which corresponds to a displacement of 3.1 mm (Supplementary Fig. 24), demonstrating the robust mechanical properties of this device.

Modifications to the manuscript (Page 33, Line 801):

The Young's modulus (E) and Poisson's ratio (ν) are $E_{\text{Polyimide}} = 4 \text{ GPa}$, $\nu = 0.34$; $E_{\text{Copper}} = 121 \text{ GPa}$, $\nu_{\text{Copper}} = 0.34$. Motion was simulated by fixing the base of the serpentine interconnects with a fixed support and the applied deformation was added to the tip of the electrodes with a pivot joint. Device deformation while implanted in the animal was analyzed using ImageJ to calculate deformation in the device in X-ray images of the rat post-implantation.

Comment 19:

Line 86: "Existing devices also typically do not involve the dual power supplies required for biphasic, charge balanced stimulation protocols that are necessary to minimize tissue and electrode damage from chronic stimulation." This statement is untrue. Dual power supplies are NOT required for charge balanced stimulation protocols. There are other ways to implement charge balancing (which fundamentally depends on the remaining charge across the electrode-tissue interface and cannot be predicted with charged balanced waveforms.)

Our Response: Thank you for this comment. We have clarified requirements for charge balancing and compared balancing methods between this and previously published devices. We have chosen this approach for biphasic stimulation because it requires the smallest footprint and least components, which is core to the device design philosophy outlined in response to comment 2. We have added additional discussion to the manuscript to highlight this.

Modifications made to the manuscript (page 4, line 85):

Existing devices capable of delivering controlled currents with precise and ~~versatile~~~~flexible~~ programming require application-specific integrated circuits (ASICs) that are expensive and do not provide a general platform that can be used across a wide range of applications²⁴. ~~Existing devices also typically do not involve the dual power supplies required for biphasic, charge-balanced stimulation protocols that are necessary to minimize tissue and electrode damage from chronic stimulation.~~ Additional challenges arise in applications targeting multiple ~~substrates~~~~areas~~ that require very different currents or voltages; depending on the ~~density of cell bodies and axons surrounding a stimulation electrode, the distance between cells and electrode as well as neural substrate and~~ electrical interface, required currents can widely vary^{25,26}.

Modifications made to the manuscript (page 5, line 111):

The devices exploit a passive resonator scheme that substantially increases power harvesting capabilities at a designed voltage, making it possible to stimulate ~~a wide range of biological substrates~~~~multiple areas of the nervous system~~ simultaneously, using the same implanted device. ~~The circuit design incorporates off-the-self components providing scalable manufacturing at a cost that enables single use deployment which is critical for chronic implantation that make reuse impossible or difficult.~~ The devices are fully implanted, are powered and controlled wirelessly, and are fabricated from flexible electronics.

Modifications made to the manuscript (page 7, line 173):

The dual voltage supply is achieved using a center-tapped antenna design with two single half-bridge rectifiers to create the positive and negative voltages with no external components and only minimally increased device footprint. ~~The receiver antenna uses a center tapped antenna to minimize circuit complexity while allowing for dual voltage supplies. A dual supply coupled with a precision current driver allows for a smaller circuit design (30 mm²) suitable for implantation while providing higher power efficiencies compared to designs using DC/DC converters to generate dual voltage supplies⁴⁹. This is significantly smaller than traditional systems using a single supply to have the same biphasic stimulation response using a Howland current pump, H-bridge, and current monitor each requiring additional electrical traces and passive components (>50 mm²)⁵⁰⁻⁵².~~ A power management system incorporates voltage protection using Zener diodes and a step-down converter accompanied by a linear voltage regulator for stable operating voltages for the digital circuit.

Comment 20:

Also, line 299: “Biphasic pulses ensure charge balancing, to avoid charge accumulation and irreversible electrochemical processes at the electrode surface which can result in tissue damage or electrode degradation”: it is common from literature that biphasic pulses do not, in fact, ensure charge balancing, see, among others, e.g.: 10.1146/annurev.bioeng.10.061807.160518 and N.d.N. Donaldson, P.E.K. Donaldson, “When are actively balanced biphasic (‘Lilly’) stimulating pulses necessary in a neurological prosthesis? I Historical background; Pt resting potential; Q studies,” Med. Biol. Eng. Comput., vol. 24, pp. 41–49, January 1986. Please correct this statement.

Our Response: We thank the reviewer for pointing out that biphasic stimulation doesn't always solve imbalanced charge accumulation. We have added additional details here to clarify.

Modification made to the manuscript (page 14, line 360):

An important aspect of chronic stimulation in clinical applications is the need for biphasic stimulation capabilities with the option to deliver anodic or cathodic leading pulses⁶³. Biphasic pulses ~~ensure~~allows for charge balancing, ~~designed to avoid~~reduce charge accumulation and irreversible electrochemical processes at the electrode surface which can result in tissue damage or electrode degradation^{71,72}. ~~Charge balancing can also be achieved using a unidirectional current and a series capacitor⁷³. However, this method requires a low output resistance and a correctly matched capacitor that depends on the electrode design. This is why in the devices described here, biphasic stimulation was used for active charge balancing which enables versatile use with custom electrodes.~~ This process is evaluated ex vivo by recording direct current through the stimulation electrode during chronic stimulation in saline solution and integrating current over time.

Comment 21:

Supplementary Fig. 11: please elaborate on what this figure demonstrates

Our Response: This figure represents the programmable control over stimulation pulse widths for both the anodic and cathodic phase. We have added more details to the figure caption to make this figure clearer.

Modification made to the supplementary figure (Supplementary Figure 13):

Supplementary Fig. 143. Wirelessly programmable control of biphasic stimulation pulse width duration with a fixed 40 μs interphase gap between anodic and cathodic phase. ~~Time response of biphasic stimulation with pre-programmed pulse widths.~~

Comment 22:

Supplementary Fig. 12b: how should we interpret the impedance values? Were the output stages of the electronics connected to resistors of this nominal value? If so, why is the voltage waveform for e.g. C6 not a rectangular one? Normally electrodes are approximated by capacitors and resistors (in series or parallel) but this does not seem to have been the case here? If so, please mention the values of the capacitors used. In any case, please justify the selected load with respect to what is expected from an application perspective (e.g. why a load of 16.5 kΩ?) Also, in line 290, a maximum current error or +/- 8 μA is mentioned, how is this calculated/measured?

Our Response: Supplementary Fig. 12b displays the capability of the device to deliver current controlled stimulation of 4.7 mA and 1 mA over a range of physiologically relevant impedances for the muscle and spinal applications using pulse durations between 100 μs and 1000 μs. This translates to the impedance values in saline which are between 1 kΩ to 16.5 kΩ.

The output of the current driver has 15 ohm resistors in series and 100 nF capacitors in parallel that are used to help protect the current driver from over voltage spiking during stimulation. We have added a circuit diagram of the device for better clarification of the stimulation circuit and to improve repeatability of this paper.

The current error was calculated based on the datasheet of both the DAC and current driver. An additional experiment verified these calculations.

Modification made to the manuscript (page 14, line 343):

Supplementary Fig. 13 and Supplementary Fig. 124b shows an example in which each stimulation channel for both the muscle and spinal electrodes is wired to resistive loads (1 k Ω and 16.5 k Ω) that are similar to typical electrode impedance magnitudes, then tested in vitro with stimulation pulse durations between 100-1000 μ s which are similar to typical durations used in FES applications (200 μ s to 600 μ s)⁶⁷⁻⁶⁹. ~~with a wide range of impedances are connected to the eight channels and stimulation pulses with different parameters are applied sequentially to each channel.~~ In this example, the maximum current error is only $\pm 812 \mu$ A (7 μ A is due to the voltage error of the DAC and 5 μ A due to the current driver) for a set current of 1mA. This matches experimental data showing maximum current error of $\pm 12 \mu$ A (Supplementary Fig. 15). This demonstrates the robustness of the analog front end. To deliver the programmed stimulation currents, especially in the case of high current applied through high impedance electrodes (Supplementary Fig. 124e), the device must be capable of operating with a high voltage compliance.

Addition to the supplementary figure (Supplementary Figure 15)

Supplementary Fig. 15. Measured current error over multiple stimulation cycles with a 10 k Ω load.

Modification made to the supplementary figure (Supplementary Figure 2)

b

R0	0 Ω	R6	10 k Ω	C1	2.2 μ F	U4	XTR300 (Current Mirror)
R1	150 k Ω	R7	20 k Ω	C2	200 pF	U5	ADGS5414 (MUX)
R2	47 k Ω	R8	1 M Ω	U0	MAXM17532 (Step Down)		
R3	4.7 k Ω	R9	1 k Ω	U1	NCP3335ADM330 (LDO)		
R4	300 Ω	R10	2 k Ω	U2	Attiny 84A (MCU)		
R5	150 k Ω	C0	44 pF	U3	DAC5311 (DAC)		

Supplementary Fig. 2. Circuit Diagram of the implantable device. (a) Schematic. (b) List of the main electrical components on device.

Comment 23:

Fig. 12e: what is M1 and M2? Please elaborate on what each part of this figure demonstrates.

Our Response: Thank you for pointing this out. While M1 and M2 have been defined in the main figures they have not been explicitly defined here. We have made changes to clarify accordingly.

Modification made to the supplementary figure (Supplementary Figure 14)

Supplementary Fig. 124. Current controlled biphasic stimulation. (a) Programmable stimulation amplitude for a maximum biphasic current of 1.8 mA. (b) Individualized current control over all stimulation channels. (c) Voltage compliance with respect to cage power while stimulating at 4.7 mA in saline. (d) Voltage compliance within the treadmill cage arena (56 cm x 16 cm) with a 5 W RF power. (e) Recorded current through various electrical loads from 1 k Ω to 16.5 k Ω for muscle electrodes labeled M1 and M2 and spinal electrodes labeled S1 through S6.

Comment 24:

- Is there enough detail provided in the methods for the work to be reproduced? The methods described here go to a depth and detail level which is common in published articles. I have my reservations regarding the extent these allow work to be reproduced, but this is not the authors' fault, they have followed common practice.

Our Response: Thank you for reviewing the methods section. We have documented all new experiments as pointed out to the standard of the field. We would even argue that we go beyond what is published in other articles, for example, publishing the outcomes of all animal experiments performed.

Reviewer #2 (Remarks to the Author):

Comment 1:

In this manuscript, Gutruf and colleagues demonstrated an implantable transient power delivery system based on a passive resonator approach coupled with active power electronics to convert alternating current to DC voltage. This approach was aimed to be suitable for powering embedded systems such as microcontrollers to create a customizable in vivo electrical stimulation. They demonstrated the capacity of this approach by developing in vivo rodent experiments that involved delivery of electrical stimulation to peripheral nervous system structures. While the manuscript is interesting, there are several issues that require the authors' attention.

Our response: We thank the reviewer for their very detailed comments that provide a foundation for us to revise our manuscript and clarify concerns.

Comment 2:

- Authors provide a nice introduction about the current challenges in powering implantable devices. Specifically, for transiently powered, coiled-based systems “the amount of power ... depends on the inductance ratio and coupling between transmitter and receiver.” However, the authors do not provide a clear statement as to what are the key advances in their approach. How does their design overcome these limitations or improve the current the efficiency of power delivery?

Our Response: In the devices we present here we co-locate the resonator with the harvesting antennas, thereby improving harvesting efficiency. This design also enables us to tune the maximum power point to the desired operation voltage while still locating all components on one dual sided flexible substrate and using minimal passive components. This is a new concept and enables high power availability in large experimental enclosures without the need of locating powering coils directly on the subject. These are the requirements to test chronic stimulation for FES in rodent subjects and we could not find any device in literature that accomplishes this.

We also point out that the device is built on scalable flexible electronic circuits that enable broad distribution to the scientific community. We provide this level of functionality without the use of ASICs which makes scalability and price suitable for one time use in subjects, thereby facilitating studies with timescales and statistical powers that can show efficacy of FES on chronic timescales. We anticipate that these features of our devices enable a much more streamlined animal experiment pipeline than is currently available.

We have added several changes to the manuscript that address these points. Please see also the response to reviewer 1 comment 2 ,5, and 6 for detailed modifications.

Comment 3:

- line 219: Authors demonstrate 500% improvement in power delivery compared to the previously reported experiments. What design element was responsible for this improvement?

Our Response: We thank the reviewer for highlighting this point. We have clarified where this 500% improvement originates from and what elements are included for this improvement. We have also elaborated on the novelty of the approach in response to reviewer 1 comment 2 which is also substantiated in changes to the manuscript.

Modifications made to the manuscript (page 11, line 260):

The device described here using larger transmitter antenna (56 cm × 16 cm; 896 cm²) and similar receiver (4 cm × 2 cm; 8 cm²) as compared to receivers used in a previous work⁵⁹ (3.5 cm × 2.5 cm; 8.75 cm²), however this device is capable of harvesting 250 mW using 4 W of RF power in comparison to 50 mW at 4W of RF power in that previous study⁵⁹. This is an increase of more than 500% harvested power over previous designs (5.71 mW cm⁻² to 31.25 mW cm⁻²), enabling effective power harvesting in large arenas^{39,59}. This performance is enabled by monolithically integrated passive resonator WPT introduced here that ~~Our 3-antenna WPT design~~ boosts power harvesting efficiencies with no increase to device footprint, requiring only an additional copper layer for the resonant antenna. This extraordinary harvesting capability allows for more sophisticated behavioral paradigms in complex 3-dimensional behavioral arenas, such as elevated, or even submerged, mazes^{8,29}.

Comment 4:

- Authors claim the center-tapped antenna enabled a dual voltage supply system. It would be important for the authors to provide justification for their approach given its added complexity compared to power rail splitting where the ground is assigned to a middle potential.

Our Response: This is a great comment. We chose the approach described in our manuscript because of its higher efficiency and the fact that it can be realized with fewer components. We have made modifications to the manuscript in response to reviewer 1's comment 19 which raised a similar question. We have also included a comparison of our approach to other approaches used to generate a dual rail supply.

Comment 5:

- Authors provide a simple validation of the device's ability to deliver electrical stimulation. It would be important to demonstrate the specific use and advantages of having a multi-channel electrical stimulation.

Our Response: We showcase the need for multichannel stimulation in fig 6 h where 2 stimulation sites enable a differentiated electrophysiological response. We consider this a very specific use case that has not been demonstrated before.

We have multiple spinal channels which span several segments of the spinal cord. This can induce different motor responses depending on the stimulation site and provide the capability of specific control of movements. For example, rostral channels will produce more flexion and caudal channels for extension, and we have access to movements of both left and right side. Also, including muscle channels together with the spinal channels is a novelty of this device as muscle stimulation will provide more specific motor responses as compared to spinal stimulation, potentially enabling novel rehabilitation strategies.

Modifications made to the manuscript (page 6, line 140):

Figure 1a displays a rendering of a device implanted subcutaneously in the posterior lumbar region of a rat. The device has the capability for both spinal and muscular functional electrical stimulation (FES) using an eight-channel electrode array positioned over the spinal cord and two intramuscular electrodes in the biceps femoris posterior (Fig. 1a). **This device can accommodate electrode designs for μA to mA of stimulation using both micro-electrodes ($<200\ \mu\text{m}$) for precise stimulation of a group of neurons and macro-electrodes ($>200\ \mu\text{m}$) for intramuscular stimulation. This multimodal stimulation capability enables a wide range of strategies for using electrical stimulation for the restoration or rehabilitation of motor function.**

Comment 6:

- Why do the authors solely reply on experiments where the rodent is fixated? One could argue that the choice of such an experimental set-up does not demonstrate the capacity of such a system; wired implants could easily function in this set-up.

Our Response: We thank the reviewer for this comment and in response to this comment and comments of reviewer 1 we have added substantial new experiments that demonstrate the use of the device in freely behaving subjects over chronic time periods that far exceed what is possible with wired and even battery powered devices. Changes are documented in response to reviewer 1 comment 4 and 5. For convenience of review, we have also included the key figure showing these new experiments below.

Figure 7. Stimulation-evoked movements in spinal cord transected animals. (a) An awake, spinal cord transected (T10) animal is stimulated. (b) A limb propulsion movement is evoked from stimulation of a spinal electrode site (c3) while the animal is locomoting on a treadmill with unparalyzed forelimbs. (c) Stimulation of spinal electrode site (c0) evoked limb propulsion and retraction. (d) Stimulation in cage with freely moving spinal cord injured animals. Middle plot illustrates the average voltage compliance at each region of the cage. Right plot illustrates movement trajectories (blue) measured over 2 hours. Green marks indicate points at which stimulation is compliant; red marks indicate uncompliant stimulation. (e) Freely behaving animal on a moving treadmill (conventions follow (d)). (f) Voltage compliances on individual trials over two 6-hour sessions and sessions recorded 25 and 28 days following device implantation. (g) Distributions of voltage compliances measured in three different enclosures of increasing size: large cage (29X36cm), treadmill (16X56cm), small cage (18X36cm).

Comment 7:

- What new experimental capabilities could this technology enable? How do the authors aim to translate this technology? These should be added in the discussion section.

Our Response: Thank you for the excellent comment. There are several new capabilities enabled by this technology. First, as mentioned in previous comments, our overall design and component choice has been made to enable the technology to be readily adopted by laboratories in the neuroscience and electrical therapy communities. The fabrication and materials scheme enables rapid panelization and automated assembly, allowing the devices to be fabricated at scale and at reasonable costs. Additionally, we utilize 13.56 MHz RF power supplies that already exist in many researchers' labs. We have previously scaled optogenetic stimulation devices that are now available on the market only 2 years after publication and at a cost that is accessible to all neuroscience laboratories which can be found at <http://www.neurolux.org> (not only elite labs with large budget).⁹ The fabrication process is based on panelized flex PCB designs that are commercially available and only require laser depaneling and SMD component assembly. This is a scalable process that can be if needed outsourced to produce devices at scale and with high yields. We have included a new supplementary figure that shows the process and we have added new discussion the conclusion section.

Second, we have included a brief discussion of the new applications that this technology can enable in the Discussion. Our device will enable evaluation of rehabilitation strategies in freely behaving animals over extended periods of time. This is in contrast to the typical approaches in which FES interventions are performed for a brief period each day and under severely constrained experimental conditions. Further, this device enables precisely timed stimulation profiles between different stimulation sites, enabling a wide range of stimulation of protocols such as those involved in spike time dependent plasticity. Finally, this device can enable interventions involving stimulation of multiple sites in the nervous system, including the brain, spinal cord, peripheral nerves, muscles in addition to any other site that might be targeted by electrical stimulation.

Additions made to the Supplementary Figure (Supplementary figure 35)

Supplementary Fig. 35 Implant panel top view. (a) Implant in panelized form with alignment markers **(b)** Implant depanelized with laser ablation process. **(c)** Device assembled with SMD components.

Modification made to the manuscript (page 28, line 648):

Characteristics of the devices demonstrated here make them suitable candidates for automated investigation of rehabilitation of neural circuits on clinically relevant timescales and with adequate statistical power to investigate underlying mechanisms that currently prevent widespread clinical adoption. This platform is designed using established techniques for flexible printed circuit board manufacturing, SMD population and laser depaneling as shown in Supplementary Fig. 35. The combination of the device capabilities and manufacturing process has been optimized to be used as an investigational tool by the neuroscience community to study neural circuit rehabilitation in freely moving subjects at a cost that enables single device use for chronic rodent studies.

Reviewer #3 (Remarks to the Author):

Comment 1:

This manuscript proposes a novel passive resonator optimized power transfer design, as an alternative to conventional battery packs for the application of electrical stimulation for assistance or rehabilitation. The motivation for higher power wireless implanted systems is well stated and logically arranged. The submission includes detailed design information, methodology and overall specifications. Evaluation is conducted including extensive testing on rats with SCI. Results also include strain tests and long term testing. The vivo tests are comprehensive, showing stimulation parameters and resulting kinematics. Overall, the system is a significant advancement in current technology. The technical material is rigorous and sufficiently detailed, however more detail could be added to place the device in the broader context of its eventual application area:

Our Response: We thank the reviewer for the detailed comments and we have added a discussion that highlights the need for single use chronic FES for small animal subjects at a price suitable to facilitate studies with timescales and statistical powers that can show efficacy of FES on chronic timescales to investigate mechanisms as well as provide solid evidence before moving on to large animal validations. We anticipate that this enables a much more streamlined animal experiment pipeline than is currently available.

Comment 2:

1) Current battery-free systems [refs 16,22,23] are mentioned (page 4) however it would be useful to provide more information on these devices --- e.g. a brief summary of their intended application area, whether tested with human subjects in human trials, and perhaps a comparison table of main specifications. It may be useful to also mention implanted FES stimulators that contain a battery, but are recharged wirelessly, e.g. BION stimulators [1]. This would assist the reader in understanding the wider context within human assistance/rehab.

[1] Schulman JH, Mobley JP, Wolfe J, Regev E, Perron CY, Ananth R, Matei E, Glukhovskiy A, Davis R. Battery powered BION FES network. Conf Proc IEEE Eng Med Biol Soc. 2004;2004:4283-6. doi: 10.1109/IEMBS.2004.1404193. PMID: 17271251.

Our Response: Thank you for the excellent comment, a detailed table with key power metrics has been provided in response to reviewer 1 comment 2. We have modified the manuscript to better highlight our system in comparison with other works of implantable FES devices. This device platform however is for investigating chronic stimulation

schemes in small to medium sized animals and is not for the direct translation towards human application. In existing literature, our devices provide a 5% improvement on operational lifetime, 17% improvement on stimulation range and more than 500% power harvesting for wirelessly powered device. For review convenience, we show the key changes to the manuscript below.

Modifications to the manuscript (page 5, line 101):

Devices are typically capable of delivering either low or high ranges of currents/voltages, but not both ranges simultaneously **that allow for simultaneous electrical stimulation of multiple neuromuscular sites and multichannel abilities.**

Here, we introduce fully implanted devices that overcome these challenges, **providing a voltage range of up to ± 20 V, current controlled stimulation with a range of 40 μ A to 4.7 mA and 8 channel count in a fully implantable package that is powered at distance. This device represents the highest figure of merit for all categories (Supplementary Table 1)^{23,24,35-41}, making ~~them~~ it suitable for long term, uninterrupted functional electrical applications for preclinical investigations in freely moving small to medium animal subjects.**

Modifications to the manuscript (page 5, line 111):

The devices exploit a passive resonator scheme that substantially increases power harvesting capabilities at a designed voltage, making it possible to stimulate ~~a wide range of biological substrates~~ **multiple areas of the nervous system** simultaneously, using the same implanted device. ~~The circuit design incorporates off-the-self components providing scalable manufacturing at a cost that enables single use deployment which is critical for chronic implantation that make reuse impossible or difficult.~~ The devices are fully implanted, are powered and controlled wirelessly, and are fabricated from flexible electronics. ~~The materials and mechanicals design of the flexible substrate provides a platform that integrates components to conform with the surrounding soft tissue using a network of ridged islands and flexible interconnects.~~ As a consequence, ~~they~~ **the devices** have minimal impact on overall animal behavior and can remain operational for extended periods of at least 6 weeks.

Modifications made to the manuscript (page 7, line 173):

The dual voltage supply is achieved using a center-tapped antenna design with two single half-bridge rectifiers to create the positive and negative voltages with no external components and only minimally increased device footprint. ~~The receiver antenna uses a center tapped antenna to minimize circuit complexity while allowing for dual voltage supplies.~~ A dual supply coupled with a precision current driver allows for a smaller circuit design (30 mm²) suitable for implantation while providing higher power efficiencies

compared to designs using DC/DC converters to generate dual voltage supplies⁴⁹. This is significantly smaller than traditional systems using a single supply to have the same biphasic stimulation response using a Howland current pump, H-bridge, and current monitor each requiring additional electrical traces and passive components (>50 mm²)⁵⁰⁻⁵². A power management system incorporates voltage protection using Zener diodes and a step-down converter accompanied by a linear voltage regulator for stable operating voltages for the digital circuit.

Modifications to the manuscript (page 9, line 211):

In order to overcome these challenges, we introduce a design that imbeds a passive resonator directly on the implanted device, improving power transfer efficiency and enabling high voltage compliances. **The monolithic integration removes the need for auxiliary circuit components such as resonators that are located elsewhere in the tissue and enables a simple to implant platform that offers a scalable modulation platform for chronic applications.** This design also requires only a small number of passive components, resulting in small device footprints and high efficiencies compared to active approaches such as maximum powerpoint tracking and management IC`s.

Modifications to the manuscript (page 5, line 120):

Further, the devices are able to deliver highly **versatileflexible**, precisely controlled, and charge-balanced stimulation and **are designed for scalable manufacturing that enables easily deployable one-time use systems to deliver long-term stimulation.** Taken together, these advances in device performance enable the development and evaluation of a range of complex chronic stimulation protocols for functional restoration that have previously been unattainable.

Comment 3:

2) The clinical application to rats is detailed and comprehensive. However more information on how the technology could transfer to human users would be useful (perhaps s short paragraph in teh discussion). In this context, is the voltage range, stimulation frequency and pulsewidth sufficient to elicit human muscle contracts? Would the cage antenna system be suitable to extend to human participants, perhaps in their own homes.

Our Response: We thank the reviewer for their comments and added a section in the conclusion to discuss the translation of this technology to humans. We have also provided a new proof of principle experiment that shows that we can operate the device with a simple wearable sized coil that is operated at low power demonstrating the potential ability to use the device in a human or large animal context.

However, we note that there are still several challenges that need to be overcome for human use. Specifically, one major challenge is designing electrodes that provide stability

for decade long stimulation while still retaining the advantage of soft mechanical properties. These requirements inherently mean thin metal films, a system level encapsulation that is thin and lasts decades and a wearable system that is imperceptible and easily worn continuously.

Modifications made to the manuscript (page 28, line 648):

Characteristics of the devices demonstrated here make them suitable candidates for automated investigation of rehabilitation of neural circuits on clinically relevant timescales and with adequate statistical power to investigate underlying mechanisms that currently prevent widespread clinical adoption. This platform is designed using established techniques for flexible printed circuit board manufacturing, SMD population and laser de-paneling as shown in Supplementary Fig. 35. The combination of the device capabilities and manufacturing process has been optimized to be used as an investigational tool by the neuroscience community to study neural circuit rehabilitation in freely moving subjects at a cost that enables single device use for chronic rodent studies. When translating this technology for use in large animals, magnetic resonant coupling starts to become inefficient due to the operational distance between the receiver and transmitting coil exceeding 1 m. To solve this issue, implanted devices can be combined with wearable systems to enable directed power delivery and communication, which is possible with power consumptions suitable for battery operation as demonstrated in Supplementary Fig. 36 where a small NFC reader chip with 150mW RF power yields 76mW of harvested power at the implant. This approach might enable the devices described here to be translated to human experiments, although further development will be required to enable such designs.

Modifications made to the manuscript (page 37 line 918):

Mobile Operation Proof of Concept

Proof of concept experiments were conducted using a small coil integrated with an NFC reader IC (STMicroelectronics, ST25R3911B-DISCO) and a 2 k Ω shunt resistor to match the operational load of the system during continuous operation while voltage measurement were taken on a digital multimeter (AstroAI, DM130B) to calculate harvested power.

Additions made to the supplementary figures (Supplementary figure 36):

Supplementary Fig. 36 Implant inside a wearable coil integrated with an NFC reader IC.

Comment 4:

Minor typos: Fig 2 contains spelling errors: receiver, wireless

Our Response: Thank you for pointing these out. We have made changes to the figure to correct for the spelling mistakes.

Modifications made to the main figure (Figure 2):

Figure 2. Characterization of the antenna design and electrical device properties
(a) Rendering of treadmill cage design with a dual loop transmitter antenna and infrared receivers. **(b)** Rendering of co-planar resonant antenna design showing the transmitter and receiver antenna coupling in Finite Element simulations of electromagnetic field. **(c)** Corresponding power vs. load curve in the center of 56 cm x 16 cm arena with RF input power of 2 W for the 3-coil WPT and 2-coil WPT. **(d)** Power harvesting capability vs. angular misalignments for both 2 W and 5 W of RF power. **(e)** Finite element simulations and measurements (displayed in numeric datapoint representing the hottest location measured) at steady state in saline solution when powered with 6 W of RF power.

REVIEWER COMMENTS

Reviewer #1 (Remarks to the Author):

Review

The authors have addressed many of my comments and have provided additional experiments in support of their claims.

I appreciate the authors' willingness to support their arguments by performing additional experiments.

According to the authors "The main goal for this manuscript is to showcase a platform that can operate over many weeks at a time in freely behaving animals and stimulate multiple substrates across a range of currents and voltages." Overall, my main criticism is that I still do not feel that the authors have rigorously characterised their platform to be able to make such claim, in particular regarding the stability of the implants (electrode materials and overall stability of encapsulation during the chronic stimulation scenarios proposed here).

The authors acknowledge that electrode materials are not optimised for this, but this is part of the platform, right? Simply substituting the material with another one, e.g. platinum, as suggested by the authors, implies new fabrication techniques, that come with a range of new challenges for which new characterisations are needed since the material interfaces will be different and stability is not guaranteed for the overall platform.

Two specific comments regarding this point:

Related to comments 8 and 9: characterisations regarding electrode stability are still performed at current values that are below what the platform claims to be able to deliver, and for shorter durations (and we already see electrode degradation).

Comment 10: accelerated lifetime tests: are these in PBS? At room temperature? Please add these details in the manuscript. The authors cannot simply extrapolate 14 days to 56 'operational days' only based on the total number of stimulation cycles. The device will degrade in vivo over time even if it passively stays there. Acceleration factors for lifetime testing include electric field as well as temperature, and these tests seem to have been performed at temperatures lower than body.

In general, I feel the authors have shown that this platform can wirelessly power stimulating implants in freely moving animals under certain scenarios/experimental setups, which are certainly of relevance for neuroscience research. However, the claim it is capable of delivering the chronic, let alone uninterrupted, stimulation protocols across the currents and voltages that has been claimed, has not been sufficiently demonstrated here (with main concerns regarding the implant degradation, see above, remaining unanswered).

Overall, I feel the authors need to either (1) present more data supporting their bold claims I pointed out above, or (2) modify those claims to more accurately reflect the work presented here.

Minor points:

There are still some typos in the text, also in the new parts added by the authors, it needs further proof reading.

Fig. 7g-this is not increasing size. Also, the figure is not clear, what is in the x-axis, other than the 3 categories?

Fig. 4b: impedance: this cannot be ohms, it's likely in kohms

Reviewer #2 (Remarks to the Author):

I found the revised manuscript satisfactory and I have no further comments.

Reviewer #3 (Remarks to the Author):

The authors have addressed my recommendation very thoroughly and I have no further suggestions or concerns. The article represents a significant advancement in electrical stimulation current technology and is supported by extensive results. The technical material is rigorous and sufficiently detailed, and the article now provides the necessary research context to understand the contribution fully.

REVIEWER COMMENTS

Reviewer #1 (Remarks to the Author):

General Comments:

The authors have addressed many of my comments and have provided additional experiments in support of their claims.

I appreciate the authors' willingness to support their arguments by performing additional experiments.

Our Response

We thank the reviewer for their comments and positive assessment of the revised manuscript.

Comment 1

According to the authors "The main goal for this manuscript is to showcase a platform that can operate over many weeks at a time in freely behaving animals and stimulate multiple substrates across a range of currents and voltages." Overall, my main criticism is that I still do not feel that the authors have rigorously characterised their platform to be able to make such claim, in particular regarding the stability of the implants (electrode materials and overall stability of encapsulation during the chronic stimulation scenarios proposed here).

The authors acknowledge that electrode materials are not optimised for this, but this is part of the platform, right? Simply substituting the material with another one, e.g. platinum, as suggested by the authors, implies new fabrication techniques, that come with a range of new challenges for which new characterisations are needed since the material interfaces will be different and stability is not guaranteed for the overall platform.

Our Response

We appreciate the reviewer's comment and have further revised the manuscript to address this point. We first note that our platform is explicitly designed to be modular with respect to the electrode interface that is used. None of the fabrication techniques used to create the core components of the device (i.e. the antenna, the interconnects between the components, etc...) depend on the electrodes used. This design capability is demonstrated in the current manuscript by the use of the muscle stimulation electrodes. Those electrodes were connected to the device with conductive epoxy and encapsulated with parylene after the other components were fabricated and encapsulated. It would have been straightforward to use other electrodes instead of the stainless steel electrodes used here without changing fabrication techniques or device performance. For example, in published work we have used the same approach to bond thin film probes^{R1} for brain stimulation that operate successfully for 36 days. The stability of our platform demonstrated in the manuscript should therefore be independent of the specific electrodes used in the platform.

Further, as also mentioned by the reviewer, there is an extensive literature characterizing electrode performance and stability for different materials and geometries. In fact, the characterizations of the muscle and spinal electrodes included in the revised manuscript are essentially what we would have expected based on that previous literature. Given our modular design, we therefore expect that performing additional experiments characterizing the performance of different electrode materials would not alter the conclusions of our study.

We have revised the manuscript further to clarify these points and emphasize the ability of our platform to be used flexibly with different electrode materials and designs.

Reference:

[R1] Burton, A., Won, S.M., Sohrabi, A.K. et al. Wireless, battery-free, and fully implantable electrical neurostimulation in freely moving rodents. *Microsyst Nanoeng* 7, 62 (2021).

Modification to the Manuscript

Page 13 Line 324

Note that the ~~core technology and fabrication processes of this device are based on the~~ modular designs ~~that of the device~~ enables a high degree of flexibility of neural interfaces: ~~including commercially available solutions or custom electrodes.~~ For example, electrode leads could, in principle, be used to stimulate intracortical electrodes or cuff electrodes on peripheral nerves. The scalable design of the monolithic device body enables free positioning of stimulation sites within the device footprint as shown in Supplementary Fig. 12a,b,c to target sites distributed across the rostrocaudal and mediolateral extent of the spinal cord.

Page 15 Line 388

These results are generally expected for the electrode sizes and materials used here and are independent from the overall performance of the device. Devices described implanted in this study generally remain functional for 4-6 weeks. Importantly, ~~because of our system's modular design, the stability of the device is independent of the electrode materials or designs utilized~~ other electrode designs (e.g., platinum vs. gold). Thus, electrodes that have better chronic performance can be easily implemented ~~with minimal changes in overall design or performance,~~ without requiring changes to the fabrication techniques or compromising electronic device performance¹⁶.

Comment 2

Related to comments 8 and 9: characterisations regarding electrode stability are still performed at current values that are below what the platform claims to be able to deliver, and for shorter durations (and we already see electrode degradation).

Our Response

In the in vitro experiments, we were interested in characterizing the performance of electrodes under conditions that they might experience in chronic applications. Simply applying the maximum current capability to spinal electrodes, which is substantially higher than any other wireless battery free device to the spinal stimulation electrodes, is not appropriate. The currents in vivo would result extreme pain for the subject and damage of the spinal cord if applied chronically, hence they would not be used in real life applications. E.g. as shown in Figure 7b, spinal responses saturated at around 300uA. The high current delivery is reserved for the muscle stimulation where much higher currents are applied more routinely. As shown in our in

vitro experiments the muscle electrodes are stable at these currents and voltages. To highlight this we have made additional modifications to the manuscript.

Modification to the Manuscript

Page 14 Line 336

An 8-bit DAC sets the resolution of the current source which is capable of delivering up to ± 24 mA. Setting the stimulation current range to ± 4.7 mA allows for the balance of fine current control ($40\mu\text{A}$) (Fig. 3d) with a dynamic range that is capable of both muscle and spinal stimulation. **It should be noted that while high currents can be delivered to all channels, this may result in injury or animal discomfort in spinal cord applications; the primary intended use for high current stimulation is for muscle stimulation.** For higher resolution, a lower dynamic range can be chosen by changing feedback resistors on the device; e.g., with a dynamic range of ± 1.8 mA the resolution is $7\mu\text{A}$ (Supplementary Fig. 14a).

Page 15 Line 396

Stimulation using constant current mode (1 hour sessions repeated 5 days a week for 16 weeks) using an accelerated lifetime test (9 million pulses at 1 mA at 1000 Hz at room temperature **matching parameters used for spinal cord stimulation**⁷⁵) which showed similar degradation (Fig. 3e) and compares to results from literature (Supplementary Fig. 18a,b,c)⁷⁶.

Comment 3

Comment 10: accelerated lifetime tests: are these in PBS? At room temperature? Please add these details in the manuscript. The authors cannot simply extrapolate 14 days to 56 'operational days' only based on the total number of stimulation cycles. The device will degrade in vivo over time even if it passively stays there. Acceleration factors for lifetime testing include electric field as well as temperature, and these tests seem to have been performed at temperatures lower than body.

Our Response

As requested by the reviewer, we have added information about the exact conditions of the test in the methods section of the manuscript. The chemical composition of the PBS and temperature of the tests have been documented. The reviewer is of course correct that these accelerated lifetime experiments cannot fully replicate in vivo conditions. They are supplemental to the other experiments we describe showing the long term stability of the device in in vivo conditions. Note that it is not possible to perform the exact experiments requested by the reviewer since it would require stimulating animals at high current levels that would induce pain or other discomfort to the animals. We believe that the experiments that we have included in the manuscript address the essential points raised by the reviewer and demonstrate the capabilities of our device. To further highlight conditions and relevant conclusions we have modified the manuscript to make these as clear as possible.

Modification to the Manuscript

Page 15 Line 375

We test spinal electrodes both within the water window of the electrode material (gold ($<\pm 1.2$ V)) and above the water window at 1 kHz for 14 days **at room temperature** (Supplementary Fig. 17a,c,e). This reflects more than 1 billion stimulation pulses; given a typical FES stimulation rate of 50Hz, **Not accounting for *in vivo* physiological factors, this corresponds to more than 40 results in several months of real experimental time.**

Page 32 Line 785

To evaluate the chronic stability of spinal and muscle electrodes, a series of tests was conducted with experimental parameters spanning the water window of the gold electrode (± 0.2 V, ± 0.6 V, ± 0.8 V, ± 1.0 V, and ± 1.2 V) which were generated by a function generator (Siglent, SDG 1032X), while monitoring changes in impedance with a digital storage oscilloscope (Siglent, SDS 1202X-E) over a 100 Ω shunt resistor at a 1kHz biphasic stimulation with 100% duty cycle (50% positive and 50% negative pulse), performed **at room temperature** in a 0.1M phosphate buffer saline (**003002, Gibco, Life Technologies**) solution. Additionally, the spinal electrode was subjected to testing at ± 1.4 V, a voltage exceeding the water window of gold to probe electrode lifetime outside of safe stimulation limits.

Comment 4

In general, I feel the authors have shown that this platform can wirelessly power stimulating implants in freely moving animals under certain scenarios/experimental setups, which are certainly of relevance for neuroscience research. However, the claim it is capable of delivering the chronic, let alone uninterrupted, stimulation protocols across the currents and voltages that has been claimed, has not been sufficiently demonstrated here (with main concerns regarding the implant degradation, see above, remaining unanswered).

Overall, I feel the authors need to either (1) present more data supporting their bold claims I pointed out above, or (2) modify those claims to more accurately reflect the work presented here.

Our Response

We acknowledge the basic point raised by the reviewer here, i.e., that we have not fully demonstrated the capabilities of our platform across the range of potential experimental paradigms in which it might be used. Such a demonstration would clearly require a large number of experiments and a large number of animals, ideally performed by multiple labs independently, that we believe are out of the scope of the current manuscript. We do want to highlight that we show capabilities in multiple settings in a range of cage sizes, with a range of hardware in the cage, and with different electrode interfaces. This is more than device papers in the field usually provide (the standard is one or at most 2 application scenarios). We therefore believe that our current manuscript clearly demonstrates the potential of our device, its suitability for chronic applications, and its significant advance relative to the current state-of-the-art. Nonetheless, we have revised the manuscript to reflect the issues raised by the reviewer.

Modification to the Manuscript

Page 27 Line 647

These capabilities are illustrated by the production of consistent amplitude movements from both spinal and muscle sites using similar ranges of currents over several weeks, demonstrating the consistent ability of devices to generate currents and voltages necessary for neuromodulation or functional restoration. Chronic stability of devices is demonstrated by stable delivery of stimuli in freely behaving subjects in environments relevant to FES research and enabling operational durations that are not achievable with battery powered systems. **It should be emphasized that additional experiments across a broader spectrum of scenarios beyond the three we have presented here are required to comprehensively determine the full potential of the chronic performance of the device. Nonetheless,** Characteristics of the devices demonstrated here make them suitable candidates for automated investigation of rehabilitation of neural circuits on clinically relevant timescales and with adequate statistical power to investigate underlying mechanisms that currently prevent widespread clinical adoption.

Page 2 Line 34

This improved performance enables multichannel, biphasic, current-controlled operation at clinically relevant voltage and current ranges with digital control and telemetry in freely behaving animals. **Preliminary chronic results indicate,** implanted devices remain operational over 6 weeks in both intact and spinal cord injured rats and are capable of producing fine control of spinal and muscle stimulation.

Page 5 Line 124

We demonstrate the capabilities of this device in one biomedical application: electrical stimulation of the spinal cord and muscles in freely behaving and spinal cord injured rats, showing that this device is capable of producing functional limb movements from spinal and muscle stimulation for over 6 weeks **indicating good chronic stability and laying the foundation for further validation studies.**

Comment 5

Minor points:

There are still some typos in the text, also in the new parts added by the authors, it needs further proof reading.

Our Response

We appreciate this comment of the reviewer in catching these typos and provide correction.

Page 5 Line 109

The circuit design incorporates off-the-shelf components providing scalable manufacturing at a cost that enables single use deployment which is critical for chronic implantation that make reuse impossible or difficult.

Page 5 Line 129

In this work, wireless, battery-free designs and a monolithic device structure, **areis** leveraged to create a soft, biocompatible, flexible device class that can be fully implanted in highly mobile areas of a rat, such as the back and hind leg.

Page 6 Line 152

The 20 mm **××** 40 mm device platform consists of a monolithic, dual-sided, flexible circuit board (height: 100 μm) composed of two rolled annealed copper layers separated by a polyimide film, enabling scalable manufacturing with off-the-shelf components.

Page 9 Line 214

This design can also be easily implanted, enabling it to be used as a scalable platform for chronic applications. This design also requires only a small number of passive components, resulting in small device footprints and high efficiencies compared to active approaches such as maximum powerpoint tracking and management IC's. The device is optimized to operate in a treadmill enclosure (56 cm **××** 16 cm) with a 2-turn transmitter antenna with 4 cm spacing between the turns, and is optimized to match the range of vertical positions of the implanted device during normal behavior (Supplementary Fig. 3).

Page 10 Line 226

Figure 2b and Supplementary Fig. 5a and 5b illustrate the simulated B-field of the 3-antenna system, in which the receiver antenna and the resonator are co-located on the top and bottom sides of the implant.

Page 11 Line 258

The device described here uses a larger transmitter antenna (56 cm \times 16 cm; 896 cm^2) and similar receiver (4 cm **××** 2 cm; 8 cm^2) as compared to receivers used in a previous work⁵⁹ (3.5 cm \times 2.5 cm; 8.75 cm^2), however this device is capable of harvesting 250 mW using 4 W of RF power in comparison to 50 mW at 4W of RF power in that previous study⁵⁹.

Page 12 Line 301

(c) Corresponding power vs. load curve in the center of 56 cm **××** 16 cm arena with RF input power of 2 W for the 3-coil WPT and 2-coil WPT.

Page 14 Line 336

Setting the stimulation current range to ± 4.7 mA allows for the balance of fine current control (40 μA) (Fig. 3d) with a dynamic range that is capable of both muscle and spinal stimulation.

Page 16 Line 414

The encapsulation is thermally stressed in an accelerated rate test at 37 $^{\circ}\text{C}$, 60 $^{\circ}\text{C}$, and 90 $^{\circ}\text{C}$ showing an acceleration factor of 0.32 using Arrhenius scaling with an estimated device lifetime of 3 months at physiological temperatures (37 $^{\circ}\text{C}$) (Supplementary Fig. 19a-d).

Page 19 Line 470

Post-operative **Xx**-rays are used to monitor the placement and integrity of the implanted device (Fig. 5a).

Page 24 Line 596

The enclosure area is larger (16 cm **××** 56 cm vs 18 cm **××** 36 cm) and results in lowered average system voltage of 12.70 ± 1.0 V and a larger number of trials are uncompliant (52/219 trials).

Page 25 Line 607

As expected, voltage compliance is reduced in larger enclosures (29 cm ~~×~~ 36 cm); however, even in the largest enclosures sufficient power is harvested to enable stimulation at high voltages (9.9 ± 2.8 V).

Page 27 Line 621

(g) Distributions of voltage compliances measured in three ~~different~~ enclosures of increasing sizes: large cage (29 cm ~~×~~ 36 cm), treadmill (16 cm ~~×~~ 56 cm), small cage (18 cm ~~×~~ 36 cm).

Page 33 Line 821

Muscles overlying the vertebrae were separated from lateral sides of spinous processes and muscle and connective tissue ~~were~~ removed from the transverse processes.

Page 34 Line 842

The incision was closed with sutures, and analgesics (Meloxicam, Buprenorphine-SR) ~~were~~ given for three days after surgery. The bladder was manually expressed twice daily.

Page 36 Line 896

We first determined the threshold current, taken as the first current~~ly~~ level capable of eliciting a visible hindlimb movement.

Page 36 Line 905

Studies were performed in a large cage (29 ~~×~~ 36 cm²), a moving treadmill (16 ~~×~~ 56 cm²) and a small cage (18 ~~×~~ 36 cm²)

Page 36 Line 910

The infrared receiver was positioned over the cage center to record stimulation parameters including intensiti~~es~~y, channels, voltages and compliances etc. from the implanted device while the animal was freely moving in the cages.

Fig. 7g-this is not increasing size. Also, the figure is not clear, what is in the x-axis, other than the 3 categories?

Our Response

We are grateful to the reviewer for catching the typo and have revised the text to state it is decreasing. We have added a scale bar to the violin plot to clearly display the frequencies for the compliance voltages during stimulation for each cage dimensions. We have also clarified the text describing the figure to clarify what is shown. The graph is a standard violin plot, which provides numeric data for one or more groups using density curves. The width of each curve corresponds with the approximate frequency of data points in each region. It's a standard plot used in neuroscience research and also works well for displaying the compliance voltages during the experiment. We appreciate that this may not be a standard graph for the engineering field, so we provide an explanation of this graph type to the supplementary information and have provided alternative graphs in the supplementary data.

Modification to the Manuscript

Page 25 Line 605

The average system voltage for day 25 is 16.2 ± 2.4 V and 15.8 ± 2.8 V for day 28 with only intermittent drops in system voltage. System performance in a variety of cages during free behavior is displayed in Fig. 7g (detailed plots shown in Supplementary Fig. 35). As expected, voltage compliance is reduced in larger enclosures (29 cm x 36 cm); however, even in the largest enclosures sufficient power is harvested to enable stimulation at high voltages (9.9 ± 2.8 V).

Addition to the Supplementary Figure (Supplementary Figure 35):

Supplementary Fig. 35. Histograms of voltage compliance distributions in three enclosures of increasing size. (a) Large cage (29 cm × 36 cm). (b) Treadmill (16 cm × 56 cm). (c) Small cage (18 cm × 36 cm).

Fig. 4b: impedance: this cannot be ohms, it's likely in kohms

Our Response

We appreciate the reviewer catching this. As described in the text, this is trace impedance, so we only look at the conductivity of the trace, not of the electrode. We have revised this x-axis label to 'Trace Impedance' and have also modified the manuscript to make this clear.

Modification to the Main Figure (Figure 4)

Fig. 4. Material overview and mechanical characterization of the device. (a) Exploded view rendering of each material layer used in the device, the return electrode, and the spinal electrode. **(b)** Change in trace impedance of the serpentine interconnect during cyclic loading electrode measured at 1kHz. **(c)** Strain stress curve of the serpentine structure for the spinal electrode, and **(d)** the return electrode.

Reviewer #2 (Remarks to the Author):

I found the revised manuscript satisfactory and I have no further comments.

Our Response

We thank the reviewer for their comments and positive assessment of the manuscript.

Reviewer #3 (Remarks to the Author):

The authors have addressed my recommendation very thoroughly and I have no further suggestions or concerns. The article represents a significant advancement in electrical stimulation current technology and is supported by extensive results. The technical material is rigorous and sufficiently detailed, and the article now provides the necessary research context to understand the contribution fully.

Our Response

We thank the reviewer for their comments and positive assessment of the manuscript.

REVIEWERS' COMMENTS

Reviewer #1 (Remarks to the Author):

I have no further comments.